# Ising Model Selection Using $\ell_1$-Regularized Linear Regression: A Statistical Mechanics Analysis

**Xiangming Meng**$^*$
Institute for Physics of Intelligence
The University of Tokyo
7-3-1, Hongo, Tokyo 113-0033, Japan
meng@g.ecc.u-tokyo.ac.jp

**Tomoyuki Obuchi**
Department of Systems Science
Kyoto University
Kyoto 606-8501, Japan
obuchi@i.kyoto-u.ac.jp

**Yoshiyuki Kabashima**
Institute for Physics of Intelligence
The University of Tokyo
7-3-1, Hongo, Tokyo 113-0033, Japan
kaba@phys.s.u-tokyo.ac.jp

## Abstract

We theoretically analyze the typical learning performance of $\ell_1$-regularized linear regression ($\ell_1$-LinR) for Ising model selection using the replica method from statistical mechanics. For typical random regular graphs in the paramagnetic phase, an accurate estimate of the typical sample complexity of $\ell_1$-LinR is obtained. Remarkably, despite the model misspecification, $\ell_1$-LinR is model selection consistent with the same order of sample complexity as $\ell_1$-regularized logistic regression ($\ell_1$-LogR), i.e., $M = \mathcal{O}\left(\log N\right)$, where $N$ is the number of variables of the Ising model. Moreover, we provide an efficient method to accurately predict the non-asymptotic behavior of $\ell_1$-LinR for moderate $M, N$, such as precision and recall. Simulations show a fairly good agreement between theoretical predictions and experimental results, even for graphs with many loops, which supports our findings. Although this paper mainly focuses on $\ell_1$-LinR, our method is readily applicable for precisely characterizing the typical learning performances of a wide class of $\ell_1$-regularized $M$-estimators including $\ell_1$-LogR and interaction screening.

## 1 Introduction

The advent of massive data across various scientific disciplines has led to the widespread use of undirected graphical models, also known as Markov random fields (MRFs), as a tool for discovering and visualizing dependencies among covariates in multivariate data [1]. The Ising model, originally proposed in statistical physics, is one special class of binary MRFs with pairwise potentials and has been widely used in different domains such as image analysis, social networking, gene network analysis [2, 3, 4, 5, 6, 7]. Among various applications, one fundamental problem of interest is called Ising model selection, which refers to recovering the underlying graph structure of the original Ising model from independent, identically distributed (i.i.d.) samples. A variety of methods have been proposed [8, 9, 10, 11, 12, 13, 14, 15, 16, 17, 18], demonstrating the possibility of successful Ising model selection even when the number of samples is smaller than that of the variables. Notably, it has been demonstrated that for the $\ell_1$-regularized logistic regression ($\ell_1$-LogR) [10, 16] and interaction screening (IS) [14, 15] estimators, $M = \mathcal{O}\left(\log N\right)$ samples suffice for an Ising model with $N$ spins under certain assumptions, which is consistent with respect to (w.r.t.) previously established

---

$^*$Corresponding author.

information-theoretic lower-bound [11]. Both $\ell_1$-LogR and IS are $\ell_1$-regularized $M$-estimators [19] with logistic and IS objective (ISO) loss functions, respectively.

In this paper, we focus on one simpler linear estimator called $\ell_1$-regularized linear regression ($\ell_1$-LinR) and theoretically investigate its *typical* learning performance using the powerful replica method [20, 21, 22, 23] from statistical mechanics. The $\ell_1$-LinR estimator, widely known as least absolute shrinkage and selection operator (LASSO) [24] in statistics and machine learning, is considered here mainly for two reasons. On the one hand, it is one representative example of model misspecification since the quadratic loss of $\ell_1$-LinR does not match the true log-conditional-likelihood as $\ell_1$-LogR, nor does it have the interaction screening property as IS. On the other hand, as one of the most popular linear estimator, $\ell_1$-LinR is more computationally efficient than $\ell_1$-LogR and IS, and thus it is of practical importance to investigate its learning performance for Ising model selection. Since it is difficult to obtain results for general graphs, as a first step we consider the random regular (RR) graphs $\mathcal{G}_{N,d,K_0}$ in the paramagnetic phase [23], where $\mathcal{G}_{N,d,K_0}$ denotes the ensemble of RR graphs with constant node degree $d$ and uniform coupling strength $K_0$ on the edges.

## 1.1 Contributions

The main contributions are summarized as follows. First, we obtain an accurate estimate of the *typical* sample complexity of $\ell_1$-LinR for Ising model selection for *typical* RR graphs in the paramagnetic phase, which, remarkably, has the same order as $\ell_1$-LogR. Specifically, for a typical RR graph $G \in \mathcal{G}_{N,d,K_0}$, using $\ell_1$-LinR with a regularization parameter $0 < \lambda < \tanh(K_0)$, one can consistently reconstruct the structure with $M > \frac{c(\lambda, K_0) \log N}{\lambda^2}$ samples, where $c(\lambda, K_0) = \frac{2(1 - \tanh^2(K_0) + d\lambda^2)}{1 + (d-1) \tanh^2(K_0)}$. The accuracy of our *typical* sample complexity prediction is verified by its excellent agreement with experimental results. To the best of our knowledge, this is the first result that provides an accurate *typical* sample complexity for Ising model selection. Interestingly, as $\lambda \to \tanh(K_0)$, a lower bound $M > \frac{2 \log N}{\tanh^2(K_0)}$ of the typical sample complexity is obtained, which has the same scaling as the information-theoretic lower bound $M > \frac{c' \log N}{K_0^2}$ [11] for some constant $c'$ at high temperatures (i.e., small $K_0$) since $\tanh(K_0) = \mathcal{O}(K_0)$ as $K_0 \to 0$.

Second, we provide a computationally efficient method to precisely predict the typical learning performance of $\ell_1$-LinR in the non-asymptotic case with moderate $M, N$, such as precision, recall, and residual sum of square (RSS). Such precise non-asymptotic predictions of $\ell_1$-LinR for Ising model selection have been unavailable even for $\ell_1$-LogR [10, 16] and IS [14, 15], nor are they the same as previous asymptotic results of $\ell_1$-LinR assuming fixed $\alpha \equiv M/N$ [25, 26, 27, 28]. Moreover, although our theoretical analysis is based on a tree-like structure assumption, experimental results on two dimensional (2D) grid graphs also show a fairly good agreement, indicating that our theoretical result can be a good approximation even for graphs with many loops.

Third, while this paper mainly focuses on $\ell_1$-LinR, our method is readily applicable to a wide class of $\ell_1$-regularized $M$-estimators [19], including $\ell_1$-LogR [10] and IS [14, 15]. Thus, an additional technical contribution is providing a generic approach for precisely characterizing the typical learning performances of various $\ell_1$-regularized $M$-estimators for Ising model selection. Although the replica method from statistical mechanics is non-rigorous, our results are conjectured to be correct, which is supported by their excellent agreement with the experimental results. Additionally, several technical advances we propose in this paper, e.g., the entropy term computation by averaging over the Haar measure and the modification of EOS to address the finite-size effect, might be of general interest to those who use the replica method as a tool for performance analysis.

## 1.2 Related works

There has been some earlier works on the analysis of Ising model selection (also known as the inverse Ising problem) using the replica method [4, 5, 6, 7, 29] from statistical mechanics. For example, in [6], the performance of the pseudo-likelihood (PL) method [30] is studied. However, instead of graph structure learning, [6] focuses on the problem of parameter learning. Then, [7] extends the analysis to the Ising model with sparse couplings using logistic regression without regularization. The recent work [29] analyzes the performance of $\ell_2$-regularized linear regression but the techniques invented there are not applicable to $\ell_1$-LinR since the $\ell_1$-norm breaks the rotational invariance property.

Regarding the study of $\ell_1$-LinR (LASSO) under model misspecification, the past few years have seen a line of research in the field of signal processing with a specific focus on the single-index

model [31, 32, 27, 33, 34]. These studies are closely related to ours but there are several important differences. First, in our study, the covariates are generated from an Ising model rather than a Gaussian distribution. Second, we focus on model selection consistency of $\ell_1$-LinR while most previous studies consider estimation consistency except [33]. However, [33] only considers the classical asymptotic regime while our analysis includes the high-dimensional setting where $M \ll N$.

As far as we have searched, there is no earlier study of $\ell_1$-LinR estimator for Ising model selection, though some are found for Gaussian graphical models [35, 36]. One closely related work [15] states that at high temperatures when the coupling magnitude is approaching zero, both logistic and ISO losses can be approximated by a quadratic loss. However, their claim is only restricted to the very small magnitude near zero while our analysis extends the validity range to the whole paramagnetic phase. Moreover, they evaluate the minimum number of samples necessary for consistently reconstructing "arbitrary" Ising models, which, however, seems much larger than that actually needed. By contrast, we provide the first accurate assessment of *typical* sample complexity for consistently reconstructing *typical* samples of Ising models defined over the RR graphs. Furthermore, [15] does not provide precise predictions of the non-asymptotic learning performance as we do.

## 2 Background and Problem Setup

### 2.1 Ising Model

Ising model is one special class of MRFs with pairwise potentials and each variable takes binary values [22, 23], which is one classical model from statistical physics. The joint probability distribution of an Ising model with $N$ variables (spins) $\boldsymbol{s} = (s_i)_{i=0}^{N-1} \in \{-1, +1\}^N$ has the form

$$P_{\text{Ising}}(\boldsymbol{s}|\boldsymbol{J}^*) = \frac{1}{Z_{\text{Ising}}(\boldsymbol{J}^*)} \exp \left\{ \sum_{i<j} J_{ij}^* s_i s_j \right\}, \tag{1}$$

where $Z_{\text{Ising}}(\boldsymbol{J}^*) = \sum_{\boldsymbol{s}} \exp \left\{ \sum_{i<j} J_{ij}^* s_i s_j \right\}$ is the partition function and $\boldsymbol{J}^* = \left( J_{ij}^* \right)_{i,j}$ are the original couplings, respectively. In general, there are also external fields but here they are assumed to be zero for simplicity. The structure of Ising model can be described by an undirected graph $G = (\mathtt{V}, \mathtt{E})$, where $\mathtt{V} = \{0, 1, ..., N-1\}$ is a collection of vertices at which the spins are assigned, and $\mathtt{E} = \left\{ (i,j) \,|\, J_{ij}^* \neq 0 \right\}$ is a collection of undirected edges, i.e., $J_{ij}^* = 0$ for all pairs of $(i,j) \notin \mathtt{E}$. For each vertex $i \in \mathtt{V}$, its neighborhood is defined as the subset $\mathcal{N}(i) \equiv \{j \in \mathtt{V} \,|\, (i,j) \in \mathtt{E}\}$.

### 2.2 Neighborhood-based $\ell_1$-regularized linear regression ($\ell_1$-LinR)

The problem of Ising model selection refers to recovering the graph $G$ (edge set $\mathtt{E}$), given $M$ i.i.d. samples $\mathcal{D}^M = \left\{ \boldsymbol{s}^{(1)}, ..., \boldsymbol{s}^{(M)} \right\}$ from the Ising model. While the maximum likelihood method has nice properties of consistency and asymptotic efficiency, it suffers from high computational complexity. To tackle this difficulty, several local learning algorithms have been proposed, notably the $\ell_1$-LogR estimator [10] and IS estimator [14]. Both $\ell_1$-LogR and IS optimize a regularized local cost function $\ell(\cdot)$ for each spin i.e., $\forall i \in \mathtt{V}$,

$$\hat{\boldsymbol{J}}_{\backslash i} = \underset{\boldsymbol{J}_{\backslash i}}{\arg\min} \left[ \frac{1}{M} \sum_{\mu=1}^{M} \ell \left( s_i^{(\mu)} h_{\backslash i}^{(\mu)} \right) + \lambda \left\| \boldsymbol{J}_{\backslash i} \right\|_1 \right], \tag{2}$$

where $h_{\backslash i}^{(\mu)} \equiv \sum_{j \neq i} J_{ij} s_j^{(\mu)}$, $\boldsymbol{J}_{\backslash i} \equiv (J_{ij})_{j(\neq i)}$, and $\|\cdot\|_1$ denotes the $\ell_1$ norm. Specifically, $\ell(x) = \log \left( 1 + e^{-2x} \right)$ for $\ell_1$-LogR and $\ell(x) = e^{-x}$ for IS, which correspond to the minus log conditional distribution [10] and the ISO [14], respectively. Consequently, the problem of recovering the edge set $\mathtt{E}$ is equivalently reduced to local neighborhood selection, i.e., recovering the neighborhood set $\mathcal{N}(i)$ for each vertex $i \in \mathtt{V}$. In particular, given the estimates $\hat{\boldsymbol{J}}_{\backslash i}$ in (2), the neighborhood set of vertex $i$ can be estimated via the nonzero coefficients, i.e.,

$$\hat{\mathcal{N}}(i) = \left\{ j | \hat{J}_{ij} \neq 0, j \in \mathtt{V} \backslash i \right\}, \ \forall i \in \mathtt{V}. \tag{3}$$

In this paper, we focus on one simple linear estimator, termed as the $\ell_1$-LinR estimator, i.e., $\forall i \in \mathtt{V}$,

$$\hat{\boldsymbol{J}}_{\backslash i} = \underset{\boldsymbol{J}_{\backslash i}}{\arg\min} \left[ \frac{1}{2M} \sum_{\mu=1}^{M} \left( s_i^{(\mu)} - h_{\backslash i}^{(\mu)} \right)^2 + \lambda \left\| \boldsymbol{J}_{\backslash i} \right\|_1 \right], \ \forall i \in \mathtt{V}, \tag{4}$$

which, recalling that $s_i^{(\mu)} \in \{-1, +1\}$, corresponds to a quadratic loss $\ell(x) = \frac{1}{2}(x-1)^2$ in (2). The neighorbood set for each vertex $i \in V$ is estimated in the same way as (3). Interestingly, the quadratic loss used in (4) implies that the postulated conditional distribution is Gaussian and thus inconsistent with the true one, which is one typical case of model misspecification. Furthermore, compared with nonlinear estimators $\ell_1$-LogR and IS, the $\ell_1$-LinR estimator is more efficient to implement.

## 3 Statistical Mechanics Analysis

In this section, a statistical mechanics analysis of the $\ell_1$-LinR estimator is presented for *typical* RR graphs in the paramagnetic phase. Our analysis is applicable to any M-estimator of the form (2) and please refer to Appendix A for a unified analysis, including detailed results for the $\ell_1$-LogR estimator.

To characterize the structure learning performance, the precision and recall are considered:

$$Precision = \frac{TP}{TP + FP}, \quad Recall = \frac{TP}{TP + FN}, \tag{5}$$

where $TP$, $FP$, $FN$ denote the number of true positives, false positives, and false negatives in the estimated couplings, respectively. The concept of *model selection consistent* for an estimator is defined in Definition 1, which is also known as the *sparsistency* property [10].

**Definition 1.** *An estimator is called model selection consistent if both the associated precision and recall satisfy $Precision \to 1$ and $Recall \to 1$ as $M \to \infty$.*

Additionally, if one is further interested in the specific values of the estimated couplings, our analysis can also yield the residual sum of squares (RSS) for the estimated couplings.

Our theoretical analysis of the learning performance builds on the statistical mechanics framework. Contrary to the probably almost correct (PAC) learning theory [37] in mathematical statistics, statistical mechanics tries to describe the *typical* (as defined in Definition 2) behavior exactly rather than to bound the *worst case* which is likely to be over-pessimistic [38].

**Definition 2.** *"typical" means not just most probable but in addition the probability for situations different from the typical one can be made arbitrarily small as $N \to \infty$ [38].*

Similarly, when referring to *typical* RR graphs, we mean tree-like RR graphs, i.e., when seen from a random node, they look like part of an infinite tree, which are *typical* realizations from the uniform probability distribution on the ensemble of RR graphs.

### 3.1 Problem Formulation

For simplicity and without loss of generality, we focus on spin $s_0$. With a slight abuse of notation, we will drop certain subscripts in following descriptions, e.g., $\boldsymbol{J}_{\backslash i}$ will be denoted as $\boldsymbol{J}$ hereafter which represents a vector rather than a matrix. The basic idea of the statistical mechanical approach is to introduce the following Hamiltonian and Boltzmann distribution induced by the loss function $\ell(\cdot)$

$$\mathcal{H}(\boldsymbol{J}|\mathcal{D}^M) = \sum_{\mu=1}^{M} \ell\left(s_0^{(\mu)} h^{(\mu)}\right) + \lambda M \|\boldsymbol{J}\|_1, \tag{6}$$

$$P(\boldsymbol{J}|\mathcal{D}^M) = \frac{1}{Z} e^{-\beta \mathcal{H}(\boldsymbol{J}|\mathcal{D}^M)}, \tag{7}$$

where $Z = \int d\boldsymbol{J} e^{-\beta \mathcal{H}(\boldsymbol{J}|\mathcal{D}^M)}$ is the partition function, and $\beta \, (> 0)$ is the inverse temperature. In the zero-temperature limit $\beta \to +\infty$, the Boltzmann distribution (7) converges to a point-wise measure on the estimator (2). The macroscopic properties of (7) can be analyzed by assessing the free energy density $f(\mathcal{D}^M) = -\frac{1}{N\beta} \log Z$, from which, once obtained, we can evaluate averages of various quantities simply by taking its derivatives w.r.t. external fields [21]. In current case, $f(\mathcal{D}^M)$ depends on the predetermined randomness of $\mathcal{D}^M$, which plays the role of quenched disorder. As $N, M \to \infty$, $f(\mathcal{D}^M)$ is expected to show the *self averaging property* [21]: for typical datasets $\mathcal{D}^M$, $f(\mathcal{D}^M)$ converges to its average over the random data $\mathcal{D}^M$:

$$f = -\frac{1}{N\beta} [\log Z]_{\mathcal{D}^M}, \tag{8}$$

where $[\cdot]_{\mathcal{D}^M}$ denotes expectation over $\mathcal{D}^M$, i.e. $[\cdot]_{\mathcal{D}^M} = \sum_{\boldsymbol{s}^{(1)}, \dots, \boldsymbol{s}^{(M)}} (\cdot) \prod_{\mu=1}^{M} P_{\text{Ising}}(\boldsymbol{s}^{(\mu)}|\boldsymbol{J}^*)$. Consequently, one can analyze the typical performance of any $\ell_1$-regularized M-estimator of the form (2) via the assessment of (8), with $\ell_1$-LinR in (4) being a special case with $\ell(x) = \frac{1}{2}(x-1)^2$.

### 3.2 Replica computation of the free energy density

Unfortunately, computing (8) rigorously is difficult. For practically overcoming this difficulty, we resort to the powerful replica method [20, 21, 22, 23] from statistical mechanics, which is symbolized using the following identity

$$f = -\frac{1}{N\beta} \left[\log Z\right]_{\mathcal{D}^M} = -\lim_{n\to 0} \frac{1}{N\beta} \frac{\partial \log \left[Z^n\right]_{\mathcal{D}^M}}{\partial n}. \tag{9}$$

The basic idea is as follows. One replaces the average of $\log Z$ by that of the $n$-th power $Z^n$ which is analytically tractable for $n \in \mathbb{N}$ in the large $N$ limit, and constructs an analytically continuable expression from $\mathbb{N}$ to $\mathbb{R}$, then takes the limit $n \to 0$ by using the expression. Although the replica method is not rigorous, it has been empirically verified from extensive studies in disorder systems [22, 23] and also found useful in the study of high-dimensional models in machine learning [39, 40]. For more details of the replica method, please refer to [20, 21, 22, 23].

Specifically, with the Hamiltonian $\mathcal{H}\left(\boldsymbol{J}|\mathcal{D}^M\right)$, assuming $n \in \mathbb{N}$ is a positive integer, the replicated partition function $[Z^n]_{\mathcal{D}^M}$ in (9) can be written as

$$[Z^n]_{\mathcal{D}^M} = \int \prod_{a=1}^n d\boldsymbol{J}^a e^{-\beta\lambda M \sum_{a=1}^n \|\boldsymbol{J}^a\|_1} \left\{ \sum_{\boldsymbol{s}} P_{\text{Ising}}\left(\boldsymbol{s}|\boldsymbol{J}^*\right) \exp\left[-\beta \sum_{a=1}^n \ell\left(s_0 h^a\right)\right] \right\}^M, \tag{10}$$

where $h^a = \sum_j J_j^a s_j$ will be termed as *local field* hereafter, and $a$ (and $b$ in the following) is index variable of the replicas. The analysis below essentially depends on the distribution of $h^a$ but it is nontrivial. To resolve it, we take a similar approach as [7, 29] and introduce the following ansatz.

**Ansatz 1 (A1):** *Denote $\Psi = \{j|j \in \mathcal{N}(0)\}$ and $\bar{\Psi} = \{j|j = 1, ..., N-1, j \notin \mathcal{N}(0)\}$ as the active and inactive sets of spin $s_0$, respectively, then for a typical RR graph $G \in \mathcal{G}_{N,d,K_0}$ in the paramagnetic phase, i.e., $(d-1)\tanh^2(K_0) < 1$, the $\ell_1$-LinR estimator in (4) is a random vector determined by random realizations of $\mathcal{D}^M$ and obeys the following form*

$$\hat{J}_j = \begin{cases} \bar{J}_j + \frac{1}{\sqrt{N}} w_j, & j \in \Psi \\ \frac{1}{\sqrt{N}} w_j, & j \in \bar{\Psi} \end{cases} \tag{11}$$

*where $\bar{J}_j$ is the mean value of the estimator and $w_j$ is a random variable which is asymptotically zero mean with variance scaled as $\mathcal{O}(1)$.*

The consistency of Ansatz 1 is checked in Appendix B. Under Ansatz 1, the local fields $h^a$ can be decomposed as $h^a = \sum_{j\in\Psi} \bar{J}_j s_j + h_w^a$ where $h_w^a \equiv \sum_j \frac{1}{\sqrt{N}} w_j^a s_j$ is the "noise" part. According to the central limit theorem (CLT), $h_w^a$ can be approximated as multivariate Gaussian variables, which, under the replica symmetric (RS) ansatz [21], can be fully described by two order parameters

$$Q \equiv \frac{1}{N} \sum_{i,j} w_i^a C_{ij}^{\backslash 0} w_j^a, \quad q \equiv \frac{1}{N} \sum_{i,j} w_i^a C_{ij}^{\backslash 0} w_j^b, (a \neq b), \tag{12}$$

where $C^{\backslash 0} \equiv \{C_{ij}^{\backslash 0}\}$ is the covariance matrix of the original Ising model without the spin $s_0$. Since the difference between $C^{\backslash 0}$ and that with $s_0$ is not essential in the limit $N \to \infty$, hereafter the superscript $^{\backslash 0}$ will be discarded. As shown in Appendix A, for quadratic loss $\ell(x) = \frac{1}{2}(1-x)^2$ of $\ell_1$-LinR, the average free energy density (9) in the limit $\beta \to \infty$ can be computed as

$$f(\beta \to \infty) = -\text{Extr}\left\{-\xi + S\right\}, \tag{13}$$

where $\text{Extr}\{\cdot\}$ denotes the extremum operation w.r.t. relevant variables and $\xi, S$ denote the energy and entropy terms:

$$S = \lim_{\beta\to\infty} \lim_{n\to 0} \frac{1}{N\beta} \frac{\partial}{\partial n} \log I, \tag{14}$$

$$I = \int \prod_{a=1}^n dw^a \prod_{a=1}^n e^{-\lambda\beta\|w^a\|_1} \delta\left(\sum_{i,j} w_i^a C_{ij} w_j^a - NQ\right) \times \prod_{a<b} \delta\left(\sum_{i,j} w_i^a C_{ij} w_j^b - Nq\right), \tag{15}$$

$$\xi = \frac{\alpha\mathbb{E}_{s,z}\left(s_0 - \sum_{j\in\Psi} \bar{J}_j s_j - \sqrt{Q}z\right)^2}{2(1+\chi)} + \alpha\lambda \sum_{j\in\Psi} |\bar{J}_j|, \tag{16}$$

where $\alpha \equiv M/N, \chi \equiv \lim_{\beta \to \infty} \beta (Q - q)$, $\mathbb{E}_{s,z}(\cdot)$ denotes the expectation operation w.r.t. $z \sim \mathcal{N}(0,1)$ and $(s_0, \boldsymbol{s}_\Psi) \sim P_{\text{Ising}}(s_0, \boldsymbol{s}_\Psi | \boldsymbol{J}^*) \propto e^{s_0 \sum_{j \in \Psi} J_j^* s_j}$ [7]. For different losses $\ell(\cdot)$, the free energy results (13) only differ in the energy term $\xi$, which in general is non-analytical (e.g., logistic loss for $\ell_1$-LogR) but can be solved numerically. Please refer to Appendix A.3 for more details.

In contrast to the case of $\ell_2$-norm in [29], the $\ell_1$-norm in (15) breaks the rotational invariance property, i.e., $\|w^a\|_1 \neq \|Ow^a\|_1$ for general orthogonal matrix $O$, making it difficult to compute the entropy term $S$. To circumvent this difficulty, we employ an observation that, when considering the RR graph ensemble $\mathcal{G}_{N,d,K_0}$ as the coupling network of the Ising model, the orthogonal matrix $O$ diagonalizing the covariance matrix $C$ appears to be distributed from the Haar orthogonal measure [41, 42]. Thus, it is assumed that $I$ in (15) can be replaced by its average $[I]_O$ over the Haar-distributed $O$:

**Ansatz 2 (A2):** *Denote $C \equiv \mathbb{E}_{\boldsymbol{s}}[\boldsymbol{s}\boldsymbol{s}^T]$, where $\mathbb{E}_{\boldsymbol{s}}[\cdot] = \sum_{\boldsymbol{s}} P_{\text{Ising}}(\boldsymbol{s}|\boldsymbol{J}^*)(\cdot)$, as the covariance matrix of spin configurations $\boldsymbol{s}$. Suppose that the eigendecomposition of $C$ is $C = O\Lambda O^T$, where $O$ is the orthogonal matrix, then $O$ can be seen as a random sample generated from the Haar orthogonal measure and thus for typical graph realizations from $\mathcal{G}_{N,d,K_0}$, $I$ in (15) is equal to the average $[I]_O$.*

The consistency of Ansatz (A2) is numerically checked in Appendix C. Under Ansatz (A2), the entropy term $S$ in (14) can be alternatively computed as $\lim_{n \to 0} \frac{1}{N\beta} \frac{\partial}{\partial n} \log [I]_O$, as shown in Appendix A. Finally, under the RS ansatz, the average free energy density (9) in the limit $\beta \to \infty$ reads

$$f(\beta \to \infty) = -\underset{\Theta}{\text{Extr}} \left\{ \begin{array}{c} -\frac{\alpha}{2(1+\chi)}\mathbb{E}_{s,z}\left( \left(s_0 - \sum_{j \in \Psi} \bar{J}_j s_j - \sqrt{Q}z\right)^2 \right) - \lambda\alpha \sum_{j \in \Psi} |\bar{J}_j| \\ + (-ER + F\eta) G'(-E\eta) + \frac{1}{2}EQ - \frac{1}{2}F\chi + \frac{1}{2}KR - \frac{1}{2}H\eta \\ -\mathbb{E}_z \underset{w}{\min}\left\{ \frac{K}{2}w^2 - \sqrt{H}zw + \frac{\lambda M}{\sqrt{N}}|w| \right\} \end{array} \right\},$$

(17)

where $z \sim \mathcal{N}(0,1)$, and $G(x)$ is a function defined as

$$G(x) = -\frac{1}{2}\log x - \frac{1}{2} + \underset{\Lambda}{\text{Extr}}\left\{ -\frac{1}{2}\int \log(\Lambda - \gamma)\rho(\gamma)\,d\gamma + \frac{\Lambda}{2}x \right\},$$

(18)

and $\rho(\gamma)$ is the eigenvalue distribution (EVD) of the covariance matrix $C$, and $\Theta$ is a collection of macroscopic parameters $\Theta = \left\{ \chi, Q, E, R, F, \eta, K, H, \left\{\bar{J}_j\right\}_{j \in \Psi} \right\}$. For details of these macroscopic parameters and $\rho(\gamma)$, please refer to Appendix A and F, respectively. Note that in (17), apart from the ratio $\alpha \equiv M/N$, $N$ and $M$ also appear as $\lambda M/\sqrt{N}$ in the free energy result, which is different from previous results [7, 29, 39]. The reason is that, thanks to the $\ell_1$-regularization term $\lambda M \|\boldsymbol{J}\|_1$, the mean estimates $\left\{\bar{J}_j\right\}_{j \in \Psi}$ in the active set $\Psi$ and the noise $w$ in the inactive set $\bar{\Psi}$ essentially give different scaling contributions to the free energy density.

Although there are no analytic solutions, these macroscopic parameters in (17) can be obtained by numerically solving the corresponding equations of state (EOS) employing the physics terminology. Specifically, for the $\ell_1$-LinR estimator, the EOS can be obtained from the extremization condition in (17) as follows (for EOS of a general M-esimator and $\ell_1$-LogR, please refer to Appendix A.3):

$$\begin{cases} E = \frac{\alpha}{(1+\chi)}, \\ F = \frac{\alpha}{(1+\chi)^2}\left[\mathbb{E}_s\left(s_0 - \sum_{j \in \Psi} s_j \bar{J}_j\right)^2 + Q\right], \\ R = \frac{1}{K^2}\left[\left(H + \frac{\lambda^2 M^2}{N}\right)\text{erfc}\left(\frac{\lambda M}{\sqrt{2HN}}\right) - 2\lambda M\sqrt{\frac{H}{N}}\frac{1}{\sqrt{2\pi}}e^{-\frac{\lambda^2 M^2}{2HN}}\right], \\ E\eta = -\int \frac{\rho(\gamma)}{\tilde{\Lambda} - \gamma}\,d\gamma, \\ Q = \frac{F}{E^2} + R\tilde{\Lambda} - \frac{(-ER + F\eta)\eta}{\int \frac{\rho(\gamma)}{(\tilde{\Lambda} - \gamma)^2}d\gamma}, \\ K = E\tilde{\Lambda} + \frac{1}{\eta}, \\ \chi = \frac{1}{E} + \eta\tilde{\Lambda}, \\ H = \frac{R}{\eta^2} + F\tilde{\Lambda} + \frac{(-ER + F\eta)E}{\int \frac{\rho(\gamma)}{(\tilde{\Lambda} - \gamma)^2}d\gamma}, \\ \eta = \frac{1}{K}\text{erfc}\left(\frac{\lambda M}{\sqrt{2HN}}\right), \\ \bar{J}_j = \frac{\text{soft}(\tanh(K_0), \lambda(1+\chi))}{1 + (d-1)\tanh^2(K_0)}, j \in \Psi, \end{cases}$$

(19)

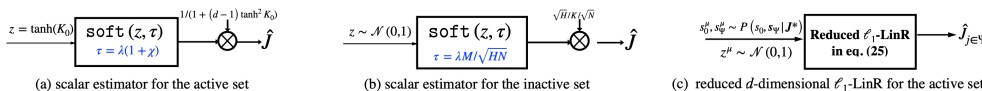

(a) scalar estimator for the active set      (b) scalar estimator for the inactive set      (c) reduced $d$-dimensional $\ell_1$-LinR for the active set

Figure 1: Equivalent low-dimensional estimators for high-dimensional $\ell_1$-LinR obtained from the statistical mechanics analysis. (a) and (b) are diagrams of the pair of scalar estimators in Eqs. (20) and (21). (c) is a schematic description of the modified estimator in Eq. (25) which takes into account the finite-size effect.

where $\tilde{\Lambda}$ satisfying $E\eta = -\int \frac{\rho(\gamma)}{\tilde{\Lambda}-\gamma} d\gamma$ is determined by the extremization condition in (18) and $\mathtt{soft}\,(z,\tau) = \mathtt{sign}\,(z)\,(|z|-\tau)_+$ is the soft-thresholding function. Once the EOS is solved, the free energy density defined in (8) is readily obtained.

### 3.3 High-dimensional asymptotic result

One important result of our replica analysis is that, as derived (see Appendix A.3) from the free energy result (17), the original high dimensional $\ell_1$-LinR estimator (4) is *decoupled* into a pair of scalar estimators, one for the active set and one for the inactive set, i.e.,

$$
\hat{J}_j = \begin{cases} \dfrac{\mathtt{soft}\,(\tanh\,(K_0)\,,\lambda\,(1+\chi))}{1+(d-1)\tanh^2\,(K_0)} \equiv \bar{J}_j, & j \in \Psi \quad (20) \\[2ex] \dfrac{\sqrt{H}}{K\sqrt{N}}\mathtt{soft}\left(z_j, \dfrac{\lambda M}{\sqrt{HN}}\right), & j \in \bar{\Psi} \quad (21) \end{cases}
$$

where $z_j \sim \mathcal{N}\,(0,1)\,, j \in \bar{\Psi}$ are i.i.d. standard Gaussian random variables. The decoupling property asserts that, once the EOS (19) is solved, the asymptotic behavior of $\ell_1$-LinR can be statistically described by a pair of simple scalar soft-thresholding estimators (see Figs. 1(a) and 1(b)).

In the high-dimensional setting where $N$ is allowed to grow as a function of $M$, one important question is that what is the minimum number of samples $M$ required to achieve model selection consistency as $N \to \infty$. Though we obtain a pair of scalar estimators (20) and (21), there are no analytical solutions to EOS (19), making it difficult to derive an explicit condition. To overcome this difficulty, as shown in Appendix D, we perform a perturbation analysis of EOS (17) and obtain an asymptotic relation $H \simeq F$, Then, we obtain that for a RR graph $G \in \mathcal{G}_{N,d,K_0}$, given $M$ i.i.d. samples $\mathcal{D}^M$, the $\ell_1$-LinR estimator (4) can consistently recover the graph structure $G$ as $N \to \infty$ if

$$
M > \frac{c\,(\lambda, K_0)\log N}{\lambda^2}, 0 < \lambda < \tanh\,(K_0)\,, \quad (22)
$$

where $c(\lambda, K_0)$ is a constant value dependent on the regularization parameter $\lambda$ and coupling strength $K_0$ and a sharp prediction (as verified in Sec. 5) is obtained as

$$
c\,(\lambda, K_0) = \frac{2\left(1-\tanh^2\,(K_0)+d\lambda^2\right)}{1+(d-1)\tanh^2\,(K_0)}. \quad (23)
$$

For details of the analysis, including the counterpart of $\ell_1$-LogR, see Appendix D. Consequently, we obtain the *typical* sample complexity of $\ell_1$-LinR for Ising model selection for typical RR graphs in the paramagnetic phase. The result in (22) is derived for $\ell_1$-LinR with a fixed regularization parameter $\lambda$. Since the value of $\lambda$ is upper bounded by $\tanh\,(K_0)$ (otherwise false negatives occur as discussed in Appendix D), a lower bound of the typical sample complexity for $\ell_1$-LinR is obtained as

$$
M > \frac{2\log N}{\tanh^2\,(K_0)}. \quad (24)
$$

Interestingly, the scaling in (24) is the same as the information-theoretic worst-case result $M > \frac{c\log N}{K_0^2}$ obtained in [11] at high temperatures (i.e., small $K_0$) since $\tanh\,(K_0) = \mathcal{O}\,(K_0)$ as $K_0 \to 0$.

### 3.4 Non-asymptotic result for moderate $M, N$

In practice, it is desirable to predict the non-asymptotic performance of the $\ell_1$-LinR estimator for moderate $M, N$. However, the scalar estimator (20) for the active set (see Fig. 1(a)) fails to capture the fluctuations around the mean estimates. This is because in obtaining the energy term $\xi$ (16) of the free energy density (17), the fluctuations around the mean estimates $\left\{\bar{J}_j\right\}_{j\in\Psi}$ are averaged out by

the expectation $\mathbb{E}_{s,z}(\cdot)$. To address this problem, we replace $\mathbb{E}_{s,z}(\cdot)$ in (17) with a sample average by accounting for the finite-size effect, thus obtaining a modified estimator for the active set as follows

$$\{\hat{J}_j\}_{j \in \Psi} = \underset{J_{j,j \in \Psi}}{\arg\min} \left[ \frac{\sum_{\mu=1}^{M} \left( s_0^\mu - \sum_{j \in \Psi} s_j^\mu J_j - \sqrt{Q} z^\mu \right)^2}{2(1+\chi)M} + \lambda \sum_{j \in \Psi} |J_j| \right], \qquad (25)$$

where $s_0^\mu, s_{j,j \in \Psi}^\mu \sim P(s_0, \boldsymbol{s}_\Psi | \boldsymbol{J}^*)$, $z^\mu \sim \mathcal{N}(0,1)$, $\mu = 1...M$. The modified $d$-dimensional estimator (25) (see Fig. 1(c) for a schematic) is equivalent to the scalar one (20) (Fig. 1(a)) as $M \to \infty$ but it enables us to capture the fluctuations of $\{\hat{J}_j\}_{j \in \Psi}$ for moderate $M$. Note that due to the replacement of expectation with sample average in the free energy density (17), the EOS (19) also needs to be modified and it can be solved iteratively as sketched in Algorithm 1. The details are shown in Appendix E.1.

---

**Algorithm 1:** Method to solve EOS (19) together with (25)

---

**Input:** $M, N, \lambda, K_0, \rho(\gamma)$ and $T_{\mathrm{MC}}$
**Output:** $\chi, Q, E, R, F, \eta, K, H, \{\hat{J}_j^t\}_{j \in \Psi}$
**Initialization:** $\chi, Q, E, R, F, \eta, K, H$
1 **MC sampling**: For $t = 1...T_{MC}$, draw random samples $s_0^{\mu,t}, \{s_j^{\mu,t}\}_{j \in \Psi} \sim P(s_0, \boldsymbol{s}_\Psi | \boldsymbol{J}^*)$ and $z^{\mu,t} \sim \mathcal{N}(0,1), \mu = 1...M$
2 **repeat**
3      **for** $t = 1$ **to** $T_{\mathrm{MC}}$ **do**
4          Solve $\{\hat{J}_j^t\}_{j \in \Psi} = \underset{J_{j,j\in\Psi}}{\arg\min} \left[ \frac{\sum_{\mu=1}^{M}\left(s_0^{\mu,t} - \sum_{j \in \Psi} s_j^{\mu,t} J_j - \sqrt{Q}z^{\mu,t}\right)^2}{2(1+\chi)M} + \lambda \sum_{j \in \Psi} |J_j| \right]$
5          Compute $\triangle^t = \frac{1}{M} \sum_{\mu=1}^{M} \left( s_0^{\mu,t} - \sum_{j \in \Psi} s_j^{\mu,t} \hat{J}_j^t \right)^2$
6      Solve the EOS (19) with $\mathbb{E}_s \left( s_i - \sum_{j \in \Psi} s_j \bar{J}_j \right)^2 = \frac{1}{T_{\mathrm{MC}}} \sum_{t=1}^{T_{\mathrm{MC}}} \triangle^t$
7      Update values of $\chi, Q, E, R, F, \eta, K, H$
8 **until** *convergence*

---

Consequently, for moderate $M, N$, the non-asymptotic statistical properties of the $\ell_1$-LinR estimator can be characterized by the reduced $d$-dimensional $\ell_1$-LinR estimator (25) (Fig. 1(c)) and scalar estimator (21) (Fig. 1(b)) using MC simulations. Denote $\{\hat{J}_j^t\}, t = 1, ..., T_{\mathrm{MC}}$ as the estimates in $t$-th MC simulation, where $\{\hat{J}_j^t\}_{j \in \Psi}$ and $\{\hat{J}_j^t\}_{j \in \bar{\Psi}}$ are solutions of (25) and (21), and $T_{\mathrm{MC}}$ is the total number of MC simulations. Then, the *Precision* and *Recall* are computed as

$$Precision = \frac{1}{T_{\mathrm{MC}}} \sum_{t=1}^{T_{\mathrm{MC}}} \frac{\left\| \hat{J}_{j,j\in\Psi}^t \right\|_0}{\left\| \hat{J}_{j,j\in\Psi}^t \right\|_0 + \left\| \hat{J}_{j,j\in\bar{\Psi}}^t \right\|_0}, \quad Recall = \frac{1}{T_{\mathrm{MC}}} \sum_{t=1}^{T_{\mathrm{MC}}} \frac{\left\| \hat{J}_{j,j\in\Psi}^t \right\|_0}{d}, \qquad (26)$$

where $\|\cdot\|_0$ is the $\ell_0$-norm indicating the number of nonzero elements. In addition, the RSS can be computed as $RSS = \sum_j \left| \hat{J}_j - J_j^* \right|^2 = \frac{1}{T_{\mathrm{MC}}} \sum_{t=1}^{T_{\mathrm{MC}}} \sum_{j \in \Psi} \left| \hat{J}_j^t - K_0 \right|^2 + R$.

## 4 Discussions

It might seem surprising that, despite apparent model misspecification due to the use of quadratic loss, the $\ell_1$-LinR estimator can still correctly infer the structure with the same order of sample complexity as $\ell_1$-LogR. Also, our theoretical analysis implies that the idea of using linear regression for binary data is not as outrageous as one might imagine. Here we provide an intuitive explanation of its success with a discussion of its limitations.

On the average, from (4), the condition for the $\ell_1$-LinR estimator is given as

$$\langle s_0 s_k \rangle - \sum_{j \neq 0} \langle s_j s_k \rangle J_j = \lambda \partial |J_k|, \ k = 1, \ldots, N, \qquad (27)$$

where $\langle \cdot \rangle$ and $\partial |J_k|$ represent average w.r.t. the Boltzmann distribution (7) and the sub-gradient of $|J_k|$, respectively. In the paramagnetic phase, $\langle s_i s_j \rangle$ decays in its magnitude exponentially w.r.t. the

distance of sites $i$ and $j$. This guarantees that once the connections $J_k$ of sites in the first nearest neighbor set $\Psi$ are given so that

$$\langle s_0 s_k \rangle = \sum_{j \in \Psi} \langle s_j s_k \rangle J_j + \lambda \mathrm{sign}(J_k), \ \forall k \in \Psi \tag{28}$$

holds, the other conditions are automatically satisfied by setting all the other connections that are not from $\Psi$ to zero. For appropriate choice of $\lambda$, (28) has solutions of $\mathrm{sign}(J_k^*)J_k > 0, \forall k \in \Psi$. Namely $\forall k \in \Psi$, the estimate of $J_k$ has the same sign as the true value $J_k^*$. This implies that on average the $\ell_1$-LinR estimator can successfully recover the network structure up to the connection signs if $\lambda$ is chosen appropriately.

The key of the above argument is that $\langle s_i s_j \rangle$ decays exponentially fast w.r.t. the distance of two sites, which does not hold after the phase transition. Thus, it is conjectured that the $\ell_1$-LinR estimator will start to fail in the network recovery just at the phase transition point. However, it is worth noting that this is in fact not limited to $\ell_1$-LinR: $\ell_1$-LogR also exhibits similar behavior unless post-thresholding is used, as reported in [43].

## 5 Experimental results

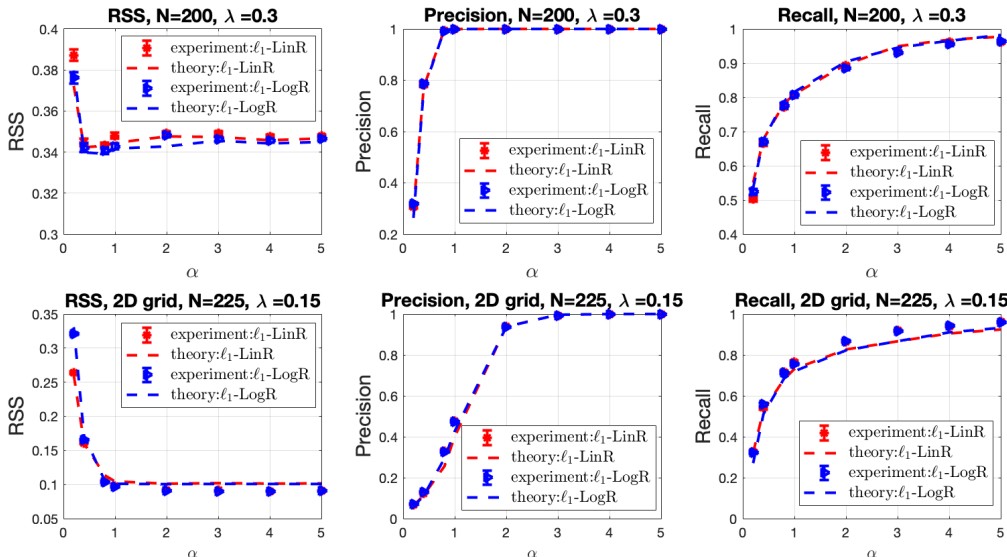

Figure 2: Theoretical and experimental results of RSS, $Precision$ and $Recall$ for RR graph and 2D grid using $\ell_1$-LinR and $\ell_1$-LogR with different values of $\alpha \equiv M/N$. The standard error bars are obtained from 1000 random runs. A good agreement between theory and experiment is achieved, even for small-size 2D grid graph with many loops. For more results, please refer to Appendix G.

In this section, we conduct numerical experiments to verify the accuracy of the theoretical analysis. The experimental procedures are as follows. First, a random graph $G \in \mathcal{G}_{N,d,K_0}$ is generated and the Ising model is defined on it. Then, the spin snapshots are obtained using the Metropolis–Hastings algorithm [44, 45, 46] in the same way as [7], yielding the dataset $\mathcal{D}^M$. We randomly choose a center spin $s_0$ and infer its neighborhood using the $\ell_1$-LinR (4) and $\ell_1$-LogR [10] estimators. To obtain standard error bars, we repeat the sequence of operations 1000 times. The RR graph $G \in \mathcal{G}_{N,d,K_0}$ with node degree $d = 3$ and coupling strength $K_0 = 0.4$ is considered, which satisfies the paramagnetic condition $(d-1)\tanh^2(K_0) < 1$. The active couplings $\{J_{ij}\}_{(i,j)\in \mathsf{E}}$ have the same probability of taking both signs of $+1$ or $-1$ [2].

We first verify the precise non-asymptotic predictions of our method described in Sec.3.4. Fig. 2 (upper figure) shows the replica and experimental results of $RSS, Precision, Recall$ for $N = 200$ with different values of $\alpha \equiv M/N$. It can be seen that for both $\ell_1$-LinR and $\ell_1$-LogR, there is a fairly

---

[2]Though this setting is different from the analysis where the nonzero couplings take a uniform sign, the result can be directly compared thanks to gauge symmetry [21].

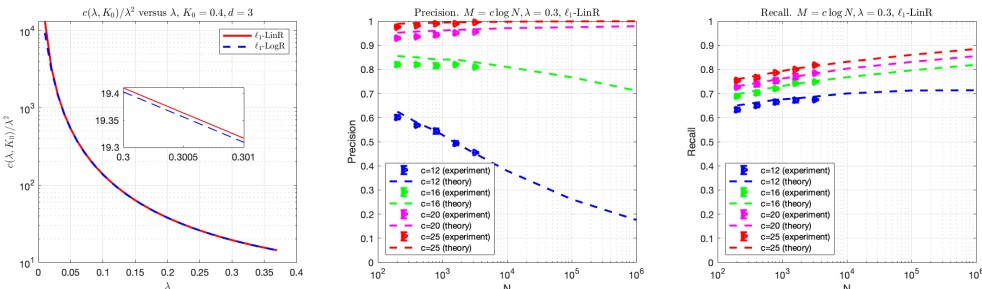

Figure 3: *Left*: critical scaling value $c_0(\lambda, K_0) \equiv \frac{c(\lambda, K_0)}{\lambda^2}$ of $\ell_1$-LinR and $\ell_1$-LogR for the RR graph $G \in \mathcal{G}_{N,d,K_0}$ with $d = 3, K_0 = 0.4$. *Middle and Right*: Precision and Recall for RR graph using $\ell_1$-LinR with $\lambda = 0.3$. Experimental results are shown for $N = 200, 400, 800, 1600, 3200$. When $c > c_0(\lambda, K_0)$ ($c_0(\lambda = 0.3, K_0) \approx 19.41$ in this case), the *Precision* increases consistently with $N$ and approaches 1 as $N \to \infty$ while it decreases consistently with $N$ when $c < c_0(\lambda, K_0)$. The *Recall* increases consistently and approach to 1 as $N \to \infty$. For more results, please refer to Appendix G.

good agreement between the theoretical predictions and experimental results, even for small $N = 200$ and small $\alpha$ (equivalently small $M$), verifying the correctness of the replica analysis. Interestingly, a quantitatively similar behavior between $\ell_1$-LinR and $\ell_1$-LogR is observed in terms of precision and recall. Regarding RSS, the two estimators actually behave differently, which can be clearly seen in Fig. 7 in Appendix G: the RSS is much smaller for $\ell_1$-LogR, which is natural since the estimates of $\ell_1$-LogR are closer to the true ones due to the model match. As our theoretical analysis is based on the typical tree-like structure assumption, it is interesting to see if it is applicable to graphs with loops. To this end, we consider the 2D 4-nearest neighbor grid with periodic boundary condition, which is one common loopy graph. Fig. 2 (lower figure) shows the results for a $15 \times 15$ 2D grid with uniform constant coupling $K_0 = 0.2$. The agreement between the theoretical and numerical results is fairly good, indicating that our theoretical result can be a good approximation even for loopy graphs. More results for different values of $N$ and $\lambda$ are shown in Fig. 7 and Fig. 8 in Appendix G.

Subsequently, the asymptotic result and sharpness of the critical scaling value $c_0(\lambda, K_0) \equiv \frac{c(\lambda, K_0)}{\lambda^2}$ in (22) are evaluated. First, Fig. 3 (left) shows comparison of $c_0(\lambda, K_0)$ between $\ell_1$-LinR and $\ell_1$-LogR for the RR graph $G \in \mathcal{G}_{N,d,K_0}$ when $d = 3, K_0 = 0.4$, indicating similar behavior of $\ell_1$-LogR and $\ell_1$-LinR. Then, we conduct experiments for $M = c \log N$ with different values of $c$ around $c_0(\lambda, K_0)$, and investigate the trend of $Precision$ and $Recall$ as $N$ increases. When $\lambda = 0.3$, Fig. 3 (middle and right) show the results of *Precision* and *Recall*, respectively. As expected, the *Precision* increases consistently with $N$ when $c > c_0(\lambda, K_0)$ and decreases consistently with $N$ when $c < c_0(\lambda, K_0)$ while the *Recall* increases consistently and approaches to 1 as $N \to \infty$, which verifies the sharpness of the critical scaling value prediction. The results for $\ell_1$-LogR, including the case of $\lambda = 0.1$ for both $\ell_1$-LinR and $\ell_1$-LogR, are shown in Fig. 9 and Fig. 10 in Appendix G.

## 6   Conclusion

In this paper, we provide a unified statistical mechanics framework for the analysis of *typical* learning performances of $\ell_1$-regularized $M$-estimators, $\ell_1$-LinR in particular, for Ising model selection on typical paramagnetic RR graphs. Using the powerful replica method, the high-dimensional $\ell_1$-regularized M-estimator is decoupled into a pair of scalar estimators, by which we obtain an accurate estimate of the typical sample complexity. It is revealed that, perhaps surprisingly, the misspecified $\ell_1$-LinR estimator is model selection consistent using $M = \mathcal{O}(\log N)$ samples, which is of the same order as $\ell_1$-LogR. Moreover, with a slight modification of the scalar estimator for the active set to account for the finite-size effect, we further obtain sharp predictions of the non-asymptotic behavior of $\ell_1$-LinR (also $\ell_1$-LogR) for moderate $M, N$. There is an excellent agreement between theoretical predictions and experimental results, even for graphs with many loops, which supports our findings. Several key assumptions are made in our theoretical analysis, such as the paramagnetic assumption which implies that the coupling strength should not be too large. It is worth noting that the restrictive paramagnetic assumption is not only limited to $\ell_1$-LinR, but also to other low-complexity estimators like $\ell_1$-LogR unless post-thresholding is used [43]. These assumptions restrict the applicability of the presented result, and thus overcoming such limitations will be an important direction for future work.

## Acknowledgements

This work was supported by JSPS KAKENHI Nos. 17H00764, 18K11463, and 19H01812, and JST CREST Grant Number JPMJCR1912, Japan.

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
