# A   Free energy density $f$ computation

The detailed derivation of the average free energy density $f = -\frac{1}{N\beta}\left[\log Z\right]_{\mathcal{D}^M}$ in (9) using the replica method is illustrated. Our method provides a unified framework for the statistical mechanics analysis of any $\ell_1$-regularized $M$-estimator of the form (2). As a result, for generality, in the following derivations, we first focus on a generic $\ell_1$-regularized $M$-estimator (2) with a generic loss function $\ell(\cdot)$. After obtaining the generic results, specific results for both the $\ell_1$-LinR estimator (4) with square loss $\ell(x) = \frac{1}{2}(x-1)^2$ and the $\ell_1$-LogR estimator with logistic loss $\ell(x) = \log\left(1 + e^{-2x}\right)$ are provided. For the IS estimator, the results can be easily obtained by substituting $\ell(x) = e^{-x}$, though the specific results are not shown.

## A.1   Energy term $\xi$ of $f$

The key of replica method is to compute the replicated partition function $[Z^n]_{\mathcal{D}^M}$. According to the definition in (10) and Ansatz (A1) in Sec. 3.2, the average replicated partition function $[Z^n]_{\mathcal{D}^M}$ can be re-written as

$$
\begin{aligned}
[Z^n]_{\mathcal{D}^M} &= \int \prod_{a=1}^{n} d\boldsymbol{J}^a e^{-\beta\lambda M \sum_{a=1}^{n} \sum_j |J_j^a|} \left\{ \sum_s P_{\text{Ising}}(s|\boldsymbol{J}^*) \exp\left[-\beta \sum_{a=1}^{n} \ell(s_0 h^a)\right] \right\}^M, \\
&\approx \int \prod_{a=1}^{n} dw^a e^{-\beta\lambda M \left(\sum_{a=1}^{n} \sum_{j\in\Psi} |\bar{J}_j| + \sum_{a=1}^{n} \frac{1}{\sqrt{N}} \|w^a\|_1\right)} \times \\
&\quad \left\{ \sum_s P_{\text{Ising}}(s|\boldsymbol{J}^*) \prod_a \int dh_w^a \delta\left(h_w^a - \frac{1}{\sqrt{N}} \sum_{j\in\bar{\Psi}} w_j^a s_j\right) e^{-\beta \sum_{a=1}^{n} \ell\left(s_0\left(\sum_{j\in\Psi} \bar{J}_j s_j + h_w^a\right)\right)} \right\}^{\alpha N} \\
&= \int \prod_{a=1}^{n} dw^a e^{-\beta\lambda M \left(n \sum_{j\in\Psi} |\bar{J}_j| + \sum_{a=1}^{n} \frac{\|w^a\|_1}{\sqrt{N}}\right)} \times \\
&\quad \left\{ \sum_{s_0, s_\Psi} \int \prod_{a=1}^{n} dh_w^a P\left(s_0, s_\Psi, \{h_w^a\}_a \mid \boldsymbol{J}^*, \{w^a\}_a\right) e^{-\beta \sum_{a=1}^{n} \ell\left(s_0\left(\sum_{j\in\Psi} \bar{J}_j s_j + h_w^a\right)\right)} \right\}^{\alpha N} \\
&\approx \int \prod_{a=1}^{n} dw^a e^{-\beta\lambda M \left(n \sum_{j\in\Psi} |\bar{J}_j| + \sum_{a=1}^{n} \frac{\|w^a\|_1}{\sqrt{N}}\right)} \times \\
&\quad \left\{ \sum_{s_0, \boldsymbol{s}_\Psi} P(s_0, \boldsymbol{s}_\Psi | \boldsymbol{J}^*) \int \prod_{a=1}^{n} dh_w^a P_{\text{noise}}\left(\{h_w^a\}_a \mid \{w^a\}_a\right) e^{-\beta \sum_{a=1}^{n} \ell\left(s_0\left(\sum_{j\in\Psi} \bar{J}_j s_j + h_w^a\right)\right)} \right\}^{\alpha N},
\end{aligned}
\tag{29}
$$

where $\left\{\frac{1}{\sqrt{N}} w_j^a, j \in \Psi\right\}$ in the finite active set $\Psi$ are neglected in the second line when $N$ is large, $P(s_0, \boldsymbol{s}_\Psi | \boldsymbol{J}^*) = \sum_{\boldsymbol{s}_{\bar{\Psi}}} P_{\text{Ising}}(s|\boldsymbol{J}^*)$ is the marginal distribution of $s_0, \boldsymbol{s}_\Psi$ that can be computed as [7], $P_{\text{noise}}\left(\{h_w^a\}_a \mid \{w^a\}_a\right)$ is the distribution of the "noise" part $h_w^a \equiv \frac{1}{\sqrt{N}} \sum_{j\in\bar{\Psi}} w_j^a s_j$ of the local field. In the last line, the asymptotic independence between $h_w^a$ and $(s_0, \boldsymbol{s}_\Psi)$ are applied as discussed in [7]. Regarding the marginal distribution $P(s_0, \boldsymbol{s}_\Psi | \boldsymbol{J}^*)$, in general we have to take into account the cavity fields in the marginal distribution. In the case considered in this paper, however, the paramagnetic assumption simplifies the marginal distribution and finally it is proportional to $e^{s_0 \sum_{j\in\Psi} J_j^* s_j}$ [7]. When $\Psi$ has a small cardinality $d$, we can compute the expectation w.r.t. $(s_0, s_\Psi)$ exactly by exhaustive enumeration. For large $d$, MC methods like the Metropolis–Hastings algorithm [44, 45, 46] might be used.

To proceed with the calculation, according to the CLT, the noise part $\{h_w^a\}_{a=1}^{n}$ can be regarded as Gaussian variables so that $P_{\text{noise}}\left(\{h_w^a\}_a \mid \{w^a\}_a\right)$ can be approximated as a multivariate Gaussian

distribution. Under the RS ansatz, two auxiliary order parameters are introduced, i.e.,

$$Q \equiv \frac{1}{N} \sum_{i,j \in \bar{\Psi}} w_i^a C_{ij}^{\backslash 0} w_j^a, \tag{30}$$

$$q \equiv \frac{1}{N} \sum_{i,j \in \bar{\Psi}} w_i^a C_{ij}^{\backslash 0} w_j^b, \ (a \neq b), \tag{31}$$

where $C^{\backslash 0} = \left\{ C_{ij}^{\backslash 0} \right\}$ is the covariance matrix of the original Ising model without $s_0$. To write the integration in terms of the order parameters $Q, q$, we introduce the following trivial identities

$$1 = N \int dQ \delta \left( \sum_{i,j \neq 0} w_i^a C_{ij}^{\backslash 0} w_j^a - NQ \right), a = 1, ..., n \tag{32}$$

$$1 = N \int dq \delta \left( \sum_{i,j \neq 0} w_i^a C_{ij}^{\backslash 0} w_j^b - Nq \right), a < b, \tag{33}$$

so that $[Z^n]_{\mathcal{DM}}$ in (29) can be rewritten as

$$[Z^n]_{\mathcal{DM}} = e^{-\beta \lambda M n \sum_{j \in \Psi} |\bar{J}_j|} \int dQ dq \int \prod_{a=1}^n dw^a e^{-\lambda \beta \frac{M}{\sqrt{N}} \sum_{a=1}^n \|w^a\|_1}$$

$$\times \prod_{a=1}^n \delta \left( \sum_{i,j} w_i^a C_{ij}^{\backslash 0} w_j^a - NQ \right) \prod_{a<b} \delta \left( \sum_{i,j} w_i^a C_{ij}^{\backslash 0} w_j^b - Nq \right) \times$$

$$\left\{ \sum_{s_0, \boldsymbol{s}_\Psi} P\left(s_0, \boldsymbol{s}_\Psi | \boldsymbol{J}^*\right) \int \prod_{a=1}^n dh_w^a P_{\text{noise}} \left( \{h_w^a\}_a | \{w^a\}_a \right) e^{-\beta \sum_{a=1}^n \ell \left( s_0 \left( \sum_{j \in \Psi} \bar{J}_j s_j + h_w^a \right) \right)} \right\}^{\alpha N} \tag{34}$$

$$= \int dQ dq I e^{M \log L}, \tag{35}$$

where

$$I \equiv \int \prod_{a=1}^n dw^a e^{-\lambda \beta \frac{M}{\sqrt{N}} \sum_{a=1}^n \|w^a\|_1} \prod_{a=1}^n \delta \left( \sum_{i,j} w_i^a C_{ij}^{\backslash 0} w_j^a - NQ \right) \prod_{a<b} \delta \left( \sum_{i,j} w_i^a C_{ij}^{\backslash 0} w_j^b - Nq \right), \tag{36}$$

$$L \equiv e^{-\beta \lambda n \sum_{j \in \Psi} |\bar{J}_j|} \sum_{s_0, \boldsymbol{s}_\Psi} P\left(s_0, \boldsymbol{s}_\Psi | \boldsymbol{J}^*\right) \int \prod_{a=1}^n dh_w^a P_{\text{noise}} \left( \{h_w^a\}_a | \{w^a\}_a \right) e^{-\beta \sum_{a=1}^n \ell \left( s_0 \left( \sum_{j \in \Psi} \bar{J}_j s_j + h_w^a \right) \right)}. \tag{37}$$

According to CLT and (30) and (31), the noise parts $h_w^a, a = 1, \ldots, n$ follow a multivariate Gaussian distribution with zero mean (paramagnetic assumption) and covariances

$$\left\langle h_w^a h_w^b \right\rangle^{\backslash 0} = Q \delta_{ab} + (1 - \delta_{ab}) q. \tag{38}$$

Consequently, by introducing two auxiliary i.i.d. standard Gaussian random variables $v_a \sim \mathcal{N}(0,1), z \sim \mathcal{N}(0,1)$, the noise parts $h_w^a, a = 1, \ldots, n$ can be written in a compact form

$$h_w^a = \sqrt{Q - q} v_a + \sqrt{q} z, a = 1, \ldots, n \tag{39}$$

so that $L$ in (37) could be written as

$$
L = e^{-\beta\lambda n \sum_{j\in\Psi}|\bar{J}_j|} \sum_{s_0,\boldsymbol{s}_\Psi} P\left(s_0,\boldsymbol{s}_\Psi|\boldsymbol{J}^*\right) \int \prod_{a=1}^{n} dh_w^a P_{\text{noise}}\left(\{h_w^a\}_a \,|\, \{w^a\}_a\right) e^{-\beta \sum_{a=1}^{n} \ell\left(s_0\left(\sum_{j\in\Psi}\bar{J}_j s_j + h_w^a\right)\right)}
$$

$$
= e^{-\beta\lambda n \sum_{j\in\Psi}|\bar{J}_j|} \sum_{s_0,\boldsymbol{s}_\Psi} P\left(s_0,\boldsymbol{s}_\Psi|\boldsymbol{J}^*\right) \int \mathcal{D}z \prod_a \mathcal{D}v_a\, e^{-\beta \sum_{a=1}^{n} \ell\left(s_0\left(\sum_{j\in\Psi}\bar{J}_j s_j + \sqrt{Q-q}v_a + \sqrt{q}z\right)\right)}
$$

$$
= e^{-\beta\lambda n \sum_{j\in\Psi}|\bar{J}_j|} \sum_{s_0,\boldsymbol{s}_\Psi} P\left(s_0,\boldsymbol{s}_\Psi|\boldsymbol{J}^*\right) \int \mathcal{D}z \left[\underbrace{\int \mathcal{D}v\, e^{-\beta\ell\left(s_0\left(\sum_{j\in\Psi}\bar{J}_j s_j + \sqrt{Q-q}v + \sqrt{q}z\right)\right)}}_{A}\right]^n
$$

$$
= e^{-\beta\lambda n \sum_{j\in\Psi}|\bar{J}_j|} \sum_{s_0,\boldsymbol{s}_\Psi} P\left(s_0,\boldsymbol{s}_\Psi|\boldsymbol{J}^*\right) \mathbb{E}_z\left(A^n\right), \tag{40}
$$

where $\mathcal{D}z = \frac{dz}{\sqrt{2\pi}}e^{-\frac{z^2}{2}}$. As a result, using the replica formula, we have

$$
\lim_{n\to 0}\frac{1}{n}\log L
$$

$$
= -\beta\lambda \sum_{j\in\Psi}|\bar{J}_j| + \lim_{n\to 0}\frac{\log \sum_{s_0,\boldsymbol{s}_\Psi} P\left(s_0,\boldsymbol{s}_\Psi|\boldsymbol{J}^*\right) E_z\left(A^n\right)}{n}
$$

$$
= -\beta\lambda \sum_{j\in\Psi}|\bar{J}_j| + \mathbb{E}_z\left[\sum_{s_0,\boldsymbol{s}_\Psi} P\left(s_0,\boldsymbol{s}_\Psi|\boldsymbol{J}^*\right)\log A\right]
$$

$$
= -\beta\lambda \sum_{j\in\Psi}|\bar{J}_j| + \sum_{s_0,\boldsymbol{s}_\Psi} P\left(s_0,\boldsymbol{s}_\Psi|\boldsymbol{J}^*\right) \int \mathcal{D}z \log \int \mathcal{D}v\, e^{-\beta\ell\left(s_0\left(\sum_{j\in\Psi}\bar{J}_j s_j + \sqrt{Q-q}v + \sqrt{q}z\right)\right)}
$$

$$
= -\beta\lambda \sum_{j\in\Psi}|\bar{J}_j| + \sum_{s_0,\boldsymbol{s}_\Psi} P\left(s_0,\boldsymbol{s}_\Psi|\boldsymbol{J}^*\right) \int \mathcal{D}z \log \int \frac{dy}{\sqrt{2\pi(Q-q)}} e^{-\frac{\left[y-s_0\left(\sum_{j\in\Psi}\bar{J}_j s_j + \sqrt{q}z\right)\right]^2}{2(Q-q)}} e^{-\beta\ell(y)}, \tag{41}
$$

where in the last line, a change of variable $y = s_0\left(\sum_{j\in\Psi}\bar{J}_j s_j + \sqrt{Q-q}v + \sqrt{q}z\right)$ is used.

As a result, from (9), the average free energy density in the limit $\beta\to\infty$ reads

$$
f(\beta\to\infty) = \lim_{\beta\to\infty} -\frac{1}{N\beta}\left\{\lim_{n\to 0}\frac{\partial}{\partial n}\log I + M\lim_{n\to 0}\frac{\partial}{\partial n}\log L\right\}
$$

$$
= -\texttt{Extr}\left\{-\xi + S\right\}, \tag{42}
$$

where $\texttt{Extr}\{\cdot\}$ denotes extremization w.r.t. some relevant variables, and $\xi, S$ are the corresponding energy and entropy terms of $f$, respectively:

$$
S = \lim_{n\to 0}\frac{1}{N\beta}\frac{\partial}{\partial n}\log I, \tag{43}
$$

$$
I = \int \prod_{a=1}^{n} dw^a e^{-\lambda\beta \sum_{a=1}^{n}\|w^a\|_1} \prod_{a=1}^{n}\delta\left(\sum_{i,j}w_i^a C_{ij}w_j^a - NQ\right) \prod_{a<b}\delta\left(\sum_{i,j}w_i^a C_{ij}w_j^b - Nq\right), \tag{44}
$$

$$
\xi = \alpha\lambda \sum_{j\in\Psi}|\bar{J}_j| + \alpha\mathbb{E}_{s,z}\left(\min_y\left[\frac{\left(y-s_0\left(\sqrt{Q}z + \sum_{j\in\Psi}\bar{J}_j s_j\right)\right)^2}{2\chi} + \ell(y)\right]\right), \tag{45}
$$

and the relation $\lim_{\beta\to\infty}\beta(Q-q)\equiv\chi$ is used [6, 7]. The extremization in the free energy result (42) comes from saddle point method in the large $N$ limit.

## A.2 Entropy term $S$ of $f$

To obtain the final result of free energy density, there is still one remaining entropy term $S$ to compute, which requires the result of $I$ (44). However, unlike the $\ell_2$-norm, the $\ell_1$-norm in (44) breaks the rotational invariance property, which makes the computation of $I$ difficult and the methods in [7, 29] are no loner applicable. To address this problem, applying the Haar Orthogonal Ansatz (A2) in Sec. 3.2, we employ a method to replace $I$ with an average $[I]_O$ over the orthogonal matrix $O$ generated from the Haar orthogonal measure.

Specifically, also under the RS ansatz, two auxiliary order parameters are introduced, i.e.,

$$R \equiv \frac{1}{N} \sum_{i,j} w_i^a w_j^a, \tag{46}$$

$$r \equiv \frac{1}{N} \sum_{i,j} w_i^a w_j^b, \ (a \neq b). \tag{47}$$

Then, by inserting the delta functions $\prod_a \delta\left((w^a)^T w^a - NR\right) \prod_{a<b} \delta\left((w^a)^T w^b - Nr\right)$, we obtain

$$I = \int \prod_{a=1}^n dw^a e^{-\frac{\lambda \beta M}{\sqrt{N}} \sum_{a=1}^n \|w^a\|_1} \prod_{a=1}^n \delta\left((w^a)^T Cw^a - NQ\right) \prod_{a<b} \delta\left((w^a)^T Cw^b - Nq\right)$$
$$\times \int dRdr \prod_a \delta\left((w^a)^T w^a - NR\right) \prod_{a<b} \delta\left((w^a)^T w^b - Nr\right). \tag{48}$$

Moreover, replacing the original delta functions in (48) as the following identities

$$\begin{cases} \delta\left((w^a)^T Cw^a - NQ\right) = \int d\hat{Q} e^{-\frac{\hat{Q}}{2}\left((w^a)^T Cw^a - NQ\right)}, \\ \delta\left((w^a)^T Cw^b - Nq\right) = \int d\hat{q} e^{\hat{q}\left((w^a)^T Cw^b - Nq\right)}, \end{cases}$$

and taking average over the orthogonal matrix $O$, after some algebra, the $I$ is replaced with the following average $[I]_O$

$$[I]_O = \int dRdrd\hat{Q}d\hat{q} \prod_{a=1}^n dw^a e^{-\frac{\lambda \beta M}{\sqrt{N}} \sum_{a=1}^n \|w^a\|_1} \prod_a \delta\left((w^a)^T w^a - NR\right) \prod_{a<b} \delta\left((w^a)^T w^b - Nr\right)$$
$$\times \exp\left\{\frac{Nn}{2}\hat{Q}Q - \frac{Nn}{2}(n-1)\hat{q}q\right\} \times \left[e^{\frac{1}{2}\text{Tr}(CL_n)}\right]_O, \tag{49}$$

$$L_n = -\left(\hat{Q} + \hat{q}\right) \sum_{a=1}^n w^a (w^a)^T + \hat{q} \left(\sum_{a=1}^n w^a\right) \left(\sum_{b=1}^n w^b\right)^T. \tag{50}$$

To proceed with the computation, the eigendecompostion of the matrix $L_n$ is performed. After some algebra, for the configuration of $w^a$ that satisfies both $(w^a)^T w^a = NR$ and $(w^a)^T w^b = Nr$, the eigenvalues and associated eigenvectors of matrix $L_n$ can be calculated as follows

$$\begin{cases} \lambda_1 = -N\left(\hat{Q} + \hat{q} - n\hat{q}\right)(R - r + nr), \\ u_1 = \sum_{a=1}^n w^a, \\ \lambda_2 = -N\left(\hat{Q} + \hat{q}\right)(R - r), \\ u_a = w^a - \frac{1}{n}\sum_{b=1}^n w^b, a = 2, ..., n, \end{cases} \tag{51}$$

where $\lambda_1$ is the eigenvalue corresponding to the eigenvector $u_1$ while $\lambda_2$ is the degenerate eigenvalue corresponding to eigenvectors $u_a, a = 2, ..., n$. To compute $\left[e^{\frac{1}{2}\text{Tr}(CL_n)}\right]_O$, we define a function $G(x)$ as

$$G(x) \equiv \frac{1}{N} \log\left[\exp\left(\frac{x}{2}\text{Tr}C\left(\mathbf{1}\mathbf{1}^T\right)\right)\right]_O$$
$$= \underset{\Lambda}{\text{Extr}}\left\{-\frac{1}{2}\int \log(\Lambda - \gamma)\rho(\gamma)d\gamma + \frac{\Lambda}{2}x\right\} - \frac{1}{2}\log x - \frac{1}{2}, \tag{52}$$

and $\rho(\gamma)$ is the eigenvalue distribution (EVD) of $C$. Then, combined with (51), after some algebra, we obtain that

$$\frac{1}{N} \log \left[ e^{\frac{1}{2} \text{Tr}(CL_n)} \right]_O = G\left( -\left( \hat{Q} + \hat{q} - n\hat{q} \right)(R - r + nr) \right) + (n-1) G\left( -\left( \hat{Q} + \hat{q} \right)(R - r) \right). \tag{53}$$

Furthermore, replacing the original delta functions in (48) as

$$\begin{cases} \delta\left( (w^a)^T w^a - NR \right) = \int d\hat{R} e^{-\frac{\hat{R}}{2}\left( (w^a)^T w^a - NR \right)}, \\ \delta\left( (w^a)^T w^b - Nr \right) = \int d\hat{r} e^{\hat{r}\left( (w^a)^T w^b - Nr \right)}, \end{cases}$$

we obtain

$$[I]_0 = \int dR dr d\hat{Q} d\hat{q} d\hat{R} d\hat{r} \prod_{a=1}^{n} dw^a \exp\left\{ -\sum_{a=1}^{n} \frac{\lambda \beta M}{\sqrt{N}} \|w^a\|_1 - \frac{\hat{R} + \hat{r}}{2} \sum_{a=1}^{n} (w^a)^T w^a + \frac{\hat{r}}{2} \sum_{a,b} (w^a)^T w^b \right\}$$

$$\times \exp\left\{ \frac{Nn}{2} \hat{R}R - \frac{Nn}{2}(n-1)\hat{r}r + \frac{Nn}{2}\hat{Q}Q - \frac{Nn}{2}(n-1)\hat{q}q \right\} \times \left[ e^{\frac{1}{2}\text{Tr}(CL_n)} \right]_O. \tag{54}$$

In addition, using a Gaussian integral, the following result can be linearized as

$$\int \prod_{a=1}^{n} dw^a \exp\left\{ -\sum_{a=1}^{n} \frac{\lambda \beta M}{\sqrt{N}} \|w^a\|_1 - \frac{\hat{R} + \hat{r}}{2} \sum_{a=1}^{n} (w^a)^T w^a + \frac{\hat{r}}{2} \sum_{a,b} (w^a)^T w^b \right\}$$

$$= \int \prod_{a=1}^{n} dw^a \exp\left\{ -\sum_{a=1}^{n}\sum_{i=1}^{N} \frac{\lambda \beta M}{\sqrt{N}} |w_i^a| - \frac{\hat{R} + \hat{r}}{2} \sum_{a=1}^{n}\sum_{i=1}^{N} (w_i^a)^2 + \frac{\hat{r}}{2} \sum_{i=1}^{N} \left( \sum_{a=1}^{n} w_i^a \right)^2 \right\}$$

$$= \prod_{i} \int \mathcal{D}z_i \int \prod_{a=1}^{n} dw^a \exp\left\{ -\sum_{a=1}^{n} \frac{\lambda \beta M}{\sqrt{N}} |w_i^a| - \frac{\hat{R} + \hat{r}}{2} \sum_{a=1}^{n} (w_i^a)^2 + \sqrt{\hat{r}} z_i \sum_{a} w_i^a \right\}$$

$$= \prod_{i} \int \mathcal{D}z_i \left\{ \int dw \exp\left[ -\frac{\hat{R} + \hat{r}}{2} w_i^2 + \left( \sqrt{\hat{r}} z - \frac{\lambda \beta M}{\sqrt{N}} \text{sign}(w_i) \right) w_i \right] \right\}^n,$$

where $\mathcal{D}z_i = \frac{dz_i}{\sqrt{2\pi}} e^{-\frac{z_i^2}{2}}$. Consequently, the entropy term $S$ of the free energy density $f$ is computed as

$$\lim_{n \to 0} \frac{1}{N} \frac{\partial}{\partial n} \log [I]_O = \left( \hat{q}(R - r) - \left( \hat{Q} + \hat{q} \right) r \right) G'\left( -\left( \hat{Q} + \hat{q} \right)(R - r) \right)$$

$$+ G\left( -\left( \hat{Q} + \hat{q} \right)(R - r) \right) + \frac{\hat{R}R}{2} + \frac{\hat{r}r}{2} + \frac{\hat{Q}Q}{2} + \frac{\hat{q}q}{2}$$

$$+ \int Dz \log \int dw \exp\left[ -\frac{\hat{R} + \hat{r}}{2} w^2 + \left( \sqrt{\hat{r}} z - \frac{\lambda \beta M}{\sqrt{N}} \text{sign}(w) \right) w \right].$$

For $\beta \to \infty$, according to the characteristic of the Boltzmann distribution, the following scaling relations are assumed to hold, i.e.,

$$\begin{cases} \hat{Q} + \hat{q} & \equiv \beta E \\ \hat{q} & \equiv \beta^2 F \\ \hat{R} + \hat{r} & \equiv \beta K \\ \hat{r} & \equiv \beta^2 H \\ \beta(Q - q) & \equiv \chi \\ \beta(R - r) & \equiv \eta \end{cases} \tag{55}$$

Finally, the entropy term is computed as

$$S = (-ER + F\eta) G'(-E\eta) + \frac{1}{2}EQ - \frac{1}{2}F\chi + \frac{1}{2}KR - \frac{1}{2}H\eta- \tag{56}$$

$$\int \min_{w} \left\{ \frac{K}{2} w^2 - \left( \sqrt{H} z - \frac{\lambda M}{\sqrt{N}} \text{sign}(w) \right) w \right\} Dz. \tag{57}$$

## A.3 Free energy density result

Combining the results (45) and (57) together, the free energy density for general loss function $\ell(\cdot)$ in the limit $\beta \to \infty$ is obtained as

$$f(\beta \to \infty) = -\operatorname*{Extr}_{\Theta} \left\{ \begin{array}{c} -\alpha \mathbb{E}_{s,z} \left( \min_{y} \left[ \frac{\left(y - s_0\left(\sqrt{Q}z + \sum_{j \in \Psi} \bar{J}_j s_j\right)\right)^2}{2\chi} + \ell(y) \right] \right) - \alpha\lambda \sum_{j \in \Psi} |\bar{J}_j| \\ + (-ER + F\eta) G'(-E\eta) + \frac{1}{2}EQ - \frac{1}{2}F\chi \\ + \frac{1}{2}KR - \frac{1}{2}H\eta - \mathbb{E}_z \left( \min_{w} \left\{ \frac{K}{2}w^2 - \left(\sqrt{H}z - \frac{\lambda M}{\sqrt{N}}\operatorname{sign}(w)\right)w \right\} \right) \end{array} \right\},$$

$$(58)$$

where the values of the parameters $\Theta = \left\{ \chi, Q, E, R, F, \eta, K, H, \{\bar{J}_j\}_{j \in \Psi} \right\}$ can be calculated by the extremization condition, i.e., solving the equations of state (EOS). For general loss function $\ell(y)$, the EOS for (58) is as follows

$$\begin{cases} \hat{y}(s, z, \chi, Q, J) = \arg\max_{y} \left\{ -\frac{\left(y - s_0\left(\sqrt{Q}z + \sum_{j \in \Psi} \bar{J}_j s_j\right)\right)^2}{2\chi} - \ell(y) \right\} \\ E = \frac{\alpha}{\sqrt{Q}} \mathbb{E}_{s,z} \left( s_0 z \frac{d\ell(y)}{dy} \big|_{y = \hat{y}(s, z, \chi, Q, J)} \right) \\ F = \alpha \mathbb{E}_{s,z} \left( \left( \frac{d\ell(y)}{dy} \big|_{y = \hat{y}(s, z, \chi, Q, J)} \right)^2 \right) \\ R = \frac{1}{K^2} \left[ \left( H + \frac{\lambda^2 M^2}{N} \right) \operatorname{erfc}\left( \frac{\lambda M}{\sqrt{2HN}} \right) - 2\lambda M \sqrt{\frac{H}{N}} \frac{1}{\sqrt{2\pi}} e^{-\frac{\lambda^2 M^2}{2HN}} \right] \\ E\eta = -\int \frac{\rho(\gamma)}{\tilde{\Lambda} - \gamma} d\gamma \\ Q = \frac{F}{E^2} + R\tilde{\Lambda} - (-ER + F\eta)\eta \frac{1}{\int \frac{\rho(\lambda)}{(\tilde{\Lambda} - \lambda)^2} d\lambda} \\ K = E\tilde{\Lambda} + \frac{1}{\eta} \\ \chi = \frac{1}{E} + \eta\tilde{\Lambda} \\ H = \frac{R}{\eta^2} + F\tilde{\Lambda} + (-ER + F\eta) E \frac{1}{\int \frac{\rho(\lambda)}{(\tilde{\Lambda} - \lambda)^2} d\lambda} \\ \eta = \frac{1}{K} \operatorname{erfc}\left( \frac{\lambda M}{\sqrt{2HN}} \right) \\ \bar{J}_{j, j \in \Psi} = \arg\min_{J_{j, j \in \Psi}} \left\{ \mathbb{E}_{s,z} \left( \left[ \frac{\left(\hat{y}(s, z, \chi, Q, J) - s_0\left(\sqrt{Q}z + \sum_{j \in \Psi} \bar{J}_j s_j\right)\right)^2}{2\chi} + \ell(\hat{y}(s, z, \chi, Q, J)) \right] \right) + \lambda \sum_{j \in \Psi} |J_j| \right\} \end{cases}$$

$$(59)$$

where $\tilde{\Lambda}$ satisfying $E\eta = -\int \frac{\rho(\gamma)}{\tilde{\Lambda} - \gamma} d\gamma$ is determined by the extremization condition in (52) combined with the free energy result (58). In general, there are no analytic solutions for the EOS (59) but it can be solved numerically.

### A.3.1 quadratic loss $\ell(y) = (y-1)^2/2$

In the case of square lass $\ell(y) = (y-1)^2/2$ for the $\ell_1$-LinR estimator, there is an analytic solution to $y$ in $\min_{y} \left[ \frac{\left(y - s_0\left(\sqrt{Q}z + \sum_{j \in \Psi} \bar{J}_j s_j\right)\right)^2}{2\chi} + \ell(y) \right]$ and thus the results can be further simplified. Specifically, the free energy can be written as follows

$$f(\beta \to \infty) = -\operatorname*{Extr}_{\Theta} \left\{ \begin{array}{c} -\frac{\alpha}{2(1+\chi)} \mathbb{E}_{s,z} \left[ \left( s_0 - \sum_{j \in \Psi} s_j \bar{J}_j - \sqrt{Q}z \right)^2 \right] - \alpha\lambda \sum_{j \in \Psi} |\bar{J}_j| \\ + (-ER + F\eta) G'(-E\eta) + \frac{1}{2}EQ - \frac{1}{2}F\chi \\ + \frac{1}{2}KR - \frac{1}{2}H\eta - \mathbb{E}_z \left[ \min_{w} \left\{ \frac{K}{2}w^2 - \left(\sqrt{H}z - \frac{\lambda M}{\sqrt{N}}\operatorname{sign}(w)\right)w \right\} \right] \end{array} \right\},$$

$$(60)$$

and the corresponding EOS can be written as

$$
\begin{cases}
E = \frac{\alpha}{(1+\chi)}, & (a) \\[4pt]
F = \frac{\alpha}{(1+\chi)^2}\left[\mathbb{E}_s\left(s_i - \sum_{j\in\Psi} s_j \bar{J}_j\right)^2 + Q\right], & (b) \\[8pt]
R = \frac{1}{K^2}\left[\left(H + \frac{\lambda^2 M^2}{N}\right)\mathrm{erfc}\left(\frac{\lambda M}{\sqrt{2HN}}\right) - 2\lambda M\sqrt{\frac{H}{N}}\frac{1}{\sqrt{2\pi}}e^{-\frac{\lambda^2 M^2}{2HN}}\right], & (c) \\[8pt]
E\eta = -\int \frac{\rho(\gamma)}{\tilde{A}-\gamma}d\gamma, & (d) \\[6pt]
Q = \frac{F}{E^2} + R\tilde{A} - (-ER + F\eta)\frac{\eta}{\int \frac{\rho(\gamma)}{(\tilde{A}-\gamma)^2}d\gamma}, & (e) \\[8pt]
K = E\tilde{A} + \frac{1}{\eta}, & (f) \\[4pt]
\chi = \frac{1}{E} + \eta\tilde{A}, & (g) \\[4pt]
H = \frac{R}{\eta^2} + F\tilde{A} + (-ER + F\eta)\frac{E}{\int \frac{\rho(\gamma)}{(\tilde{A}-\gamma)^2}d\gamma}, & (h) \\[8pt]
\eta = \frac{1}{K}\mathrm{erfc}\left(\frac{\lambda M}{\sqrt{2HN}}\right), & (i) \\[6pt]
\bar{J}_j = \frac{\mathtt{soft}(\tanh(K_0),\lambda(1+\chi))}{1+(d-1)\tanh^2(K_0)}, j\in\Psi, & (j)
\end{cases}
\tag{61}
$$

Note that the mean estimates $\left\{\bar{J}_j, j\in\Psi\right\}$ in (61) is obtained by solving the following reduced optimization problem

$$
\underset{\{\bar{J}_j\}}{\arg\min}\left\{\frac{1}{2(1+\chi)}\mathbb{E}_{s,z}\left[\left(s_0 - \sum_{j\in\Psi} s_j \bar{J}_j - \sqrt{Q}z\right)^2\right] - \lambda\sum_{j\in\Psi}|\bar{J}_j|\right\},
\tag{62}
$$

where the corresponding fixed-point equation associated with any $\bar{J}_k, k\in\Psi$ can be written as follows

$$
\frac{1}{1+\chi}\mathbb{E}_s\left[s_k\left(s_0 - \sum_{j\in\Psi} s_j \bar{J}_j\right)\right] - \lambda\,\mathtt{sign}\left(\bar{J}_k\right) = 0, \forall k\in\Psi,
\tag{63}
$$

where the $\mathtt{sign}(\cdot)$ denotes an element-wise application of the standard sign function. For a RR graph $G\in\mathcal{G}_{N,d,K_0}$ with degree $d$ and coupling strength $K_0$, without loss of generality, assuming that all the active couplings are positive, we have $\mathbb{E}_s\left(s_0 s_k\right) = \tanh\left(K_0\right), \forall k\in\Psi$, and $\mathbb{E}_s\left(s_k s_j\right) = \tanh^2\left(K_0\right), \forall k,j\in\Psi, k\neq j$. Given these results and thanks to the the symmetry, we obtain

$$
\bar{J}_j = \frac{\mathtt{soft}\left(\tanh\left(K_0\right), \lambda\left(1+\chi\right)\right)}{1+(d-1)\tanh^2\left(K_0\right)}, j\in\Psi,
\tag{64}
$$

where $\mathtt{soft}\left(z,\tau\right) = \mathtt{sign}\left(z\right)\left(|z|-\tau\right)_+$ is the soft-thresholding function, i.e.,

$$
\mathtt{soft}\left(z,\tau\right) \equiv \mathtt{sign}\left(z\right)\left(|z|-\tau\right)_+ \equiv
\begin{cases}
z - \tau, & z > \tau \\
0, & |z|\leq \tau \\
z + \tau, & z < -\tau
\end{cases}
\tag{65}
$$

On the other hand, in the inactive set $\bar{\Psi}$, each component of the scaled noise estimates can be statistically described as the solution to the scalar estimator $\min_w\left\{\frac{K}{2}w^2 - \left(\sqrt{H}z - \frac{\lambda M}{\sqrt{N}}\mathtt{sign}\left(w\right)\right)w\right\}$ in (58). Consequently, recalling the definition of $w$ in (11), the estimates $\left\{\hat{J}_j, j\in\bar{\Psi}\right\}$ in the inactive set $\bar{\Psi}$ are

$$
\hat{J}_j = \frac{\sqrt{H}}{K\sqrt{N}}\mathtt{soft}\left(z_j, \frac{\lambda M}{\sqrt{HN}}\right)
$$

$$
= \underset{J_j}{\arg\min}\left[\frac{1}{2}\left(J_j - \frac{1}{K}\sqrt{\frac{H}{N}}z_j\right)^2 + \frac{\lambda M}{KN}|J_j|\right], j\in\bar{\Psi},
\tag{66}
$$

which $z_j\sim\mathcal{N}\left(0,1\right), j\in\bar{\Psi}$ are i.i.d. random Gaussian noise.

Consequently, it can be seen that from (64) and (66), statistically, the $\ell_1$-LinR estimator is decoupled into two scalar thresholding estimators for the active set $\Psi$ and inactive set $\bar{\Psi}$, respectively.

### A.3.2 Logistic loss $\ell\left(y\right) = \log\left(1 + e^{-2y}\right)$

In the case of logistic lass $\ell\left(y\right) = \log\left(1 + e^{-2y}\right)$ for the $\ell_1$-LogR estimator, however, there is no analytic solution to $y$ in $\min\limits_{y}\left[\frac{\left(y - s_0\left(\sqrt{Q}z + \sum_{j \in \Psi} \bar{J}_j s_j\right)\right)^2}{2\chi} + \ell\left(y\right)\right]$ and we have to solve it together iteratively with other parameters $\Theta$. After some algebra, we obtain the EOS for the $\ell_1$-LogR estimator:

$$
\begin{cases}
\frac{\hat{y}(s,z,\chi,Q,J) - s_0\left(\sqrt{Q}z + \sum_{j \in \Psi} \bar{J}_j s_j\right)}{\chi} = 1 - \tanh\left(\hat{y}\left(s, z, \chi, Q, J\right)\right), \\
E = \alpha \mathbb{E}_{s,z}\left(\frac{s_0 z}{\sqrt{Q}}\tanh\left(\hat{y}\left(S, z, \chi, Q, J\right)\right)\right), \\
F = \alpha \mathbb{E}_{s,z}\left(\left(1 - \tanh\left(\hat{y}\left(S, z, \chi, Q, J\right)\right)\right)^2\right), \\
R = \frac{1}{K^2}\left[\left(H + \frac{\lambda^2 M^2}{N}\right)\text{erfc}\left(\frac{\lambda M}{\sqrt{2HN}}\right) - 2\lambda M\sqrt{\frac{H}{N}}\frac{1}{\sqrt{2\pi}}e^{-\frac{\lambda^2 M^2}{2HN}}\right], \\
E\eta = -\int \frac{\rho(\gamma)}{\tilde{\Lambda} - \gamma}d\gamma, \\
Q = \frac{F}{E^2} + R\tilde{\Lambda} - \left(-ER + F\eta\right)\eta\frac{1}{\int \frac{\rho(\lambda)}{(\tilde{\Lambda} - \lambda)^2}d\lambda}, \\
K = E\tilde{\Lambda} + \frac{1}{\eta}, \\
\chi = \frac{1}{E} + \eta\tilde{\Lambda}, \\
H = \frac{R}{\eta^2} + F\tilde{\Lambda} + \left(-ER + F\eta\right)E\frac{1}{\int \frac{\rho(\lambda)}{(\tilde{\Lambda} - \lambda)^2}d\lambda}, \\
\eta = \frac{1}{K}\text{erfc}\left(\frac{\lambda M}{\sqrt{2HN}}\right), \\
\bar{J}_j = \frac{\text{soft}\left(\mathbb{E}_{s,z}\left(\hat{y}(s,z,\chi,Q,J)s_0 \sum_{j \in \Psi} s_j\right), \lambda d\chi\right)}{d(1 + (d-1)\tanh^2(K_0))}, j \in \Psi.
\end{cases}
\tag{67}
$$

In the active set $\Psi$, the mean estimates $\left\{\bar{J}_j, j \in \Psi\right\}$ can be obtained by solving a reduced $\ell_1$-regularized optimization problem

$$
\min_{\left\{\bar{J}_j\right\}_{j \in \Psi}}\left\{\mathbb{E}_{s,z}\left(\min_{y}\left[\frac{\left(y - s_0\left(\sqrt{Q}z + \sum_{j \in \Psi} \bar{J}_j s_j\right)\right)^2}{2\chi} + \log\left(1 + e^{-2y}\right)\right]\right) + \lambda\sum_{j \in \Psi}\left|\bar{J}_j\right|\right\}.
\tag{68}
$$

In contrast to the $\ell_1$-LinR estimator, the mean estimates $\left\{\bar{J}_j, j \in \Psi\right\}$ in (68) for the $\ell_1$-LogR estimator do not have analytic solutions and also have to be solved numerically. For a RR graph $G \in \mathcal{G}_{N,d,K_0}$ with degree $d$ and coupling strength $K_0$, after some algebra, the corresponding fixed-point equations for $\left\{\bar{J}_j = J, j \in \Psi\right\}$ are obtained as follows

$$
J = \frac{\text{soft}\left(\mathbb{E}_{s,z}\left(\hat{y}\left(s, z, \chi, Q, J\right)s_0 \sum_{j \in \Psi} s_j\right), \lambda d\chi\right)}{d\left(1 + (d-1)\tanh^2\left(K_0\right)\right)},
\tag{69}
$$

which can be solved iteratively.

The estimates in the inactive set $\bar{\Psi}$ are the same as (66) that of $\ell_1$-LinR, which can be described by a scalar thresholding estimator once the EOS is solved.

## B  Check the consistency of ansatz (A1)

To check the consistency of Ansatz (A1), first we categorize the estimators based on the distance or generation from the focused spin $s_0$. Considering the original Ising model whose coupling network is a tree-like graph, we can naturally define generations of the spins according to the distance from the focused spin $s_0$. We categorize the spins directly connected to $s_0$ as the first generation and denote the corresponding index set as $\Omega_1 = \left\{i | J_i^* \neq 0, i \in \{1, \ldots, N-1\}\right\}$. Each spin in $\Omega_1$ is connected to some other spins except for $s_0$, and those spins constitute the second generation and we denote its index set as $\Omega_2$. This recursive construction of generations can be unambiguously continued on the tree-like graph, and we denote the index set of the $g$-th generation from spin $s_0$ as $\Omega_g$. The overall construction of generations is graphically represented in Fig. 4. Generally, assume that the set of nonzero values of the $\ell_1$-LinR estimator is denoted as $\Psi = \{\Omega_1, \ldots, \Omega_g\}$. Then, Ansatz (A1) means that the correct active set of the mean estimates is $\Psi = \{\Omega_1\}$.

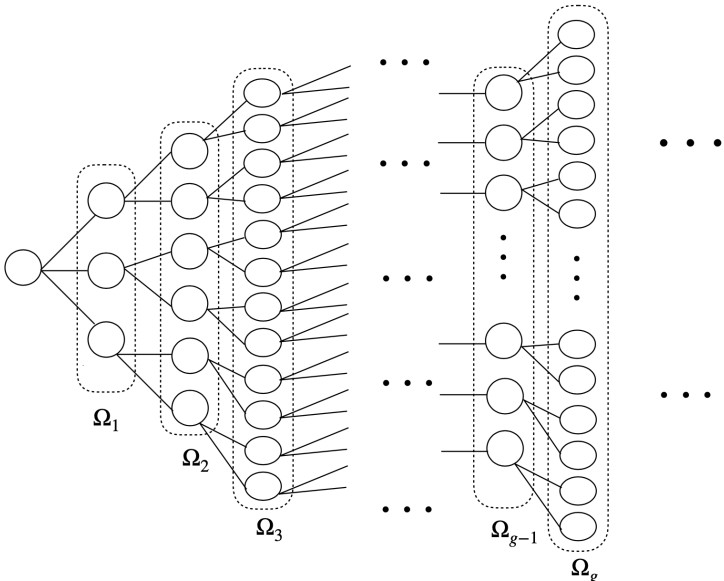

Figure 4: Schematic of generations of spins. In general, the $g$-th generation of spin $s_0$ is denoted as $\Omega_g$, whose distance from spin $s_0$ is $g$.

To verify this, we examine the values of mean estimates based on (60). Due to the symmetry, it is expected that for each $a = 1, ..., g$, the values of the mean estimates $\bar{J}_{j \in \Omega_a} = J_a$ are identical to each other within the same set $\Omega_a, a = 1...g$. In addition, if the solutions satisfy Ansatz (A1) in (11), i.e., $J_1 = J, J_a = 0, a \geq 2$, from (60) we obtain

$$\begin{cases} \frac{1}{1+\chi} \left[ \tanh\left(K_0\right) - \left(1 + (d-1)\tanh^2\left(K_0\right)\right) J \right] - \lambda = 0, & j \in \Omega_1; \\ \left| \frac{1}{1+\chi} \left[ \tanh^a\left(K_0\right) - \tanh^{a-1}\left(K_0\right) \left(1 + (d-1)\tanh^2\left(K_0\right)\right) J \right] \right| \leq \lambda, & j \in \Omega_a, a \geq 2, \end{cases} \tag{70}$$

where the result $\mathbb{E}_s\left(s_i s_j\right) = \tanh^{d_0}\left(K_0\right)$ is used for any two spins $s_i, s_j$ whose distance is $d_0$ in the RR graph $G \in \mathcal{G}_{N,d,K_0}$. Note that the solution of the first equation in (70) automatically satisfies the second equation (sub-gradient condition) since $|\tanh\left(K_0\right)| \leq 1$, which indicates that $J_1 = J, J_a = 0, a \geq 2$ is one valid solution. Moreover, the convexity of the quadratic loss function indicates that this is the unique and correct solution, which checks the Ansatz (A1).

## C   Check the consistency of ansatz (A2)

We here check the consistency of a part of the Ansatz (A2) in Sec.3.2, the orthogonal matrix $O$ diagonalizing the covariance matrix $C$ is distributed from the Haar orthogonal measure. To achieve this, we compare certain properties of the orthogonal matrix generated from the diagonalization of the covariance matrix $C$ with the orthogonal matrix which is actually generated from the Haar orthogonal measure. Specifically, we compute the cumulants of the trace of the power $k$ of the orthogonal matrix. All cumulants with degree $r \geq 3$ are shown to disappear in the large $N$ limit [41, 42]. The nontrivial cumulants are only second order cumulant with the same power $k$. We have computed these cumulants about the orthogonal matrix from the covariance matrix $C$ and found that they exhibit the same behavior as the ones generated from the true Haar measure, as shown in Fig. 5.

## D   Details of the High-dimensional asymptotic result

Here the asymptotic performance of $Precision$ and $Recall$ are considered for both the $\ell_1$-LinR estimator and the $\ell_1$-LogR estimator. Recall that perfect Ising model selection is achieved if and only if $Precision = 1$ and $Recall = 1$

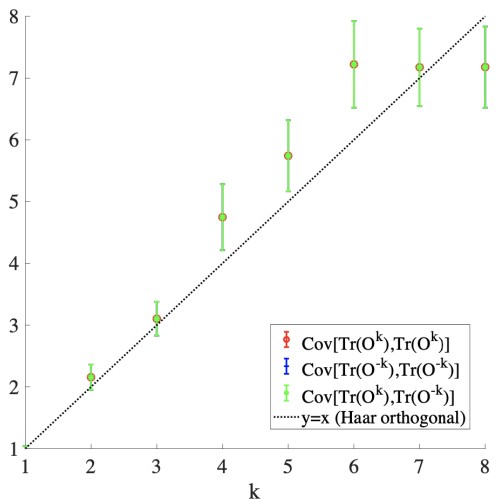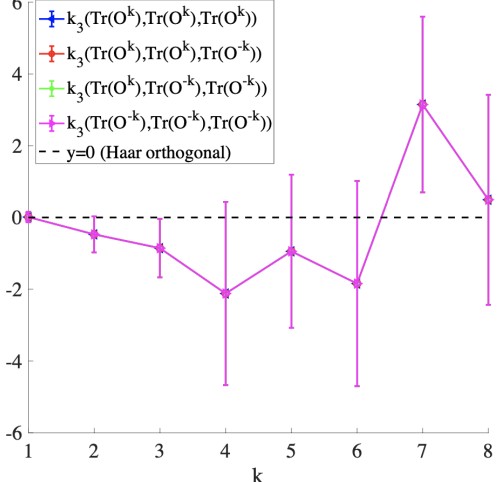

Figure 5: The RR graph $G \in \mathcal{G}_{N,d,K_0}$ with $N = 1000, d = 3, K_0 = 0.4$ is generated and we compute the associated covariance matrix $C$ and then diagonalize it as $C = O\Lambda O^T$, obtaining the orthogonal matrix $O$. Then the $\mathtt{Tr}\left(O^k\right), \mathtt{Tr}\left(O^{-k}\right)$ for several $k$ ($k = 1 \sim 8$) are computed, where $\mathtt{Tr}\left(\cdot\right)$ is the trace operation. This procedure is repeated 200 times with different random numbers, from which we obtain the ensemble of $\mathtt{Tr}\left(O^k\right)$ and $\mathtt{Tr}\left(O^{-k}\right)$. Consequently, the cumulants of 1st, 2nd, and 3rd orders are computed. All of them exhibit the expected theoretical behavior.

## D.1    Recall rate

According to the definition in (5), the recall rate is only related to the statistical properties of estimates in the active set $\Psi$ and thus the mean estimates $\left\{\bar{J}_j\right\}_{j \in \Psi}$ in the limit $M \to \infty$ are considered.

### D.1.1    quadratic loss

In this case, in the limit $M \to \infty$, the mean estimates $\left\{\bar{J}_j = J\right\}_{j \in \Psi}$ in the active set $\Psi$ are shown in (64) and rewritten as follows for ease of reference

$$J = \frac{\mathtt{soft}\left(\tanh\left(K_0\right), \lambda\left(1 + \chi\right)\right)}{1 + (d-1)\tanh^2\left(K_0\right)}. \tag{71}$$

As a result, as long as $\lambda\left(1 + \chi\right) < \tanh\left(K_0\right)$, $J > 0$ and thus we can successfully recover the active set so that $Recall = 1$. In addition, when $M = \mathcal{O}\left(\log N\right)$, $\chi \to 0$ as $N \to \infty$, as demonstrated later by the relation in (81). As a result, the regularization parameter needs to satisfy $0 < \lambda < \tanh\left(K_0\right)$.

### D.1.2    Logistic loss

In this case, in the limit $M \to \infty$, the mean estimates $\left\{\bar{J}_j = J\right\}_{j \in \Psi}$ in the active set $\Psi$ are shown in (69) and rewritten as follows for ease of reference

$$J = \frac{\mathtt{soft}\left(\mathbb{E}_{s,z}\left(\hat{y}\left(s, z, \chi, Q, J\right) s_0 \sum_{j \in \Psi} s_j\right), \lambda d\chi\right)}{d\left(1 + (d-1)\tanh^2\left(K_0\right)\right)}. \tag{72}$$

There is no analytic solution for $\hat{y}\left(s, z, \chi, Q, J\right)$ and the following fixed-point equation has to be solved numerically

$$\frac{\hat{y}\left(s, z, \chi, Q, J\right) - s_0\left(\sqrt{Q}z + J\sum_{j \in \Psi} s_j\right)}{\chi} = 1 - \tanh\left(\hat{y}\left(s, z, \chi, Q, J\right)\right). \tag{73}$$

Then one can determine the valid choice of $\lambda$ to enable $J > 0$. Numerical results show that the choice of $\lambda$ is similar to that of the quadratic loss.

## D.2 Precision rate

According to the definition in (5), to compute the $Precision$, the number of true positives $TP$ and false positives $FP$ are needed, respectively. On the one hand, as discussed in Appendix D.1, in the limit $M \to \infty$, the recall rate approach to one and thus we have $TP = d$ for a RR graph $G \in \mathcal{G}_{N,d,K_0}$. On the other hand, the number of false positives $FP$ can be computed as $FP = FPR \cdot N$, where $FPR$ is the false positive rate (FPR).

As shown in Appendix A.3, the estimator in the inactive set $\bar{\Psi}$ can be statistically described by a scalar estimator (66) and thus the $FPR$ can be computed as

$$FPR = \mathrm{erfc}\left(\frac{\lambda M}{\sqrt{2HN}}\right), \tag{74}$$

which depends on $\lambda, M, N, H$. However, for both the quadratic loss and logistic loss, there is no analytic result for $H$ in (59). Nevertheless, we can obtain some asymptotic result using perturbative analysis.

Specifically, we focus on the asymptotic behavior of the macroscopic parameters, e.g., $\chi, Q, K, E, H, F$, in the regime $FPR \to 0$, which is necessary for successful Ising model selection. From $\eta = \frac{1}{K}\mathrm{erfc}\left(\frac{\lambda M}{\sqrt{2HN}}\right)$ in EOS (59) and the $FPR$ in (74), there is $FPR = K\eta$. Moreover, by combining $E\eta = -\int \frac{\rho(\gamma)}{\tilde{\Lambda}-\gamma}d\gamma$ and $K = E\tilde{\Lambda} + \frac{1}{\eta}$, the following relation can be obtained

$$\mathrm{erfc}\left(\frac{\lambda M}{\sqrt{2HN}}\right) = 1 - \int \frac{\rho(\gamma)}{1-\frac{\gamma}{\tilde{\Lambda}}}d\gamma. \tag{75}$$

Thus as $FPR = \mathrm{erfc}\left(\frac{\lambda M}{\sqrt{2HN}}\right) \to 0$, there is $\int \frac{\rho(\gamma)}{1-\frac{\gamma}{\tilde{\Lambda}}}d\gamma \to 1$, implying that the magnitude of $\tilde{\Lambda} \to \infty$. Consequently, using the truncated series expansion, we obtain

$$\begin{aligned} E\eta &= -\int \frac{\rho(\gamma)}{\tilde{\Lambda}-\gamma}d\gamma \\ &= -\frac{1}{\tilde{\Lambda}}\sum_{k=0}^{\infty}\frac{\langle\gamma^k\rangle}{\tilde{\Lambda}^k} \\ &\simeq -\frac{1}{\tilde{\Lambda}} - \frac{\langle\gamma\rangle}{\tilde{\Lambda}^2}, \end{aligned} \tag{76}$$

where $\langle\gamma^k\rangle = \int \rho(\gamma)\gamma^k d\gamma$. Then, solving the quadratic equation (76), we obtain the solution (the other solution is not considered since it is a smaller value) of $\tilde{\Lambda}$ as

$$\tilde{\Lambda} = \frac{-1 - \sqrt{1 - 4E\eta\langle\gamma\rangle}}{2E\eta} \simeq \langle\gamma\rangle - \frac{1}{E\eta}. \tag{77}$$

To compute $\int \frac{\rho(\gamma)}{(\tilde{\Lambda}-\gamma)^2}d\gamma$, we use the following relation

$$f(\tilde{\Lambda}) = -\int \frac{\rho(\gamma)}{\tilde{\Lambda}-\gamma}d\gamma \simeq -\frac{1}{\tilde{\Lambda}} - \frac{\langle\gamma\rangle}{\tilde{\Lambda}^2}, \tag{78}$$

$$\frac{df(\tilde{\Lambda})}{d\tilde{\Lambda}} = \int \frac{\rho(\gamma)}{(\tilde{\Lambda}-\gamma)^2}d\gamma \simeq \frac{1}{\tilde{\Lambda}^2} + 2\frac{\langle\gamma\rangle}{\tilde{\Lambda}^3}. \tag{79}$$

Substituting the results (77) - (79) into (59), after some algebra, we obtain

$$K \simeq E\langle\gamma\rangle, \tag{80}$$

$$\chi \simeq \eta\langle\gamma\rangle, \tag{81}$$

$$Q \simeq \frac{\langle\gamma\rangle^3 E^2\eta^2 R - \langle\gamma\rangle^3 EF\eta^3 + 3\langle\gamma\rangle^2 F\eta^2 - R\langle\gamma\rangle}{3E\eta\langle\gamma\rangle - 1}, \tag{82}$$

$$H \simeq \frac{\langle\gamma\rangle^3 E^2\eta^2 F - \langle\gamma\rangle^3 R\eta E^3 + 3\langle\gamma\rangle^2 RE^2 - F\langle\gamma\rangle}{3E\eta\langle\gamma\rangle - 1}. \tag{83}$$

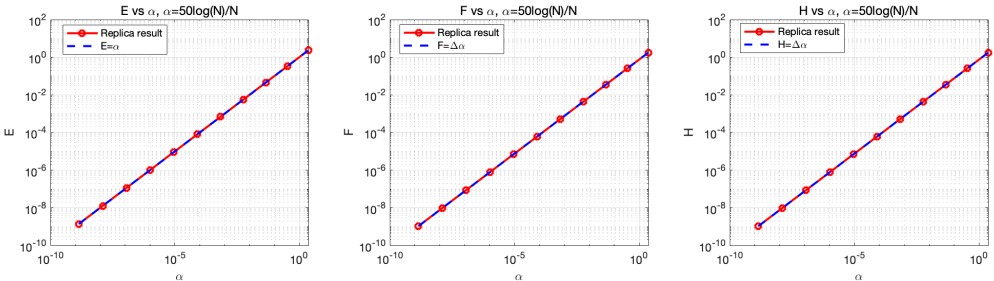

Figure 6: $E, F, H$ versus $\alpha$ when $\alpha = 50(\log N)/N$ for $N = 10^2 \sim 10^{12}$ for RR graph $G \in \mathcal{G}_{N,d,K_0}$ with $d = 3, K_0 = 0.4$. Note that in this case, there is $\langle \gamma \rangle = 1$.

In addition, as $FPR = \text{erfc}\left(\frac{\lambda M}{\sqrt{2HN}}\right) \to 0$, from (59) we obtain

$$
\begin{aligned}
R &= \frac{1}{K^2}\left[\left(H + \frac{\lambda^2 M^2}{N}\right)\text{erfc}\left(\frac{\lambda M}{\sqrt{2HN}}\right) - 2\lambda M\sqrt{\frac{H}{N}}\frac{1}{\sqrt{2\pi}}e^{-\frac{\lambda^2 M^2}{2HN}}\right] \\
&\simeq \frac{H}{K^2}\text{erfc}\left(\frac{\lambda M}{\sqrt{2HN}}\right) \simeq \frac{H}{K}\eta \simeq \frac{H}{E\langle\gamma\rangle}\eta,
\end{aligned} \tag{84}
$$

where the first result in $\simeq$ uses the asymptotic relation $\text{erfc}(x) \simeq \frac{1}{x\sqrt{\pi}}e^{-x^2}$ as $x \to \infty$ and the last result in $\simeq$ results from the asymptotic relation in (80). Then, substituting (84) into (83) leads to the following relation

$$(3E\eta\langle\gamma\rangle - 1)H \simeq \langle\gamma\rangle^3 E^2\eta^2 F - \langle\gamma\rangle^2 \eta^2 E^2 H + 3E\eta\langle\gamma\rangle H - F\langle\gamma\rangle. \tag{85}$$

Interestingly, the common terms $3E\eta\langle\gamma\rangle H$ in both sides of (85) cancel with each other. Therefore, the key result for $H$ is obtained as follows

$$H \simeq F\langle\gamma\rangle. \tag{86}$$

In addition, from (86) and (82), $Q$ can be simplified as

$$Q \simeq R\langle\gamma\rangle. \tag{87}$$

As shown in (59), $F = \alpha\mathbb{E}_{s,z}\left(\frac{d\ell(y)}{dy}\Big|_{y=\hat{y}(s,z,\chi,Q,J)}\right)^2$, thus the result $H \simeq F\langle\gamma\rangle$ in (86) implies that there is a linear relation between $H$ and $\alpha \equiv M/N$. The relation between $E, F, H$ and $\alpha$ are also verified numerically in Fig. 6 when $M = 50(\log N)$ for $N = 10^2 \sim 10^{12}$ using the $\ell_1$-LinR estimator.

In the paramagnetic phase, it can be obtained that the mean value of the eigenvalue $\langle\gamma\rangle$. Specifically, we have $\langle\gamma\rangle = \frac{1}{N}\sum_{i=1}^N \gamma_i = \frac{1}{N}\text{Tr}C = (1/N) \times N = 1$. Denote by $H \simeq F\langle\gamma\rangle \equiv \alpha\triangle$, where $\triangle = \mathbb{E}_{s,z}\left(\frac{d\ell(y)}{dy}\Big|_{y=\hat{y}(s,z,\chi,Q,J)}\right)^2 = \mathcal{O}(1)$, then the $FPR$ in (74) can be rewritten as follows

$$
\begin{aligned}
FPR &= \text{erfc}\left(\frac{\lambda M}{\sqrt{2\alpha\triangle N}}\right) \\
&= \text{erfc}\left(\lambda\sqrt{\frac{M}{2\triangle}}\right) \\
&\leq \frac{1}{\sqrt{\pi}}e^{-\frac{\lambda^2 M}{2\triangle} - \frac{1}{2}\log\left(\frac{\lambda^2 M}{2\triangle}\right)},
\end{aligned} \tag{88}
$$

where the last inequality uses the upper bound of erfc function, i.e., $\text{erfc}(x) \leq \frac{1}{x\sqrt{\pi}} e^{-x^2}$. Consequently, the number of false positives $FP$ satisfies

$$
\begin{aligned}
FP &\leq \frac{N}{\sqrt{\pi}} e^{-\frac{\lambda^2 M}{2\triangle} - \frac{1}{2}\log\left(\frac{\lambda^2 M}{2\triangle}\right)} \\
&= \frac{1}{\sqrt{\pi}} e^{-\frac{\lambda^2 M}{2\triangle} - \frac{1}{2}\log\left(\frac{\lambda^2 M}{2\triangle}\right) + \log N} \\
&< \frac{1}{\sqrt{\pi}} e^{-\frac{\lambda^2 M}{2\triangle} + \log N},
\end{aligned}
\tag{89}
$$

where the last inequality holds when $\frac{\lambda^2 M}{2\triangle} > 1$, which is necessary when $FP \to 0$ as $N \to \infty$. Consequently, to ensure $FP \to 0$ as $N \to \infty$, from (89), the term $\frac{\lambda^2 M}{2\triangle}$ should grow at least faster than $\log N$, i.e.,

$$
M > \frac{2\triangle \log N}{\lambda^2}.
\tag{90}
$$

Meanwhile, the number of false positives $FP$ will decay as $\mathcal{O}\left(e^{-c\log N}\right)$ for some constant $c\,(>0)$.

### D.2.1 Quadratic loss

In this case, when $0 < \lambda < \tanh(K_0)$, from (61), we can obtain an analytic result for $\triangle$ as follows

$$
\triangle \simeq \mathbb{E}_{s_0}\left(s - \sum_{j\in\Psi} s_j \bar{J}_j\right)^2
\tag{91}
$$

$$
= \frac{1 - \tanh^2 K_0 + d\lambda^2}{1 + (d-1)\tanh^2 K_0}.
\tag{92}
$$

On the other hand, from the discussion in Appendix D.1, the recall rate $Recall \to 1$ as $M \to \infty$ when $0 < \lambda < \tanh K_0$. Overall, for a RR graph $G \in \mathcal{G}_{N,d,K_0}$ with degree $d$ and coupling strength $K_0$, given $M$ i.i.d. samples $\mathcal{D}^M = \left\{s^{(1)}, ..., s^{(M)}\right\}$, using $\ell_1$-LinR estimator (4) with regularization parameter $\lambda$, perfect recovery of the graph structure $G$ can be achieved as $N \to \infty$ if the number of samples $M$ satisfies

$$
M > \frac{c(\lambda, K_0)\log N}{\lambda^2}, \lambda \in (0, \tanh(K_0))
\tag{93}
$$

where $c(\lambda, K_0)$ is a value dependent on the regularization parameter $\lambda$ and coupling strength $K_0$, which can be approximated in the limit $N \to \infty$ as:

$$
c(\lambda, K_0) = \frac{2\left(1 - \tanh^2(K_0) + d\lambda^2\right)}{1 + (d-1)\tanh^2(K_0)}.
\tag{94}
$$

### D.2.2 Logistic loss

In this case, from (67), the value of $\triangle$ can be computed as

$$
\triangle \simeq \mathbb{E}_{s,z}\left((1 - \tanh(\hat{y}(S, z, \chi, Q, J)))^2\right).
\tag{95}
$$

However, different from the case of $\ell_1$-LinR estimator, there is no analytic solution but it can be calculated numerically. It can be seen that the $\ell_1$-LinR estimator only differs in the value of scaling factor $\triangle$ with the $\ell_1$-LogR estimator for Ising model selection.

## E   Details of the non-asymptotic result for moderate $M, N$

As demonstrated in Appendix A.3, from the replica analysis, both $\ell_1$-LinR and $\ell_1$-LogR estimators are decoupled and their asymptotic behavior can be described by two scalar estimators for the active set and inactive set, respectively. It is desirable to obtain the non-asymptotic result for moderate

$M, N$. However, it is found that the behavior of the two scalar estimators by simply inserting the finite values of $M, N$ into the EOS does not always lead to good consistency with the experimental results, especially for the *Recall* when $M$ is small. This can be explained by the derivation of the free energy density. In calculating the energy term $\xi$, the limit $M \to \infty$ is taken implicitly when assuming the limit $N \to \infty$ with $\alpha \equiv M/N$. As a result, the scalar estimator associated with the active set can only describe the asymptotic performance in the limit $M \to \infty$. Thus, one cannot describe the fluctuating behavior of the estimator in the active set such as the recall rate for finite $M$. To characterize the non-asymptotic behavior of the estimates in the active set $\Psi$, we replace the expectation $\mathbb{E}_s(\cdot)$ in (58) by the sample average over $M$ samples, and the corresponding estimates are obtained as

$$
\left\{\hat{J}_j\right\}_{j \in \Psi} = \underset{J_{j}, j \in \Psi}{\arg\min} \left\{\frac{1}{M} \sum_{\mu=1}^{M} \min_{y^\mu} \left[\frac{\left(y^\mu - s_0^\mu \left(\sqrt{Q} z^\mu + \sum_{j \in \Psi} J_j s_j^\mu\right)\right)^2}{2\chi} + \ell\left(y^\mu\right)\right] + \lambda \sum_{j \in \Psi} |J_j|\right\},
$$
(96)

where $z^\mu \sim \mathcal{N}(0,1)$ and $s_0^\mu, s_{j, j \in \Psi}^\mu \sim P(s_0, \boldsymbol{s}_\Psi | \boldsymbol{J}^*)$ are random samples $\mu = 1, ..., M$. Note that the mean estimates $\left\{\bar{J}_j\right\}_{j \in \Psi}$ are replaced by $\left\{\hat{J}_j\right\}_{j \in \Psi}$ in (96) as we now focus on its fluctuating behavior due to the finite size effect. In the limit $M \to \infty$, the sample average will converge to the expectation and thus (96) is equivalent to (68) when $M \to \infty$.

### E.1 quadratic loss $\ell(y) = (y-1)^2/2$

In the case of quadratic loss $\ell(y) = (y-1)^2/2$, there is an analytic solution to $y$ in $\min_{y} \left[\frac{\left(y - s_0\left(\sqrt{Q} z + \sum_{j \in \Psi} \bar{J}_j s_j\right)\right)^2}{2\chi} + \ell(y)\right]$. Consequently, similar to (62), the result of (96) for the $\ell_1$-LinR estimator becomes

$$
\left\{\hat{J}_j\right\}_{j \in \Psi} = \underset{J_{j}, j \in \Psi}{\arg\min} \left[\frac{1}{2(1+\chi)M} \sum_{\mu=1}^{M} \left(s_i^\mu - \sum_{j \in \Psi} s_j^\mu J_j - \sqrt{Q} z^\mu\right)^2 + \lambda \sum_{j \in \Psi} |J_j|\right].
$$
(97)

As the mean estimates $\left\{\bar{J}_j\right\}_{j \in \Psi}$ are modified as in (97), the corresponding solution to the EOS in (61) also needs to be modified, and this can be solved iteratively as sketched in Algorithm 1. For a practical implementation of Algorithm 1, the details are described in the following.

First, in the EOS (19), we need to obtain $\tilde{\Lambda}$ satisfying the following relation

$$
E\eta = -\int \frac{\rho(\gamma)}{\tilde{\Lambda} - \gamma} d\gamma,
$$
(98)

which is difficult to solve directly. To obtain $\tilde{\Lambda}$, we introduce an auxiliary variable $\Gamma \equiv -\frac{1}{\tilde{\Lambda}}$, by which (98) can be rewritten as

$$
\Gamma = \frac{E\eta}{\int \frac{\rho(\gamma)}{1+\Gamma\gamma} d\gamma},
$$
(99)

which can be solved iteratively. Accordingly, the $\chi, Q, K, H$ in EOS (19) can be equivalently written in terms of $\Gamma$.

Second, when solving the EOS (19) iteratively using numerical methods, it is helpful to improve the convergence of the solution by introducing a small amount of damping factor `damp` $\in [0, 1)$ for $\chi, Q, E, R, F, \eta, K, H, \Gamma$ in each iteration.

The detailed implementation of Algorithm 1 is shown in Algorithm 2.

**Algorithm 2:** Detailed implementation of Algorithm 1 for the $\ell_1$-LinR estimator with moderate $M, N$.

---

**Input:** $M, N, \lambda, K_0, \rho(\gamma), \texttt{damp}, T_{\mathrm{MC}}$
**Output:** $\chi, Q, E, R, F, \eta, K, H, \Gamma, \{\hat{J}_{j,j \in \Psi}^t\}_{t=1}^{T_{MC}}$
**Initialization:** $\chi, Q, E, R, F, \eta, K, H, \Gamma$

1 **MC sampling**: For $t = 1...T_{MC}$, draw random samples $s_0^{\mu,t}, \{s_j^{\mu,t}\}_{j \in \Psi} \sim P(s_0, \boldsymbol{s}_\Psi | \boldsymbol{J}^*)$ and $z^{\mu,t} \sim \mathcal{N}(0,1), \mu = 1...M$

2 **repeat**

3    **for** $t = 1$ **to** $T_{\mathrm{MC}}$ **do**

4      Solve $\hat{J}_{j,j \in \Psi}^t = \underset{J_{j,j \in \Psi}}{\arg\min} \left[ \frac{\sum_{\mu=1}^M \left(s_0^{\mu,t} - \sum_{j \in \Psi} s_j^{\mu,t} J_j - \sqrt{Q} z^{\mu,t}\right)^2}{2(1+\chi)M} + \lambda \sum_{j \in \Psi} |J_j| \right]$

5      Compute $\triangle(t) = \frac{1}{M} \sum_{\mu=1}^M \left( s_0^{\mu,t} - \sum_{j \in \Psi} s_j^{\mu,t} \hat{J}_j^t \right)^2$

6    Set $\bar{\triangle} = \frac{1}{T_{\mathrm{MC}}} \sum_{t_1=1}^{T_{\mathrm{MC}}} \triangle(t)$

7    $E = (1 - \texttt{damp}) \frac{\alpha}{(1+\chi)} + \texttt{damp} \cdot E$

8    $F = (1 - \texttt{damp}) \frac{\alpha}{(1+\chi)^2} (\bar{\triangle} + Q) + \texttt{damp} \cdot F$

9    $R = (1 - \texttt{damp}) \frac{1}{K^2} \left[ \left(H + \frac{\lambda^2 M^2}{N}\right) \mathrm{erfc}\left(\frac{\lambda M}{\sqrt{2HN}}\right) - 2\lambda M \sqrt{\frac{H}{N}} \frac{1}{\sqrt{2\pi}} e^{-\frac{\lambda^2 M^2}{2HN}} \right] + \texttt{damp} \cdot R$

10    **repeat**

11      $\Gamma = (1 - \texttt{damp}) \frac{E\eta}{\int \frac{\rho(\gamma)}{1+\Gamma\gamma} d\gamma} + \texttt{damp} \cdot \Gamma$

12    **until** *convergence*

13    $K = (1 - \texttt{damp}) \left(-\frac{E}{\Gamma} + \frac{1}{\eta}\right) + \texttt{damp} \cdot K$

14    $\chi = (1 - \texttt{damp}) \left(-\frac{\eta}{\Gamma} + \frac{1}{E}\right) + \texttt{damp} \cdot \chi$

15    $Q = (1 - \texttt{damp}) \left( \frac{F}{E^2} - \frac{R}{\Gamma} - \frac{(-ER+F\eta)\eta}{\Gamma^2 \int \frac{\rho(\gamma)}{(1+\Gamma\gamma)^2} d\gamma} \right) + \texttt{damp} \cdot Q$

16    $H = (1 - \texttt{damp}) \left( \frac{R}{E^2} - \frac{F}{\Gamma} - \frac{(-ER+F\eta)E}{\Gamma^2 \int \frac{\rho(\gamma)}{(1+\Gamma\gamma)^2} d\gamma} \right) + \texttt{damp} \cdot H$

17    $\eta = (1 - \texttt{damp}) \frac{1}{K} \mathrm{erfc}\left(\frac{\lambda M}{\sqrt{2HN}}\right) + \texttt{damp} \cdot \eta$

18 **until** *convergence*

---

## E.2  Logistic loss $\ell(y) = \log\left(1 + e^{-2y}\right)$

In the case of square lass $\ell(y) = \log\left(1 + e^{-2y}\right)$, since there is no analytic solution to $y$ in $\min_y \left[ \frac{\left(y - s_0\left(\sqrt{Q}z + \sum_{j \in \Psi} \bar{J}_j s_j\right)\right)^2}{2\chi} + \ell(y) \right]$, the result of (96) for the $\ell_1$-LogR estimator becomes

$$\hat{J}_{j,j \in \Psi} = \underset{J_{j,j \in \Psi}}{\arg\min} \left[ \frac{1}{M} \sum_{\mu=1}^M \min_{y^\mu} \left[ \frac{\left(y^\mu - s_0^\mu \left(\sqrt{Q}z^\mu + \sum_{j \in \Psi} J_j s_j^\mu\right)\right)^2}{2\chi} + \log\left(1 + e^{-2y}\right) \right] + \lambda \sum_{j \in \Psi} |J_j^\mu| \right],$$
(100)

Similarly as the case for quadratic loss, as the mean estimates $\{\bar{J}_j\}_{j \in \Psi}$ are modified as in (100), the corresponding solutions to the EOS in (67) also need to be modified, which can be solved iteratively as shown in Algorithm 3.

## F  Eigenvalue Distribution $\rho(\gamma)$

From the replica analysis presented, the learning performance will depend on the eigenvalue distribution (EVD) $\rho(\gamma)$ of the covariance matrix $C$ of the original Ising model.

There are two issues to be noted. One is about the formula connecting the performance of the estimator and the spectral density, and the other is the numeric values of quantities which are computed from

**Algorithm 3:** Detailed implementation of solving the EOS (67) together with (100) for $\ell_1$-LogR with moderate $M, N$.

---

**Input:** $M, N, \lambda, K_0, \rho(\gamma), \texttt{damp}, T_{\mathrm{MC}}$
**Output:** $\chi, Q, E, R, F, \eta, K, H, \Gamma, \{\hat{J}^t_{j,j\in\Psi}\}_{t=1}^{T_{MC}}$
**Initialization:** $\chi, Q, E, R, F, \eta, K, H, \Gamma$

1 **MC sampling**: For $t = 1...T_{MC}$, draw random samples $s_0^{\mu,t}, \{s_j^{\mu,t}\}_{j\in\Psi} \sim P(s_0, \boldsymbol{s}_\Psi|\boldsymbol{J}^*)$ and $z^{\mu,t} \sim \mathcal{N}(0,1), \mu = 1...M$

2 **repeat**

3     **for** $t = 1$ **to** $T_{\mathrm{MC}}$ **do**

4         Initialization $\hat{J}^t_{j,j\in\Psi}$

5         **repeat**

6             $\hat{y}^{\mu,t} = \underset{y^{\mu,t}}{\arg\min}\left[\frac{(y^\mu - s_0^{\mu,t}(\sqrt{Q}z^{\mu,t} + \sum_{j\in\Psi}\hat{J}_j s_j^{\mu,t}))^2}{2\chi} + \log\left(1 + e^{-2y^\mu}\right)\right], \mu = 1...M$

7             $\hat{J}^t_{j,j\in\Psi} = \underset{J_{j,j\in\Psi}}{\arg\min}\Big\{\frac{1}{M}\sum_{\mu=1}^M\left[\frac{(\hat{y}^{\mu,t} - s_0^{\mu,t}(\sqrt{Q}z^{\mu,t} + \sum_{j\in\Psi}J_j^t s_j^{\mu,t}))^2}{2\chi} + \log\left(1 + e^{-2\hat{y}^{\mu,t}}\right)\right] +$
            $\lambda\sum_{j\in\Psi}|J_j|\Big\}$

8         **until** *convergence*

9         Compute $\triangle_1(t) = \frac{1}{M}\sum_{\mu=1}^M\left(\frac{s_0^{\mu,t}z^{\mu,t}}{-\sqrt{Q}}(1 - \tanh(\hat{y}^{\mu,t}))\right)$

10         Compute $\triangle_2(t) = \frac{1}{M}\sum_{\mu=1}^M(1 - \tanh(\hat{y}^{\mu,t}))^2$

11     Set $\bar{\triangle}_1 = \frac{1}{T_{\mathrm{MC}}}\sum_{t=1}^{T_{\mathrm{MC}}}\triangle_1(t)$ and $\bar{\triangle}_2 = \frac{1}{T_{\mathrm{MC}}}\sum_{t=1}^{T_{\mathrm{MC}}}\triangle_2(t)$

12     $E = (1 - \texttt{damp})\cdot\alpha\bar{\triangle}_1 + \texttt{damp}\cdot E$

13     $F = (1 - \texttt{damp})\cdot\alpha\bar{\triangle}_2 + \texttt{damp}\cdot F$

14     $R = (1 - \texttt{damp})\frac{1}{K^2}\left[\left(H + \frac{\lambda^2 M^2}{N}\right)\mathrm{erfc}\left(\frac{\lambda M}{\sqrt{2HN}}\right) - 2\lambda M\sqrt{\frac{H}{N}}\frac{1}{\sqrt{2\pi}}e^{-\frac{\lambda^2 M^2}{2HN}}\right] + \texttt{damp}\cdot R$

15     **repeat**

16         $\Gamma = (1 - \texttt{damp})\frac{E\eta}{\int\frac{\rho(\gamma)}{1+\Gamma\gamma}d\gamma} + \texttt{damp}\cdot\Gamma$

17     **until** *convergence*

18     $K = (1 - \texttt{damp})\left(-\frac{E}{\Gamma} + \frac{1}{\eta}\right) + \texttt{damp}\cdot K$

19     $\chi = (1 - \texttt{damp})\left(-\frac{\eta}{\Gamma} + \frac{1}{E}\right) + \texttt{damp}\cdot\chi$

20     $Q = (1 - \texttt{damp})\left(\frac{F}{E^2} - \frac{R}{\Gamma} - \frac{(-ER+F\eta)\eta}{\Gamma^2\int\frac{\rho(\gamma)}{(1+\Gamma\gamma)^2}d\gamma}\right) + \texttt{damp}\cdot Q$

21     $H = (1 - \texttt{damp})\left(\frac{R}{E^2} - \frac{F}{\Gamma} - \frac{(-ER+F\eta)E}{\Gamma^2\int\frac{\rho(\gamma)}{(1+\Gamma\gamma)^2}d\gamma}\right) + \texttt{damp}\cdot H$

22     $\eta = (1 - \texttt{damp})\frac{1}{K}\mathrm{erfc}\left(\frac{\lambda M}{\sqrt{2HN}}\right) + \texttt{damp}\cdot\eta$

23 **until** *convergence*

---

the formula. For the first point, no assumption about the spectral density is needed to obtain the formula itself and this formula is valid when the graph structure is tree-like and the Ising model defined on the graph is in the paramagnetic phase. For the second point, we need the specific form of the spectral density to obtain numeric solutions in general. As a demonstration, we assume the random regular graph with constant coupling strength for which the spectral density can be obtained analytically as has already been known before in [7].

In general, it is difficult to obtain this EVD; however, for sparse tree-like graphs such as RR graph $G \in \mathcal{G}_{N,d,K_0}$ with constant node degree $d$ and sufficiently small coupling strength $K_0$ that yields the paramagnetic state ($\mathbb{E}_{\boldsymbol{s}}(\boldsymbol{s}) = \boldsymbol{0}$), it can be computed analytically. For this, we express the covariances as

$$C_{ij} = \mathbb{E}_{\boldsymbol{s}}(s_i s_j) - \mathbb{E}_{\boldsymbol{s}}(s_i)\mathbb{E}_{\boldsymbol{s}}(s_j) = \frac{\partial^2 \log Z(\boldsymbol{\theta})}{\partial\theta_i\partial\theta_j}, \tag{101}$$

where $Z(\boldsymbol{\theta}) = \int d\boldsymbol{s} P_{\text{Ising}}(\boldsymbol{s}|J^*) \exp(\sum_{i=0}^{N-1} \theta_i s_i)$ and the assessment is carried out at $\boldsymbol{\theta} = \mathbf{0}$.

In addition, for technical convenience we introduce the Gibbs free energy as

$$A\left(\boldsymbol{m}\right) = \max_{\boldsymbol{\theta}} \left\{\boldsymbol{\theta}^T \boldsymbol{m} - \log Z\left(\boldsymbol{\theta}\right)\right\}. \tag{102}$$

The definition of (102) indicates that following two relations hold:

$$\frac{\partial m_i}{\partial \theta_j} = \frac{\partial^2 \log Z(\boldsymbol{\theta})}{\partial \theta_i \partial \theta_j} = C_{ij},$$

$$\frac{\partial \theta_i}{\partial m_j} = [C^{-1}]_{ij} = \frac{\partial^2 A(\boldsymbol{m})}{\partial m_i \partial m_j}, \tag{103}$$

where the evaluations are performed at $\boldsymbol{\theta} = \mathbf{0}$ and $\boldsymbol{m} = \arg\min_{\boldsymbol{m}} A(\boldsymbol{m})$ ($= \mathbf{0}$ under the paramagnetic assumption).

Consequently, we can focus on the computation of $A\left(\boldsymbol{m}\right)$ to obtain the EVD of $C^{-1}$. The inverse covariance matrix of a RR graph $G \in \mathcal{G}_{N,d,K_0}$ can be computed from the Hessian of the Gibbs free energy [7, 47, 48] as

$$\begin{aligned}\left[C^{-1}\right]_{ij} &= \frac{\partial A\left(\boldsymbol{m}\right)}{\partial m_i \partial m_j} \\ &= \left(\frac{d}{1 - \tanh^2 K_0} - d + 1\right)\delta_{ij} - \frac{\tanh\left(J_{ij}\right)}{1 - \tanh^2\left(J_{ij}\right)}\left(1 - \delta_{ij}\right),\end{aligned} \tag{104}$$

and in matrix form, we have

$$C^{-1} = \left(\frac{d}{1 - \tanh^2 K_0} - d + 1\right)\mathbf{I} - \frac{\tanh\left(\boldsymbol{J}\right)}{1 - \tanh^2\left(\boldsymbol{J}\right)}, \tag{105}$$

where $\mathbf{I}$ is an identity matrix of proper size, and the operations $\tanh\left(\cdot\right), \tanh^2\left(\cdot\right)$ on matrix $\boldsymbol{J}$ are defined in the component-wise manner. For RR graph $G \in \mathcal{G}_{N,d,K_0}$, $\boldsymbol{J}$ is a sparse matrix, therefore the matrix $\frac{\tanh(\boldsymbol{J})}{1-\tanh^2(\boldsymbol{J})}$ also corresponds to a sparse coupling matrix (whose nonzero coupling positions are the same as $\boldsymbol{J}$) with constant coupling strength $K_1 = \frac{\tanh(K_0)}{1-\tanh^2(K_0)}$ and fixed connectivity $d$, the corresponding eigenvalue (denoted as $\zeta$) distribution can be calculated as [49]

$$\rho_\zeta\left(\zeta\right) = \frac{d\sqrt{4K_1^2\left(d-1\right) - \zeta^2}}{2\pi\left(K_1^2 d^2 - \zeta^2\right)}, \ |\zeta| \le 2K_1\sqrt{d-1}. \tag{106}$$

From (105), the eigenvalue $\eta$ of $C^{-1}$ is

$$\eta_i = \frac{d}{1 - \tanh^2 K_0} - d + 1 - \zeta_i, \tag{107}$$

which, when combined with (106), readily yields the EVD of $\eta$ as $N \to \infty$ as follows:

$$\begin{aligned}\rho_\eta\left(\eta\right) &= \rho_\zeta\left(\frac{d}{1 - \tanh^2 K_0} - d + 1 - \eta\right) \\ &= \frac{d\sqrt{4\left(\frac{\tanh(K_0)}{1-\tanh^2(K_0)}\right)^2\left(d-1\right) - \left(\frac{d}{1-\tanh^2 K_0} - d + 1 - \eta\right)^2}}{2\pi\left(\left(\frac{\tanh(K_0)}{1-\tanh^2(K_0)}\right)^2 d^2 - \left(\frac{d}{1-\tanh^2 K_0} - d + 1 - \eta\right)^2\right)},\end{aligned} \tag{108}$$

where $\eta \in \left[\frac{d}{1-\tanh^2 K_0} - d + 1 - \frac{2\tanh(K_0)\sqrt{d-1}}{1-\tanh^2(K_0)}, \frac{d}{1-\tanh^2 K_0} - d + 1 + \frac{2\tanh(K_0)\sqrt{d-1}}{1-\tanh^2(K_0)}\right]$.

Consequently, since $\gamma = 1/\eta$, we obtain the EVD of $\rho\left(\gamma\right)$ as follows

$$\begin{aligned}\rho\left(\gamma\right) &= \frac{1}{\gamma^2}\rho_\eta\left(\eta = \frac{1}{\gamma}\right) \\ &= \frac{d\sqrt{4\left(\frac{\tanh(K_0)}{1-\tanh^2(K_0)}\right)^2\left(d-1\right) - \left(\frac{d}{1-\tanh^2 K_0} - d + 1 - \frac{1}{\gamma}\right)^2}}{2\pi\gamma^2\left(\left(\frac{\tanh(K_0)}{1-\tanh^2(K_0)}\right)^2 d^2 - \left(\frac{d}{1-\tanh^2 K_0} - d + 1 - \frac{1}{\gamma}\right)^2\right)}\end{aligned} \tag{109}$$

where $\gamma \in \left[1/\left(\frac{d}{1-\tanh^2 K_0} - d + 1 + \frac{2\tanh(K_0)\sqrt{d-1}}{1-\tanh^2(K_0)}\right), 1/\left(\frac{d}{1-\tanh^2 K_0} - d + 1 - \frac{2\tanh(K_0)\sqrt{d-1}}{1-\tanh^2(K_0)}\right)\right]$.

## G  Additional Experimental Results

Fig. 7 and Fig. 8 show the full results of non-asymptotic learning performance prediction when $\lambda = 0.1$ and $\lambda = 0.3$, respectively. Good agreements between replica results and experimental results are achieved in all cases. As can be seen, there is negligible difference in $Precision$ and $Recall$ between $\ell_1$-LinR and $\ell_1$-LogR. Meanwhile, compared to Fig. 7 when $\lambda = 0.1$, the difference in RSS between $\ell_1$-LinR and $\ell_1$-LogR is reduced when $\lambda = 0.3$. In addition, by comparing Fig. 7 and Fig. 8, it can be seen that under the same setting, when $\lambda$ increases, the $Precision$ becomes larger while the $Recall$ becomes smaller, implying a tradeoff in choosing $\lambda$ in practice for Ising model selection with finite $M, N$.

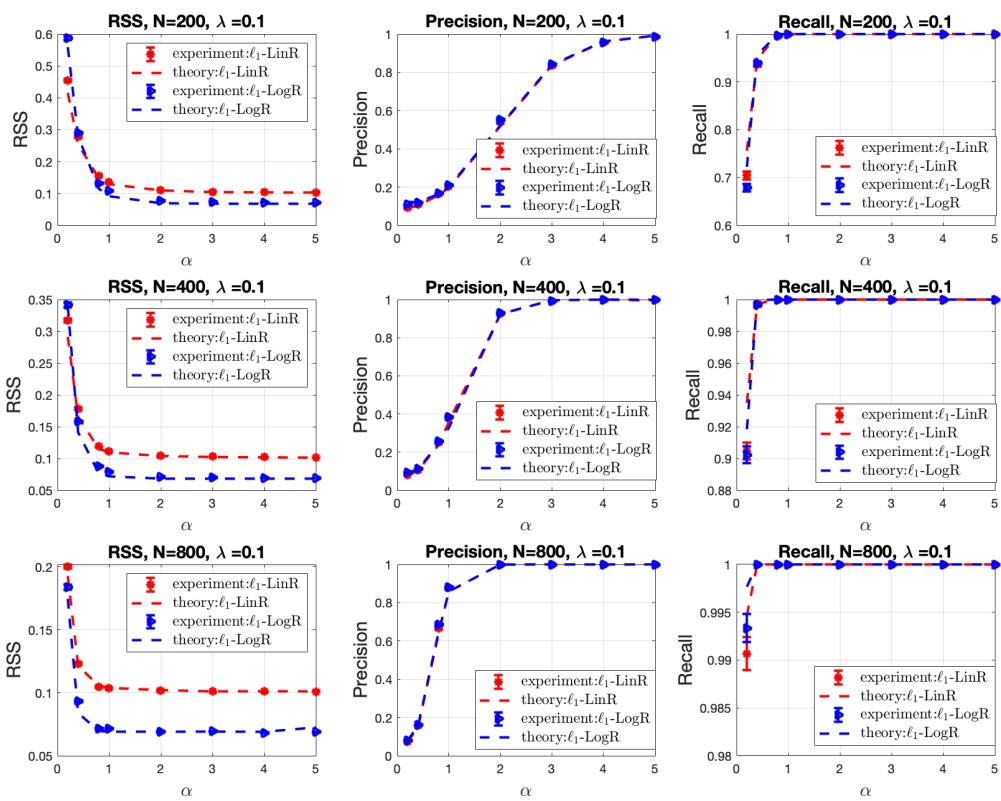

Figure 7: Theoretical and experimental results of $RSS$, $Precision$ and $Recall$ for both $\ell_1$-LinR and $\ell_1$-LogR when $\lambda = 0.1$, $N = 200, 400, 800$ with different values of $\alpha \equiv M/N$. The standard error bars are obtained from 1000 random runs. An excellent agreement between theory and experiment is achieved, even for small $N = 200$ and small $\alpha$ ( small $M$).

Fig. 9 and Fig. 10 show the full results of critical scaling prediction when $\lambda = 0.1$ and $\lambda = 0.3$, respectively. For comparison, both the results of $\ell_1$-LinR and $\ell_1$-LogR are shown. It can be seen that apart from the good agreements between replica results and experimental results, the prediction of the scaling value $c_0\left(\lambda, K_0\right) \equiv \frac{c(\lambda, K_0)}{\lambda^2}$ is very accurate.

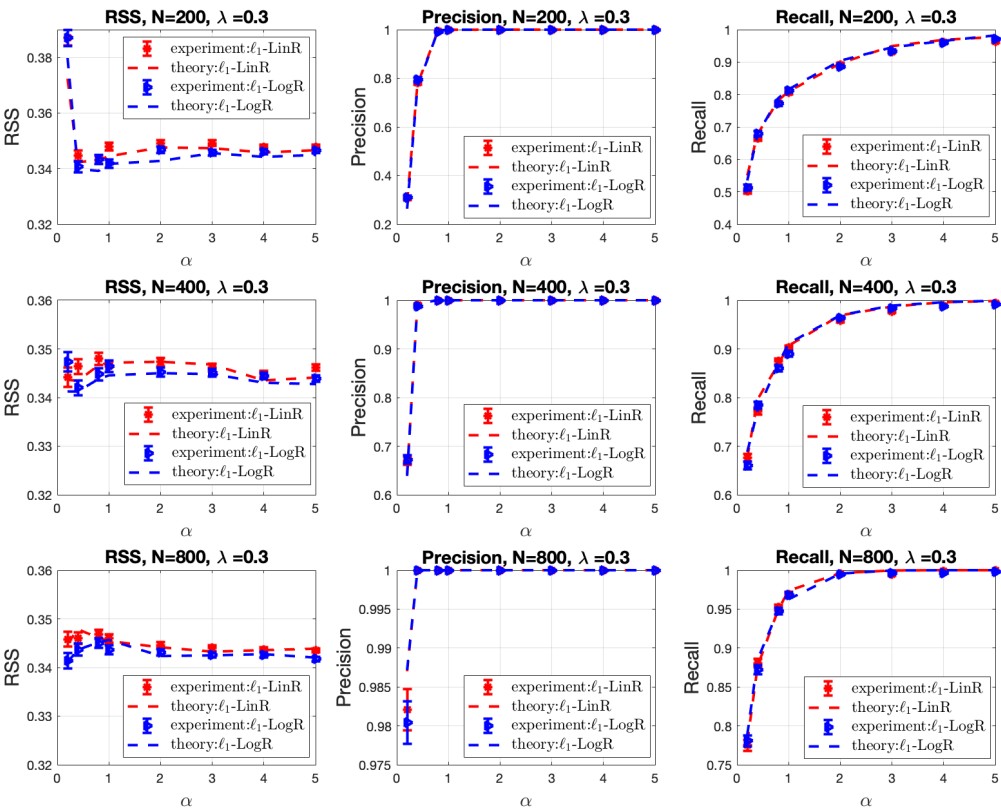

Figure 8: Theoretical and experimental results of RSS, *Precision* and *Recall* for both $\ell_1$-LinR and $\ell_1$-LogR when $\lambda = 0.3$, $N = 200, 400, 800$ with different values of $\alpha \equiv M/N$. The standard error bars are obtained from 1000 random runs. An excellent agreement between theory and experiment is achieved, even for small $N = 200$ and small $\alpha$ ( small $M$).

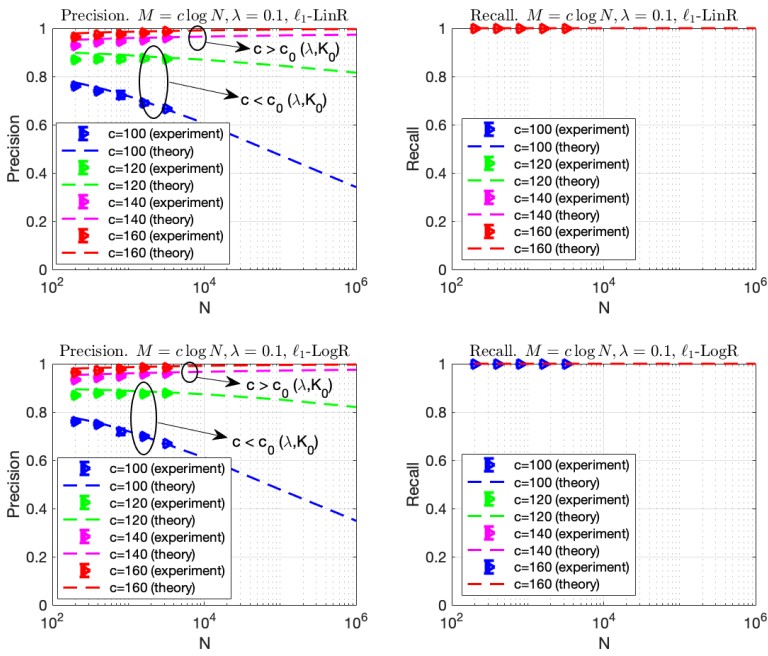

Figure 9: *Precision* and *Recall* versus $N$ when $M = c \log N$ and $K_0 = 0.4$ for $\ell_1$-LinR and $\ell_1$-LogR when $\lambda = 0.1$, where $c_0 (\lambda, K_0) \equiv \frac{c(\lambda, K_0)}{\lambda^2} \approx 137$. When $c > c_0 (\lambda, K_0)$, the *Precision* increases consistently with $N$ and approaches 1 as $N \to \infty$ while it decreases consistently with $N$ when $c < c_0 (\lambda, K_0)$.

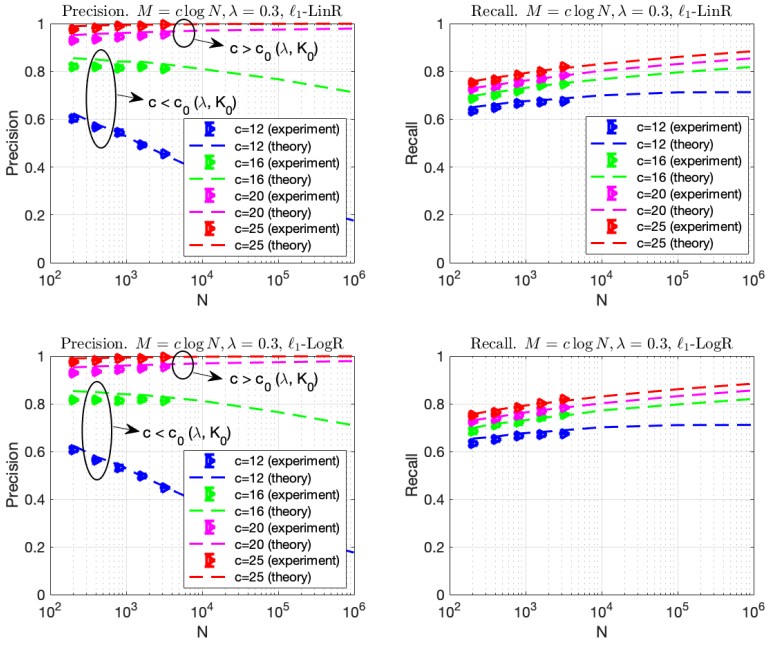

Figure 10: *Precision* and *Recall* versus $N$ when $M = c \log N$ and $K_0 = 0.4$ for $\ell_1$-LinR and $\ell_1$-LogR when $\lambda = 0.3$, where $c_0 (\lambda, K_0) \equiv \frac{c(\lambda, K_0)}{\lambda^2} \approx 19.4$. When $c > c_0 (\lambda, K_0)$, the *Precision* increases consistently with $N$ and approaches 1 as $N \to \infty$ while it decreases consistently with $N$ when $c < c_0 (\lambda, K_0)$. The *Recall* increases consistently and approach to 1 as $N \to \infty$.