# OpenReview forum: "Ising Model Selection Using $\ell_{1}$-Regularized Linear Regression: A Statistical Mechanics Analysis"
_NeurIPS.cc/2021/Conference — NeurIPS 2021 Poster_

### Official Review · Reviewer_hC3k · 2021-07-05

**Rating:** 5
**Confidence:** 3

**Summary:**

This paper analyses the behaviour of $\ell_{1}$ regularized linear regression for estimating Ising models, as compared to the canonical approach of $\ell_{1}$ regularized logistic regression. The authors use tools from statistical mechanics to arrive at their result, which is in the paramagnetic regime, and corroborate some of their claims using experiments.

**Limitations And Societal Impact:**

The authors describe the only limitation of the work being that the paramagnetic regime is too constrained. I agree with this assessment.

There are no direct societal impacts of this work.

**Main Review:**

### Originality
Using techniques from statistical mechanics to show results relevant in the statistical learning theory literature is a nice unification. Some of the claims in the work seem to be unsubstantiated as I describe in later comments.

### Quality
- I found it interesting that $c(\lambda, K_0)$ decreases with $d$, standard results have $M > max(d^{2}, \lambda^{-2}) \log(N)$ to estimate graph structure. Furthermore, the precision, recall can be easily computed once we've learned the graph, and it appears that the authors do the same from Algorithm 1 by estimating $J$ which is sufficient, so it is not clear where the benefit in their approach arises from. Non-asymptotic rates for logistic regression have been derived in Wu et al (2019) more recently.

### Clarity
I think the writing can be significantly improved to be more accessible to members of the theoretical machine learning community. A lot of the notation and techniques used are very esoteric, and specialized to the physics community (even remarked by the authors). As someone familiar with statistical learning theory literature more than statistical mechanics, my difficulty in reading the manuscript is reflected in my detailed comments below.

### Significance
While it is interesting to see such a result, it is unclear about the utility of the result w.r.t. predicting the recall and precision. Often times, we intend on finding the edges of the graph rather than the recall and precision alone, and estimating the graph gives us this automatically.

Additional comments below:

Section 1 and 2:
- IS is interaction screening, and not intersection screening as the authors refer multiple times in the introduction.
- The model misspecification appears to be deliberate, because if we know that the datapoints are binary valued, then a linear regression approach is probably too much.
- lasso vs. $\ell_{1}$ regularized logistic regression comparison: why is the former "computationally efficient" than the latter? Are there any computational complexity analyses to validate this statement. Practically, software like glmnet have bridged any gap between regularized linear and logistic regression.
- The authors address to certain things as "typical" in Section 1.1. What is "typical"?
- Line 40: assessing learning performance is not very novel. refer Wu et al (2019) for a comparison.
- Typo Line 92: IOS losses --> ISO losses

Section 3: for someone with a statistical learning theory background, a lot of the notation and use of techniques are alien, and my review of this section will reflect this inexperience.

- It would have been nice if the authors provided a primer for this. For instance, it is unclear to me what the exact purpose of the Hamiltonian and Boltzmann distribution derived in eqns. (6) and (7) are, and subsequent propositions about convergence to a point wise measure when the temperature tends to 0. Additionally the notation in line 143 defining the partition function is also conflicting because it is unclear if that definition and the one in eqn (1) are equivalent for Ising models.

- It appears that the self-averaging property is similar to the law of large numbers. However, it is not clear why a result like this would hold only when $N \to \infty$.

- The expectation under a dataset is defined in a strange way: are we considering multiple draws of M samples from the Ising model (as the summation indicates)? If so, a) how many draws?

- I feel that the replica method could have been briefly described, or at least the essence of it. Eqn (9) has undefined terminology as well: $Z^{n}$, derivatives of the log partition function w.r.t. n. The authors describe the RHS of the equation in the paragraph, but it is not immediately apparent why this is relevant. On further reading, it seems as though the authors have made use of this technique due to its prior use in disorder systems and in a generalisation analysis. The citation to the paper by Bresler on learning Ising models is irrelevant (there is no mention of "replica" in that paper).

- A similar issue appears with Eqn (10) i.e., undefined notation: $||J^{a}||\_{1}$, $J^{a}\_{j}$. It would have been useful to include a citation for this equation, because a derivation isn't available and $Z^{n}$ has not been defined prior.

- The Ansatz and subsequent corollaries are very hard to read due to excessive notation and terminology. Line 173 has $w_{i}$ where the index of summation is $j$. The purpose of $Q$ and $q$ are unclear, the introduction of $a$ and $b$ on Equation 12 is also unprecedented. Eqn (13) introduces $\texttt{Extr}$ which is previously undefined too. Eqn (14) -- (16) are unhelpful, because it is not clear what the purpose of those equations are. It is also unclear as to what the estimator is in this case: is it the L1-LinR estimator that is being referred to here?

- Line 189: why is this true? Is this a valid assumption? The Ansatz appears to be partly checked, does this mean that the claim is verified but not proven?

- Unfortunately, despite the authors' effort, I find the majority of page 6 inaccessible, as many of the definitions and concepts are unknown in the standard statistical learning theory. Are the EOS equations directly relevant to the results in this work? Additionally, why are the authors focusing on the temperature tending to 0 case for a result in the paramagnetic regime?

- Eqn (20) is derived using the Ansatz stated earlier and Eqn (17). However, Eqn (17) talks about the expression in the limit $\beta \to \infty$. Additionally, how expensive is solving the EOS, because they appear to be a set of highly nonlinear equations.

Section 4 contain experimental results, and here it would have been interesting to note the variation of their results with $d$, which the authors seem to have not considered.

**Time Spent Reviewing:**

5

---

> ### Author Response · Authors · 2021-08-10
> **Thank you for your review. Response to Reviewer hC3k (Part 1).**
>
> Thank you for your constructive comments and suggestions. Regarding your concerns, please find our responses below.
>
> $\mathbf{Q1}$: It is unclear about the utility of the result w.r.t. predicting the recall and precision. Oftentimes, we intend on finding the edges of the graph rather than the recall and precision alone, and estimating the graph gives us this automatically. The precision, recall can be easily computed once we've learned the graph, and it appears that the authors do the same from Algorithm 1, so it is not clear where the benefit in their approach arises from. Non-asymptotic rates for logistic regression have been derived in Wu et al (2019) more recently.  Assessing learning performance is not very novel. Refer Wu et al (2019) for a comparison.
>
>
>
> $\mathbf{A1}$: First, we emphasize that the precision and recall rates are significantly important metrics to characterize the performance of structure learning. Second, estimating the graph does not give us the recall and precision automatically, unless we already know the true graph structure, which is clearly not the case in practice. Third, what we do in Algorithm 1 is fundamentally different from that  "after” empirically learning a number of specific graphs. Algorithm 1 does not involve learning any specific graphs. Instead, we obtain an effective probabilistic model with only a few variables, a reduced $d$-dimensional $\ell_1$-LinR estimator (25) and scalar estimator (21), which is derived by using the presented analytical methods from the original model of a large number $N$ ($\to \infty$) of variables. From this effective model, one can efficiently predict the recall and precision of the original estimator of any $N$ (even $\to \infty$) without actually conducting time-consuming inferences on real datasets. What Algorithm 1 does is to solve the effective few-variable (actually $d$-variable, $d \ll N$) problem. In contrast, the paper by Wu et al (2019) derived the sample and time complexities for computing $\ell_{2,1}$ constrained logistic regression but they did not derive such an effective probabilistic model with a few variables, nor did they provide such precision/recall analysis. Hence, our work is much different from the one by Wu et al. Besides, although Wu et al (2019) presents an approximation of the estimator with time complexity $O(N^2)$, our method requires only $O(d^2)$ computational cost per update. In short, we theoretically derived a simple yet effective probabilistic model to describe the statistical properties of the original $\ell_1$ estimator, by which we can not only accurately predict the recall and precision, but also provide a sharp prediction of the number of samples required to achieve any predefined performance, as verified by the fairly good match in Sec. 4. We will add discussions in the revision.
>
> $\mathbf{Q2}$: $c(\lambda, K_0)$ decreases with $d$.
>
> $\mathbf{A2}$: It actually increases with $d$ which can be verified by the derivative w.r.t. $d$.
>
> $\mathbf{Q3}$: Writing can be improved to be more accessible to members of the theoretical machine learning community. A lot of the notation and techniques used are very esoteric and alien, and specialized to the physics community.
>
> $\mathbf{A3}$: Thank you for your patience in reading. We think the two communities should interact more and more due to the fact that, although the techniques used in those communities are very different, they share many similar goals in the research. Nevertheless, we will manage to revise the manuscript accordingly to make it as easy to understand as possible.
>
> $\mathbf{Q4}$: The model misspecification appears to be deliberate, because if we know that the data points are binary valued, then a linear regression approach is probably too much.
>
> $\mathbf{A4}$: We do not quite understand the meaning of "too much". Our results demonstrate that a linear regression still can achieve comparative results as logistic regression.
>
> $\mathbf{Q5}$: lasso vs. regularized logistic regression comparison: why is the former "computationally efficient" than the latter? Are there any computational complexity analyses to validate this statement? Practically, software like glmnet have bridged any gap between regularized linear and logistic regression.
>
> $\mathbf{A5}$: We agree that the computational complexity is the same order of polynomials of $N, M$ in the state-of-the-art algorithm. However, the linear model is much simpler than the logistic model and thus there are various situations where it requires smaller computational resources/time: an example is a sensitivity analysis with perturbation to a few variables. In this sense, we state that the linear model is more computationally efficient.
>
>
>
> $\mathbf{Q6}$: What is "typical" in Section 1.1.
>
> $\mathbf{A6}$: It is similar in spirit to the typical sequences in information theory. For $\forall{\epsilon}>0$, the typical sample $D^M$ is defined by the condition $f(D^M)-[f(D^M)]_{D^M} < \epsilon$. For $\forall{\epsilon}>0$, the fraction of typical samples tends to unity as $M,N\to \infty$.
>
> $\mathbf{Q7}$: What is the exact purpose of the Hamiltonian and Boltzmann distribution derived in eqns. (6) and (7) are, and subsequent propositions about convergence to a point-wise measure when the temperature tends to 0. Additionally, the notation in line 143 defining the partition function is also conflicting because it is unclear if that definition and the one in eqn (1) are equivalent for Ising models.
>
> $\mathbf{A7}$: The primal object to be computed is the cumulant generating function of the estimator $\hat{J}$. We think the reason for computing this would be clear also for the theoretical machine learning community. Unfortunately, this task is not easy and we need several techniques to compute it. Hamiltonian and Boltzmann distributions are introduced to enable the computation, and the partition function, the normalization constant of the Boltzmann distribution, is actually equivalent to the cumulant generating function if we take the log and take the zero-temperature limit. This is why we introduce and compute these quantities. The partition function in line 143 is different from the one in (1). In the revision, we will state these points more clearly.
>
>
> $\mathbf{Q8}$: It appears that the self-averaging property is similar to the law of large numbers. However, it is not clear why a result like this would hold only when $N\to \infty$.
>
> $\mathbf{A8}$: It is similar but different. In terms of free energy, self-averaging refers to the phenomena that, the average free energy per-dimension, $f=-\frac{1}{N\beta}\log Z$ of one specific Ising model of dimension N, approaches to the average of $f_{av}=-\frac{1}{N\beta}\[\log Z]$ when $N \to \infty$. Intuitively, one system of large dimension $N$ can be divided into a large number of subsystems, each one showing some macroscopic properties with randomness. As the dimension $N \to \infty$, the number of subsystems also goes to $\infty$, and thus the macroscopic properties of the whole system will behave like the average of a large number of subsystems, thus the name self-averaging property. For more details, please refer to reference [17].
>
> $\mathbf{Q9}$: The expectation under a dataset is defined in a strange way: are we considering multiple draws of $M$ samples from the Ising model (as the summation indicates)? If so, how many draws?
>
> $\mathbf{A9}$: We think this expectation over the random dataset is clear and well defined. For i.i.d. samples, the distribution of one dataset with $M$ samples can be written as a product of the distribution for each sample. Then, by definition of expectation, we need to perform a summation over the enumeration of all possible values of each vector $\boldsymbol{s}^{\mu}$ of dimension $N$.  Since each spin can only take $-1$ or $+1$, $\boldsymbol{s}^{\mu}$ can take $2^N$ possible values.
>
> $\mathbf{Q10}$: I feel that the replica method could have been briefly described, or at least the essence of it. Eqn (9) has undefined terminology as well: $Z^n$, derivatives of the log partition function w.r.t. n. The authors describe the RHS of the equation in the paragraph, but it is not immediately apparent why this is relevant. On further reading, it seems as though the authors have made use of this technique due to its prior use in disorder systems and in a generalization analysis. The citation to the paper by Bresler on learning Ising models is irrelevant.
>
> $\mathbf{A10}$: We think the equation itself is defined unambiguously, but agree that the meaning is not apparent for people unfamiliar with this. As explained above, $\log{Z}$ is the cumulant generating function and it is dominated by the exponential rate of $Z, f$, in the large $N$ limit. This rate is called free energy density in physics. Then, we want to compute the average of $\log{Z}$ over the dataset, but it is not easy to take the average analytically. For circumventing this, we use the identity that $[\log{Z}]$ equals to the partial derivative of $\log{[Z^n]}$ in the $n \to 0$ limit (this is just a simple exercise: please use the identity $[Z^n]=[e^{n\log Z}]$ and expand it w.r.t. $n$.). Taking the average of both sides yields Eq. (9). The computations after this are full of technicalities and hard to summarize in a few words. For the details of computations, we referred to [17-19] instead of the paper by Bresler (thank you for pointing out this typo. We will fix it), and also stated the details in the appendix. In the revision, we will describe these points as concise and understandable as possible.

---

> > ### Author Response · Authors · 2021-08-10
> > **Thank you for your review. Response to Reviewer hC3k (Part 2).**
> >
> > $\mathbf{Q11}$: Eqn (10) i.e., undefined notation: $\left\Vert J^a \right\Vert$. It would have been useful to include a citation for this equation because a derivation isn't available and $Z^n$ has not been defined prior.
> >
> > $\mathbf{A11}$: $\left\Vert J^a \right\Vert_1$ is the $\ell_1$ norm of the integration variables $J^a$. $Z^n$ is simply the $n$ th power of $Z$ and can be written as $\prod_{\text{a=1}}^{n}Z_{a}$, where $Z_a=\int d\boldsymbol{J}^ae^{-\beta\mathcal{H}\left(\boldsymbol{J}^a|\mathcal{D}^{M}\right)}$, where  $\boldsymbol{J}^a, a=1,...,n$ can be viewed as different replicas of $\boldsymbol{J}$. Then, given the definition of expectation over the dataset, Eqn (10) can be readily obtained.  We will clarify these in the revision.
> >
> >
> > $\mathbf{Q12}$: The Ansatz and subsequent corollaries are very hard to read due to excessive notation and terminology. Line 173 has $w_i$ where the index of summation is $j$. The purpose of $Q$ and $q$ are unclear, the introduction of $a$ and $b$ on Equation 12 is also unprecedented.  Eqn (13) introduces $\mathbf{\mathtt{Extr}}$ which is undefined. Eqn (14) -- (16) are unhelpful because it is not clear what the purpose of those equations is. It is also unclear as to what the estimator is in this case: is it the L1-LinR estimator that is being referred to here?
> >
> > $\mathbf{A12}$: First of all, we will fix the typo about $w_i$ and clarify the definition of $\mathbf{\mathtt{Extr}}$ (we have actually defined it in line 182-183). Thanks for pointing them out.   The reason why $Q$ and $q$ are introduced is that the integral result just depends on these quantities. a, b are just the dummy variables of the replica index running on the sum $\sum_{a=1}^{n}$.  Eqn (14) -- (16) are quite essential and useful since they represent the computation of free energy, which is the core of our goal. We believe that once the basic idea of the replica method is understood, the importance of (14)-(16) will become natural and straightforward. Yes, Eqn (14) -- (16) corresponds to the $\ell_1$-LinR estimator and we will clarify this in the revision.
> >
> > $\mathbf{Q13}$: Line 189: why is this true? Is this a valid assumption? The Ansatz appears to be partly checked, does this mean that the claim is verified but not proven?
> >
> > $\mathbf{A13}$: This is in a similar spirit to the self-averaging property as A8. Yes, it is not rigorously proven but its consistency is numerically checked.
> >
> > $\mathbf{Q14}$: Unfortunately, despite the authors' effort, I find the majority of page 6 inaccessible, as many of the definitions and concepts are unknown in the standard statistical learning theory. Are the EOS equations directly relevant to the results in this work? Additionally, why are the authors focusing on the temperature tending to 0 case for a result in the paramagnetic regime?
> >
> > $\mathbf{A14}$: The EOS is quite relevant and essential in this work. This object is derived using the saddle-point (or Laplace) method applied for computing the integral appearing in the free energy, and the EOS is nothing but the saddle-point condition. This saddle-point condition is characterized by a few variables such as $Q, q, \bar{J}_i, \forall i\in \Psi$ thanks to the original problem’s symmetry. From the solution of the EOS, we can derive a number of nontrivial predictions about the performance of the estimators such as Precision and Recall. Although solving the EOS is still nontrivial (but not difficult) since it consists of nontrivial integral equations, the reduction from the original problem with $N$ variables to the EOS with a few variables is significant. The derivation of the EOS is detailed in Appendix, and for further information please refer to [17-19].
> >
> > The reason why we focus on zero temperature ($\beta \to \infty$) is that in this case the Boltzmann distribution (7) converges to a point-wise measure on the estimator (2) so that we can analyze (2) by evaluating (7). Note that this (inverse) temperature in (7) is different from that of the Ising model (1) from which the samples are drawn. We will clarify this point in the revision.
> >
> > $\mathbf{Q15}$: Eqn (20) is derived using the Ansatz stated earlier and Eqn (17). However, Eqn (17) talks about the expression in the limit $\beta \to \infty$. Additionally, how expensive is solving the EOS, because they appear to be a set of highly nonlinear equations.
> >
> > $\mathbf{A15}$: The limit $\beta \to \infty$ is already taken. Although highly nontrivial,  the EOS can be efficiently solved using numerical methods in about several minutes, or even tens of seconds in most cases.
> >
> > $\mathbf{Q16}$: Section 4 contains experimental results, and here it would have been interesting to note the variation of their results with $d$, which the authors seem to have not considered.
> >
> > $\mathbf{A16}$: Thank you for the suggestion. However, we believe current experiments with $d=3,4$ suffice to support our theoretical analysis.
> >
> > $\mathbf{Q17}$: Remaining minor comments and typos: e.g., Line 92: IOS losses --> ISO losses, intersection screening  --> interaction screening
> >
> >
> > $\mathbf{A17}$: Thank you for pointing out them. We will fix them all in the revision.

---

> > > ### Comment · Reviewer_hC3k · 2021-08-21
> > > **Thank you for your rebuttal (part 2)**
> > >
> > > > The EOS is quite relevant and essential in this work.
> > >
> > > Thank you for making this clear, but I still find it extremely difficult to parse and understand the essence of these equations without prior knowledge of them. If there is a more concise way of stating this, I would suggest that the authors consider it, and instead provide some background for the EOS, and some insight into why the EOS works the way it does / include references and a minor discussion.
> > >
> > > > However, we believe current experiments with $d = 3, 4$ suffice to support our theoretical analysis.
> > >
> > > I feel that an increase in $d$ would be interesting to the reader. If it not too much trouble, I would like to see if there is a discernible trend with increase in $d$. In addition to this, are the results statistically significant (as in, have to run this on multiple graphs with the same values of $d, M$?

---

> > > > ### Author Response · Authors · 2021-08-24
> > > > **Response to further comments of reviewer hC3k (Part 2)**
> > > >
> > > > $\mathbf{Q7}$: Thank you for making this clear, but I still find it extremely difficult to parse and understand the essence of these equations without prior knowledge of them. If there is a more concise way of stating this, I would suggest that the authors consider it, and instead provide some background for the EOS, and some insight into why the EOS works the way it does/include references and a minor discussion.
> > > >
> > > > $\mathbf{A7}$: We will do our best to explain the meaning of the EOS as clearly as possible in the revision. Also, for further details please refer to [R7] and [17,18,19] as explained in the last response A14.
> > > >
> > > > $\mathbf{Q8}$: I feel that an increase in $d$ would be interesting to the reader. If it not too much trouble, I would like to see if there is a discernible trend with increase in $d$. In addition to this, are the results statistically significant (as in, have to run this on multiple graphs with the same values of $d, M$?)
> > > >
> > > > $\mathbf{A8}$: The purpose of the present paper is to propose a quantitative theory being accurate for large values of $N$, and hence all the experiments in the manuscript are conducted very carefully to check the consistency with the theoretical prediction. These careful experiments can be a heavy burden and hence it is difficult to conduct additional experiments with changing $d$ in this limited time. We would like to appreciate your understanding of this.

---

> > > > > ### Comment · Reviewer_hC3k · 2021-08-24
> > > > > **Thank you for the response - part 2**
> > > > >
> > > > > > A8: These careful experiments can be a heavy burden and hence it is difficult to conduct additional experiments with changing $d$ in this limited time.
> > > > >
> > > > > I thought that running these experiments would have been feasible, since you claim the method is computationally efficient, but given the short period of time, I believe it's ok since the focus of the paper is more theoretical.

---

> > ### Comment · Reviewer_hC3k · 2021-08-21
> > **Thank you for your rebuttal (part 1)**
> >
> > > First, we emphasize that the precision and recall rates are significantly important metrics to characterize the performance of structure learning. Second, estimating the graph does not give us the recall and precision automatically, unless we already know the true graph structure, which is clearly not the case in practice.
> >
> > Thank you for elucidating this, I think I understand the contributions of the paper well now, and this is something that I missed in my preliminary reading of the work. However, my followup is: is there any intuition for why the $\ell\_{1}-$ regularized logistic regression estimator not do this. Why does it have to be $\ell\_{1}$-regularized linear regression.
> >
> > > Third, what we do in Algorithm 1 is fundamentally different from that "after” empirically learning a number of specific graphs. Algorithm 1 does not involve learning any specific graphs. Instead, we obtain an effective probabilistic model with only a few variables, a reduced -dimensional $\ell\_{1}$-LinR estimator (25) and scalar estimator (21), which is derived by using the presented analytical methods from the original model of a large number ($N \to \infty$) of variables. From this effective model, one can efficiently predict the recall and precision of the original estimator of any  (even $\to \infty$) without actually conducting time-consuming inferences on real datasets.
> >
> > While this has a lot of mathematical significance, I find the justification of the method to real scenarios a little confusing: why would you want to evaluate the precision and recall of the method alone? Say you do get a good precision and recall, you'll have to estimate the parameters or the graph structure anyways, thus incurring the computation costs that you have stated.
> >
> > >  In short, we theoretically derived a simple yet effective probabilistic model to describe the statistical properties of the original
> >  estimator, by which we can not only accurately predict the recall and precision, but also provide a sharp prediction of the number of samples required to achieve any predefined performance, as verified by the fairly good match in Sec. 4. We will add discussions in the revision.
> >
> > Thank you for this brief note. In most scenarios, the data is given to us in an offline fashion, in which case the utility of the prediction is the quality of the graph that _could_ be learned.
> >
> > >  It actually increases with $d$ which can be verified by the derivative w.r.t. $d$.
> >
> > My calculations give me the derivative of $c(\lambda, K\_{0})$ w.r.t. $d$ to be $\frac{2(\lambda^{2} - \tanh^{2}(K\_{0}))(1 - \tanh^{2}(K\_{0}))}{(1 - \tanh^{2}(K\_{0}) + d\tanh^{2}(K\_{0}))^{2}}$. In your case, $\lambda < \tanh(K\_{0})$, so this is $< 0$.
> >
> > This shown the $c(\lambda, K_{0})$ decreases with $d$, and consequently denser graphs are easier to learn?
> >
> > > Thank you for your patience in reading. We think the two communities should interact more and more due to the fact that, although the techniques used in those communities are very different, they share many similar goals in the research. Nevertheless, we will manage to revise the manuscript accordingly to make it as easy to understand as possible.
> >
> > Thank you for sharing this perspective, and I completely agree with the fact that both communities should engage and share techniques. Nonetheless, given that this is a more machine learning oriented conference, I feel that authors from other communities who are willing to engage and share could present their material in a more accessible manner.
> >
> > > We do not quite understand the meaning of "too much". Our results demonstrate that a linear regression still can achieve comparative results as logistic regression.
> >
> > I'm sorry for the imprecise comment, I merely felt that when the data is known to be binary, a linear regression approach which usually maps inputs to real valued outputs, often with no outcome constraints was a bit strange. My comment was directed more towards intuition instead of the results that the authors have presented.
> >
> > > We agree that the computational complexity is the same order of polynomials of  in the state-of-the-art algorithm. However, the linear model is much simpler than the logistic model and thus there are various situations where it requires smaller computational resources/time
> >
> > This seems to be a very subjective comment with regard to simplicity. I would instead argue (as above) that when the outputs are binary valued, something that immediately pops to mind is a logistic regression model. However, there is some credence to the author's arguments in that logistic regression doesn't have a closed form solution, but you could say the same about linear regression with $\ell\_{1}$ regularization.
> >
> > > It is similar in spirit to the typical sequences in information theory.
> >
> > This is not evident from the context, because the authors also say "typical RR graph". I think it would be better if the author's included a brief note of what they mean by typical and chose alternative language while describing other objects (e.g. standard RR graph).
> >
> > ----
> >
> > Thank you for your replies acknowledging the areas of improvement.

---

> > > ### Author Response · Authors · 2021-08-24
> > > **Response to further comments of reviewer hC3k (Part 1)**
> > >
> > > Thank you for the further comments. Regarding your further concerns, please find our responses below.
> > >
> > > $\mathbf{Q1}$: However, my followup is: is there any intuition for why the $\ell_1$ regularized logistic regression estimator not do this. Why does it have to be $\ell_1$-regularized linear regression.
> > >
> > > $\mathbf{A1}$: Sorry we don't quite understand your followup. Do you mean why precision and recall are considered here for $\ell_1$-LinR but not for $\ell_1$-LogR in previous studies?  If so, then it has nothing to do with the estimators (note that we have done this for both estimators). In fact, for any estimator, to characterize its structure learning performance, one must consider two types of errors, namely type I error and type II error [R1].  The type I error is false positives (FP), i.e., falsely including variables into the true neighborhood. The type II error is false negatives (FN), i.e., falsely excluding variables out of the true neighborhood. Structure recovery is successful if and only if neither of them occurs, which is equivalent to both precision and recall being one [R1].  Here we use precision and recall since they are common metrics for structure recovery and are particularly useful for imbalanced data [R2]. In [R3], the precision and recall of $\ell_1$-LogR are also (empirically) evaluated. In [R4], the two types of errors of $\ell_1$-LogR are considered in a direct manner, which is for ease of proof. Specifically, in theorem 1 (b) of [R4], the first part “correctly excludes all edges not in the neighborhood” refers to the absence of type I error while the second part “correctly includes all edges” refers to the absence of type II error. In Corollary 1 of [R5], the setting of the threshold $\epsilon<\eta/2$ also ensures the simultaneous absence of both errors.
> > >
> > > So, it is not why $\ell_1$-LinR has to consider this while $\ell_1$-LogR not; rather, all estimators, though possibly in different ways, should consider the two types of errors when evaluating the structure learning performance.
> > >
> > > $\mathbf{Q2}$: While this has a lot of mathematical significance, I find the justification of the method to real scenarios a little confusing: why would you want to evaluate the precision and recall of the method alone? Say you do get a good precision and recall, you'll have to estimate the parameters or the graph structure anyways, thus incurring the computation costs that you have stated.
> > >
> > > $\mathbf{A2}$: To avoid possible confusion, please let us stress the difference between one learning algorithm and theoretical analysis of its learning performance. The goal of  our method in Algorithm 1 is not providing a low-complexity algorithm to obtain estimates for one specific graph, but rather obtaining the typical (please refer to A6 for “typical”) learning performance (in terms of precision and recall) of $\ell_1$-LinR for a typical set of random graphs, which is one fundamental problem in the theoretical analysis of any learning algorithm [R7]. Such theoretical analysis is not only of mathematical interest but also of great practical significance. As explained in A1 in our first rebuttal, in real scenarios one cannot assess the estimated graphs by comparing them with the (unknown) true graphs. Instead, our theoretical analysis does not tell us “what” the estimates are, but rather “how good” (in a statistical average manner) these estimates are. The excellent agreement with experimental results verifies the accuracy of our theoretical analysis. This is in a similar spirit to analyzing the (generalization) accuracy for image classification.
> > >
> > > $\mathbf{Q3}$: My calculations give me the derivative of $c(\lambda, K_0)$  w.r.t. $d$ to be $\frac{2(\lambda^2-\tanh^2{K_0})(1-\tanh^2{K_0})}{1-\tanh^2{K_0}+d\tanh^2{K_0}}$. In your case, $\lambda < \tanh{K_0}$, so this is $<0$. This shown the $c(\lambda, K_0)$ decreases with $d$, and consequently denser graphs are easier to learn?
> > >
> > > $\mathbf{A3}$: Yes, your calculation of the derivative is right and sorry for our previous incorrect calculation. However, one needs to be careful about the statement that denser graphs are easier to learn. Recall that $c(\lambda, K_0)$ is defined for a fixed $\lambda$ where $\lambda < \tanh{K_0}$ and thus a lower bound can be obtained as $M>\frac{2\log N}{\tanh^{2}\left(K_{0}\right)}$ when $\lambda \to \tanh{K_0}$. Moreover, the paramagnetic condition enforces $d-1 < 1/\tanh^2(K_0)$ so that  it is necessary to reduce the magnitude of $K_0$ to $\mathcal{O}(1/{\sqrt{d}})$ for avoiding the phase transition. This means that the lower bound grows as $M>\frac{2\log N}{\tanh^{2}\left(K_{0}\right)} > \mathcal{O}(d \log N)$ for large $d$.
> > >
> > > $\mathbf{Q4}$: I completely agree with the fact that both communities should engage and share techniques. Nonetheless, given that this is a more machine learning oriented conference, I feel that authors from other communities who are willing to engage and share could present their material in a more accessible manner.
> > >
> > > $\mathbf{A4}$: We agree that the material needs to be  presented in an accessible manner and will try our best to make our presentations accessible in the revision.
> > >
> > > $\mathbf{Q5}$: I'm sorry for the imprecise comment, I merely felt that when the data is known to be binary, a linear regression approach which usually maps inputs to real valued outputs, often with no outcome constraints was a bit strange. My comment was directed more towards intuition instead of the results that the authors have presented.
> > >
> > > This seems to be a very subjective comment with regard to simplicity. I would instead argue (as above) that when the outputs are binary valued, something that immediately pops to mind is a logistic regression model. However, there is some credence to the author's arguments in that logistic regression doesn't have a closed form solution, but you could say the same about linear regression with $\ell_1$ regularization.
> > >
> > > $\mathbf{A5}$: We reply together since both comments concern the motivations of using linear regression for binary data.
> > >
> > > First, the idea of using linear regression for binary data is not as outrageous as one might imagine. To the best of our knowledge, this problem was first studied by David R Brillinger [27], who demonstrated that linear regression provides useful estimates for (even unknown) nonlinear transforms. Please refer to Sec. 1.2 of our manuscript for an introduction of related works on this topic. Moreover, perhaps surprisingly, linear regression can even outperform logistic regression for binary valued data in some cases. For example, a recent study [R6] demonstrated that linear regression is generally the best strategy (more effective than logistic regression) to estimate the causal effects of treatments on binary valued data. These counterintuitive results are really interesting and we are curious if it still works for Ising model selection. The current affirmative answer under the paramagnetic condition is inspiring. Please refer to A1 to reviewer Y3wG for an intuitive explanation.
> > >
> > > Second, it is a representative example of model mismatch. One might argue that such mismatch is kind of deliberate as it is “natural” to think of using logistic regression (possibly not model match either) for binary valued data, but what if the data is still real valued but going through an unknown nonlinear operation? In practice model mismatch is inevitable and thus it is interesting to study the learning behaviour under model mismatch. Our study shows that at least model match is not essential for some specific learning tasks.
> > >
> > > In short, the idea of using linear regression for binary valued data is by no means naive and its success for some specific settings is highly nontrivial and of scientific significance.
> > >
> > > $\mathbf{Q6}$: “It is similar in spirit to the typical sequences in information theory”. This is not evident from the context, because the authors also say "typical RR graph". I think it would be better if the author's included a brief note of what they mean by typical and chose alternative language while describing other objects (e.g. standard RR graph).
> > >
> > > $\mathbf{A6}$: Please refer to A7 to reviewer 8TQ3 for our illustration of "typical RR graph". Generally speaking, “typical” is one basic concept in statistical mechanics and information theory, which means not just most probable but in addition the probability for situations different from the typical one can be made arbitrarily small [R7,R8]. Thanks for your suggestion. We will clarify this in our revision.
> > >
> > > References:
> > >
> > > [R1] https://en.wikipedia.org/wiki/Precision_and_recall
> > >
> > > [R2] Saito, Takaya, and Marc Rehmsmeier. "The precision-recall plot is more informative than the ROC plot when evaluating binary classifiers on imbalanced datasets." PloS one 10.3 (2015): e0118432.
> > >
> > > [R3] Wainwright, M. J., Ravikumar, P., & Lafferty, J. D. (2007). High-dimensional graphical model selection using l~ 1-regularized logistic regression. Advances in neural information processing systems, 19, 1465.
> > >
> > > [R4] Ravikumar, P., Wainwright, M. J., & Lafferty, J. D. (2010). High-dimensional Ising model selection using ℓ1-regularized logistic regression. The Annals of Statistics, 38(3), 1287-1319.
> > >
> > > [R5] Wu, Shanshan, Sujay Sanghavi, and Alexandros G. Dimakis. "Sparse logistic regression learns all discrete pairwise graphical models." Advances in Neural Information Processing Systems 32 (2019): 8071-8081.
> > >
> > > [R6] Gomila, R. (2021). Logistic or linear? Estimating causal effects of experimental treatments on binary outcomes using regression analysis. Journal of Experimental Psychology: General, 150(4), 700–709.
> > >
> > > [R7] Engel, Andreas, and Christian Van den Broeck. Statistical mechanics of learning. Cambridge University Press, 2001.
> > >
> > > [R8] Cover, Thomas M. Elements of information theory. John Wiley & Sons, 1999.

---

> > > > ### Comment · Reviewer_hC3k · 2021-08-24
> > > > **Thank you for the response**
> > > >
> > > > Thank you for your detailed response to my followup questions.
> > > >
> > > > > A1: Sorry we don't quite understand your followup.
> > > >
> > > > Sorry about any confusion this may have caused. To quote the abstract:
> > > > > Moreover, we provide a computationally efficient method to accurately predict the non-asymptotic behavior of $\ell\_{1}$-LinR for moderate M and N, such as the precision and recall rates.
> > > >
> > > > and your response in A2:
> > > > > Instead, our theoretical analysis does not tell us “what” the estimates are, but rather “how good” (in a statistical average manner) these estimates are.
> > > >
> > > > In other words, my question was: what was the purpose of using the $\ell\_{1}$- regularized linear regression estimator, rather than the $\ell\_{1}-$ regularized logistic regression estimator, and do you have any intuition for why such methods don't exist to predict non-asymptotic behavior of $\ell\_{1}$-regularized logistic regression estimator, such as the precision and recall rates (basically saying "how good")?
> > > >
> > > > This is not a question about which estimator to use in practice, or about any metrics, but instead, more about the intuition behind selecting the $\ell\_{1}$-regularized linear regression estimator. Is there something special that the $\ell\_{1}$-regularized linear regression provide in terms of capability that allows predicting both the recall and precision without knowing the true graph (which you claim is a saliency of this work).
> > > >
> > > > -----
> > > >
> > > > Thank you for the responses to the other comments, I have no further questions regarding them.

---

> > > > > ### Author Response · Authors · 2021-08-25
> > > > > **Response to additional question of reviewer hC3k**
> > > > >
> > > > > $\mathbf{Q1}$: In other words, my question was: what was the purpose of using the $\ell_1$- regularized linear regression estimator, rather than the $\ell_1$-regularized logistic regression estimator, and do you have any intuition for why such methods don't exist to predict non-asymptotic behavior of $\ell_1$-regularized logistic regression estimator, such as the precision and recall rates (basically saying "how good")?
> > > > > This is not a question about which estimator to use in practice, or about any metrics, but instead, more about the intuition behind selecting the $\ell_1$ -regularized linear regression estimator. Is there something special that the $\ell_1$-regularized linear regression provides in terms of capability that allows predicting both the recall and precision without knowing the true graph (which you claim is a saliency of this work)?
> > > > >
> > > > > $\mathbf{A1}$: Thank you for clarifying your question. It seems that there is still some misunderstanding and you also miss our results for the $\ell_1$-regularized logistic regression estimator, which are already included in both section 4 and supplementary materials.
> > > > >
> > > > > Actually, as already stated in our third contribution in section 1.1, our method can be used to predict the non-asymptotic performances of a wide class of $\ell_{1}$-regularized estimators, including the $\ell_1$-regularized logistic regression estimator. In our supplementary material (appendix A, D, E, G), apart from the $\ell_1$- regularized linear regression estimator, we have also provided a detailed parallel analysis for the $\ell_1$ -regularized logistic regression estimator using our method. In particular, for the non-asymptotic performance evaluation, please refer to Appendix E, especially Algorithm 3, which is a counterpart of Algorithm 2 for the $\ell_1$-regularized linear regression estimator.  Moreover, for the $\ell_1$-regularized logistic regression estimator, we have also compared our theoretical results (Algorithm 2) of precision and recall with experimental ones.  For details, please refer to figure 1 (blue lines) in section 4 of the manuscript, figure 6 (blue lines), figure 7 (blue lines), figure 8 (lower part), and figure 9 (lower part) in Appendix G. It can be seen that there is also an excellent agreement for the $\ell_1$ -regularized logistic regression estimator.
> > > > >
> > > > > Therefore, our method is not restricted to $\ell_1$-regularized linear regression but applies to a wide class of $\ell_1$-regularized estimators. Regarding the motivations/intuitions of our focus on $\ell_1$-regularized linear regression, please refer to A5 in the second rebuttal.

---

> > > > > > ### Comment · Reviewer_hC3k · 2021-08-25
> > > > > > **Thank you for this clarification.**
> > > > > >
> > > > > > Thank you for the detailed clarification about logistic regression vs. linear regression. It definitely makes sense to me now. From the original title of the work, it was my impression that linear regression was preferred, but for some underlying reason, which is not exactly the case as you have pointed out (to the Appendix).
> > > > > >
> > > > > > I will be preserving my rating due to the relative inaccessibility of the paper (lack of sufficient exposition), but I do feel that the results are very impressive.

---

### Official Review · Reviewer_gune · 2021-07-17

**Rating:** 7
**Confidence:** 3

**Summary:**

This paper investigates the L1-regularized linear regression (L1-LinR) for learning Ising models. The authors apply the replica methods from statistical physics and obtain meaningful learning performance results for random regular graphs in the paramagnetic phase.

**Limitations And Societal Impact:**

It would be helpful to include some discussions about why the L1-LinR estimators perform well (only) in the paramagnetic phase. How good is L1-LinR beyond the paramagnetic phase? Would it become inaccurate since the replica methods fail? Some intuition from statistical physics is sufficient.

**Main Review:**

The Ising model is one of the most popular statistical models that are used in various fields of computer science. It belongs to the class of undirected graphical models (also called Markov random fields) for modeling high dimensional binary data. This paper studies the problem of Ising model selection, which tries to learn the edge set of the Ising model from iid samples. The authors focus on the case of random regular graphs in the paramagnetic phase. As a main result, the authors propose the L1-regularized linear regression (L1-LinR) and show that it has an optimal logarithm sample complexity for typical random regular graphs. This estimator is simpler and computationally more efficient compared with other estimators like L1-LogR and IS. It is also representative for model misspecification. The authors provide detailed analysis for the learning performance of L1-LinR in the non-asymptotic case with strong experimental supports. The methods in this paper are potentially applicable to other estimators and models.

**Time Spent Reviewing:**

1

---

> ### Author Response · Authors · 2021-08-10
> **Thank you for your review. Response to Reviewer gune.**
>
> Thank you for your positive comments and constructive suggestions. Regarding your concerns, please find our responses below.
>
> $\mathbf{Q1}$: It would be helpful to include some discussions about why the L1-LinR estimators perform well (only) in the paramagnetic phase. How good is L1-LinR beyond the paramagnetic phase? Would it become inaccurate since the replica methods fail? Some intuition from statistical physics is sufficient.
>
> $\mathbf{A1}$: Thank you for the suggestions. We first emphasize that the success of the quadratic loss is highly non-trivial even in the paramagnetic phase. In the paramagnetic phase of tree-like models, this can be intuitively understood as follows. On average, the condition for the $\ell_1$-LinR estimator is given as
>
> $<s_0s_k> - \sum_{j \ne 0} <s_j s_k> J_j  = \lambda \partial |J_k|$, ($k = 1,\ldots, N$)
>
> where $<\cdot>$ and $\partial |J_k|$ represent average with respect to the Boltzmann distribution (7) and the sub-gradient of $|J_k|$, respectively. In the paramagnetic phase, $<s_is_j>$ decays exponentially with respect to the distance $d_{ij}$ of sites $i$ and $j$ as $\tanh^{d_{ij}}(K_0)$ in its magnitude. This guarantees that once the connections $J_k$ of sites in the first nearest neighbor set $\Psi$ are given so that
>
> $<s_0s_k> = \sum_{j \in \Psi} <s_j s_k> J_j + \lambda \rm{sign} (J_k)  (\forall{k} \in \Psi)$  (*)
>
> holds, the other conditions are automatically satisfied by setting all the other connections that are not from $\Psi$ to zero.  For appropriate choices of $\lambda$, (*) has solutions of  $\rm{sign}(J_j^0)J_j>0, \forall{k} \in \Psi$, i.e., $\forall{k} \in \Psi$, the estimate $(J_k)$ has the same sign as the true value $J_k^0$. These conclude that on average the $\ell_1$-LinR estimator successfully recovers the network structure up to the connection signs if $\lambda$ is chosen appropriately. Although this argument is based on the properties of tree-like models, the results of experiments on 2D square lattices shown in Fig. 1 indicate that it also holds, at least in a good approximation level, even if graphs include cycles.
>
> The key of the above argument is that $<s_is_j>$ decays exponentially with respect to the distance of two sites, which does not hold after the phase transition. Actually, we have a preliminary result that $\ell_1$-LinR starts to fail in the network recovery just at the phase transition point. This is not limited to $\ell_1$-LinR; the matched estimator $\ell_1$-LogR also exhibits similar behavior unless post-thresholding is used, as reported in Montanari et al [R1]. However, we would like to leave its details for another occasion as the analysis is rather complicated and is difficult to include in the current paper due to the space limitation.
>
> We will happilly add explanations and discussions in the revision.
>
> [R1] Montanari, A.; and Pereira, J. A. Which graphical models are difficult to learn? In Advances in Neural Information Processing Systems, 1303–1311, 2009.

---

### Official Review · Reviewer_QMbt · 2021-07-17

**Rating:** 7
**Confidence:** 4

**Summary:**

The manuscript contains a theoretical investigation of the Ising model selection problem.
The task is to recover the support (the graph topology) of the interaction network from observations of Ising configurations.

An objective function in the M-estimator family, named L1-LinR by the authors, is analyzed.
It contains the typical quadratic data-dependent loss from linear regression and an L1 regularization term,
therefore it is a mismatched objective compared to L1-LogR (pseudolikelihood) and the Interaction Screening.


**Limitations And Societal Impact:**

A major critic is that L1-LinR objective has little relevance in
the Ising model selection context, where practitioners use
matched M-estimators.
Nonetheless, given the quality of the results obtained and the complexity
of these kinds of analysis, I think the paper would represent a valid contribution
to NeurIPS literature.

**Main Review:**

The paper is clear and well written.

The analysis is carried out using the replica technique,
with a quenched disorder given by the random
regular graph ensemble. Couplings are uniform and ferromagnetic.
The paper mostly builds on the analyses of Ref. [7] and Ref. [25]

- The authors show the mismatched L1-linR is a consistent
  estimator with performance similar to L1-logR

- The authors provide the “typical” sample complexity, as opposed to
  worst-case bounds given in the literature

- They provide nonasymptotic results replacing an expectation
  with sampling in the state evolution equations.

- The authors manage to carry out a quite impressive replica calculation,
  relying on a few ansatz and approximations that could be of general
  interest.

- They numerically validate their theoretical predictions for recall and precision
  finding excellent agreement with experiments.
  Moreover, also results on 2d grids, which are beyond the scope of the
  technique employed, are surprisingly accurate.


Other comments:


- Results are obtained through a saddle point equation, but some non-asymptotic effects are
are retained using the finite sampling of eq. 25 instead of the full expectation. It is not
clear how these non-asymptotic effects are more relevant than e.g. gaussian fluctuations around
the saddle point that are not taken into account.

- Related to the above question: Does the extremization condition of the free energy functional come from the
large M or N limit or from the zero-temperature limit?
From the appendix I assume the latter, but this could be made more explicit.

- Comment on the large M and N limit at e.g. fixed alpha=M/N. Below eq. 19 the authors argue that 2 different scalings are involved in
the equation of state. It is not clear at this point what is expected in the infinite system size limit, what are
the possible sensible scaling for M and N.

- Computation of eq. 25 is the only one in which not O(1) terms are involved in the EOS.
 Would it be possible to express it analytically through a 1/M expansion instead?

Minor comments:
- Sec 3: clarify that J is a vector (not the full matrix)

- Ansatz 1: specify that the \hat{J} is a random variable over the realization of the dataset
(if I understand correctly)

- Eq. 13, lim \beta \to \infty also on the r.h.s.?

- The authors should comment on the additional complexity given by using different losses, such as the logistic one.
It seems to me that this change would only affect the energetic term in equation 16, and correspondingly the computation
of the F order parameter (and possibly others), without much complexity increase.

- On line 182: (s0, s\psi) seem to be distributed as P_{ising} once the non-active set is marginalized out. It is not clear though why on the
same line this marginal distribution is assumed to be proportional to exp(s0\sum_j∈Ψ J^* j sj) which seems wrong at first sight.
I assume cavity fields coming from the RRG are involved, those fields being zero in the paramagnetic phase.
How to sample from this marginal distribution should be explicit (e.g. after line 447 of the appendix), it's hard for the
reader to dig the relevant piece of information from Ref. [7].

- Also, since J^* is a matrix, the notation needs to be made more consistent.


**Time Spent Reviewing:**

3

---

> ### Author Response · Authors · 2021-08-10
> **Thank you for your review. Response to Reviewer QMbt.**
>
> Thank you for your constructive comments and suggestions. Regarding your concerns, please find our responses below.
>
> $\mathbf{Q1}$: Results are obtained through a saddle point equation, but some non-asymptotic effects are retained using the finite sampling of eq. 25 instead of the full expectation. It is not clear how these non-asymptotic effects are more relevant than e.g. gaussian fluctuations around the saddle point that are not taken into account.
>
> $\mathbf{A1}$: We think the most direct answer to this question is that we are interested in Precision and Recall. By definition, the fluctuation of the estimator has a great impact on these quantities, and the sampling finiteness has a direct influence on the estimator's fluctuation. We think the Gaussian fluctuation in the saddle point also influences the result, but its effect on the probability distribution of the estimator is indirect.
>
>
> $\mathbf{Q2}$: Does the extremization condition of the free energy functional come from the large $M$ or $N$ limit or from the zero-temperature limit? From the appendix, I assume the latter, but this could be made more explicit.
>
> $\mathbf{A2}$: This comes from the large $N$ limit. The zero-temperature limit is used to obtain the energy term $\xi$ in eq (16) (as detailed in appendix A.1), i.e, the min condition w.r.t. $y$ in eq (43) and the result of $\chi$. We will clarify them in the revision.
>
>
> $\mathbf{Q3}$: Comment on the large $M$ and $N$ limit at e.g. fixed $\alpha=M/N$. Below eq. 19 the authors argue that 2 different scalings are involved in the equation of state. It is not clear at this point what is expected in the infinite system size limit, what are the possible sensible scaling for $M$ and $N$.
>
> $\mathbf{A3}$: We consider rather general scalings of $M$ simultaneously. The saddle-point treatment would be correct for any scaling of $M$ as long as $M$ diverges when $N$ goes to infinity. Indeed, at this point, it is still not clear what is the possible sensible scaling for $M, N$ and this is why we keep them explicitly in the free energy functional in eq (13) and the EOS (17). Afterward, we use perturbation analysis of EOS (17), as detailed in Appendix D, to obtain asymptotic relations between different variables in the infinite limit and finally achieve the sensible scaling result for successful structure recovery.
>
>
> $\mathbf{Q4}$: Computation of eq. 25 is the only one in which not O(1) terms are involved in the EOS. Would it be possible to express it analytically through a 1/M expansion instead?
>
> $\mathbf{A4}$: It might be possible, but we could not find a reasonable way to achieve it. If $M \to \infty$, we can obtain an analytical solution as eq (20). However, for finite M, the symmetry between different estimates $\hat{J}_{j}, {j\in\Psi}$ are broken due to statistical fluctuations and thus it is difficult to obtain analytical solutions.
>
>
> $\mathbf{Q5}$: The authors should comment on the additional complexity given by using different losses, such as the logistic one. It seems to me that this change would only affect the energetic term in equation 16, and correspondingly the computation of the $F$ order parameter (and possibly others), without much complexity increase.
>
> $\mathbf{A5}$: You are right. This can be seen from the general result (for M-estimators with general losses) of the free energy in eq (56) of appendix A.3. For square loss, an analytical solution to the min condition w.r.t. $y$ in eq (56) in appendix exits, which simplifies $E, F$, and $\bar{J}$, while for other losses, one might have to compute the solution $\hat{y}$ although without much complexity. We will make this point clearer in the revision.
>
> $\mathbf{Q6}$: On line 182: $(s_0, s_{\Psi})$ seem to be distributed as $P_{\rm{Ising}}$ once the non-active set is marginalized out. It is not clear though why on the same line this marginal distribution is assumed to be proportional to $e^{s_0\sum_{j\in\Psi}J_{j}^{*} s_j}$ which seems wrong at first sight. I assume cavity fields coming from the RRG are involved, those fields being zero in the paramagnetic phase. How to sample from this marginal distribution should be explicit (e.g. after line 447 of the appendix), it's hard for the reader to dig the relevant piece of information from Ref. [7].
>
> $\mathbf{A6}$: You are right, in general, we have to take into account the cavity fields in the marginal distribution. In the present case, however, the paramagnetic assumption simplifies the marginal distribution and finally, it is proportional to $e^{s_0\sum_{j\in\Psi}J_{j}^{*} s_j}$. When $\Psi$ has a small cardinality $d$, we can compute the expectation w.r.t. $(s_0, s_{\Psi})$ in the EOS exactly by exhaustive enumeration. For large $d$, MC methods like the Gibbs sampling method might be used.  We will clarify these points explicitly in the revision.
>
>
> $\mathbf{Q7}$: Minore comments: since $J^*$ is a matrix, the notation needs to be made more consistent; Sec 3: clarify that $J$ is a vector (not the full matrix); specify in Ansatz 1 that the $\hat{J}$ is a random variable over the realization of the dataset; Eq. 13, $\lim \beta \to \infty$ also on the r.h.s..
>
> $\mathbf{A7}$:  Thank you for the suggestions. We will amend them as suggested in the revision.

---

### Official Review · Reviewer_gfUX · 2021-07-19

**Rating:** 5
**Confidence:** 3

**Summary:**

This paper studies the problem of estimating the edge couplings in a Markov random field using ell-one regularized least-squares estimation. It is assumed throughout that all nodes have the same degree and the nonzero coupling have the same value. Using the replica method, the authors derive a set of couple equations of state (EOS), which depend on the underlying parameters of the model (e.g., the degree and coupling strength),  the ratio between the number of samples and the number of nodes, and the spectral density of the covariance of the resulting MRF. By analyzing the solutions to the equations, the authors provide numerical predictions of the reconstruction performance (e.g, precision and recall) in the limit of high dimension. Further analysis of these expressions gives a sharp predicition for the number of samples needed for exact recovery as a function of the the degree and the coupling strength. Additionally, the authors suggest a modification of the EOS that uses the observed data to produce more accurate predictions. Because the analysis depends on the replica method the results are not rigorous. The accuracy of the theoretical analysis is supported by numerical simulations.

**Limitations And Societal Impact:**

These are adequately addressed.

**Main Review:**

This paper provides a sharp analysis of an interesting problem. On the positive side, I think there is value is obtaining working through the details of the replica method for this problem. Also, the modifications used for the moderate dimensional setting are interesting and could be useful for other problem settings as well.

One weakness of the paper is that the results are not rigorous because they depend on the replica method. In the context of NeurIPS, I think there is a higher bar (in terms of the novelty of the insight and level of interest in the particular problem) for acceptance of a non-rigorous theoretical analysis. While I think the results in this paper and worth publishing, it is not clear to me if they meet the very high bar given by NeurIPS.

I am also a bit unclear about whether the spectral density used in (18) is something that is known in practice. In Appendix F it seems the authors derive this density for the problem of interest, but I am not sure if there are any more assumptions that are needed here. In any event, I think it would help to say a bit more what is and is not known about this density in the main text since it plays a prominent role in the analysis.

Some technical comments and questions are the following:

- In Ansantz 1 the term \bar{J} is defined as the the mean-valued of the estimator. I am not sure what this means. Is this the conditional mean given that j is in the neighborhood of index 0?

- It seems that the discussion of the CLT  in lines 172-174 is meaningful only if the degree is large.

- In the context of Ansantz 1, does the expression (21) also describe the noise term for the estimator in the active set?

- With respect to the upper bound on lambda discussed in lines 235-237, would it be possible to consider a two-stage approach where one the final estimate is based on adding certain ``residuals'' to the estimates obtained from the ell-one regularized problem and then thresholding a second time. Such approaches have been shown to work for support recovery in sparse linear regression problems.

- How the regular graphs sampled for the numerical experiments?

- What are the values of alpha used in Figure 1?

**Time Spent Reviewing:**

3

---

> ### Author Response · Authors · 2021-08-10
> **Thank you for your review. Response to Reviewer gfUX.**
>
> Thank you for your constructive comments and suggestions. Regarding your concerns, please find our responses below.
>
> $\mathbf{Q1}$: One weakness of the paper is that the results are not rigorous because they depend on the replica method. In the context of NeurIPS, I think there is a higher bar (in terms of the novelty of the insight and level of interest in the particular problem) for acceptance of a non-rigorous theoretical analysis. While I think the results in this paper and worth publishing, it is not clear to me if they meet the very high bar given by NeurIPS.
>
> $\mathbf{A1}$: We agree that there is a lack of mathematical rigor in our current results by using the replica method and we also have acknowledged this point in the manuscript. Nevertheless, our results are the first results that provide an accurate estimate of the  typical sample complexity for Ising model selection, and a computationally efficient method to precisely predict the typical learning performances, namely, the precision and recall rates, for both of $\ell_1$-LinR and $\ell_1$-LogR (also applicable to other M-estimators with a general loss) with moderate $M$ and $N$. Such sharp analysis of the _typical_ learning performances of Ising model selection is _novel_ and _unavailable_ before. In addition, we demonstrate, _for the first time_, the success of quadratic loss for Ising model selection in the _whole_ paramagnetic regime, which is highly nontrivial and perhaps surprising, given the apparent discrepancy of quadratic loss between the true log-likelihood.  In practice model mismatch is inevitable and thus it is interesting to study the typical learning behavior under model mismatch. Our study shows that model match is not essential at least for some specific tasks and provides some insights for the general setting.  While it is an open problem to provide a rigorous proof, extensive numerical simulations are conducted which demonstrate that our theoretical predictions are fairly accurate even at moderately small sizes.
>
> Regarding your concern of the unrigorous replica method, we would like to further remark that it has actually been widely used in the analysis of a variety of machine learning problems and shown remarkable efficacy from early the days of research, e.g. [R1-R2], until more recent works in NeurIPS and ICML, e.g. [R3-R7]. Please also refer to our response A1 to reviewer 8TQ3 for a rising recognition of the interplay between statistical physics methods (of course not restricted to replica method) and machine learning. There are various scientific discoveries in various disciplines first derived from the replica method and later rigorously proved to be exact [R8].
>
>
> References
>
> [R1] _Engel, Andreas, and Christian Van den Broeck. Statistical mechanics of learning. Cambridge University Press, 2001._
>
> [R2 ] _Seung, H. S., Sompolinsky, H., & Tishby, N. (1992). Statistical mechanics of learning from examples. Physical review A, 45(8), 6056._
>
> [R3] _Sarao Mannelli Stefano, Giulio Biroli, Chiara Cammarota, Florent Krzakala, Pierfrancesco Urbani, et al.. Complex Dynamics in Simple Neural Networks: Understanding Gradient Flow in Phase Retrieval. 2020 Conference on Neural Information Processing Systems (NeurIPS 2020), Dec 2020, Vancouver, Canada._
>
> [R4] _d’Ascoli, S., Refinetti, M., Biroli, G., & Krzakala, F. Double trouble in double descent: Bias and variance (s) in the lazy regime. In International Conference on Machine Learning (ICML), 2020._
>
> [R5] _Gerace, F., Loureiro, B., Krzakala, F., Mézard, M., & Zdeborová, L. Generalisation error in learning with random features and the hidden manifold model. In International Conference on Machine Learning (ICML), 2020._
>
> [R6] _Bordelon, Blake, Abdulkadir Canatar, and Cengiz Pehlevan. "Spectrum dependent learning curves in kernel regression and wide neural networks." In International Conference on Machine Learning (ICML), 2020._
>
> [R7] _Canatar, A., Bordelon, B., & Pehlevan, C.. Spectral bias and task-model alignment explain generalization in kernel regression and infinitely wide neural networks. Nature communications, 12(1), 1-12, 2021._
>
> [R8] _Marc Mézard, Giorgio Parisi, and Miguel Virasoro. Spin glass theory and beyond: An Introduction to the Replica Method and Its Applications, volume 9. World Scientific Publishing Company, 1987._
>
> $\mathbf{Q2}$: I am also a bit unclear about whether the spectral density used in (18) is something that is known in practice. In Appendix F it seems the authors derive this density for the problem of interest, but I am not sure if there are any more assumptions that are needed here. In any event, I think it would help to say a bit more what is and is not known about this density in the main text since it plays a prominent role in the analysis.
>
> $\mathbf{A2}$: There are two issues to be noted. One is about the formula connecting the performance of the estimator and the spectral density, and the other is the numeric values of quantities that are computed from the formula. For the first point, no assumption about the spectral density is needed to obtain the formula itself and this formula is valid when the graph structure is tree-like and the Ising model defined on the graph is in the paramagnetic phase. For the second point, we need the specific form of the spectral density to obtain numeric solutions in general. As a demonstration, we assume the random regular graph with constant coupling strength for which the spectral density can be obtained analytically as has already been known before in [39]. In the revision, we will state these points clearly.
>
>
> $\mathbf{Q3}$: In Ansatz 1 the term $\bar{J}$ is defined as the mean value of the estimator. I am not sure what this means. Is this the conditional mean given that $j$ is in the neighborhood of index 0?
>
> $\mathbf{A3}$:  This assumes that the mean of the estimator is zero if $j$ is not in the neighborhood $\Psi$ of index 0, while it may take nonzero values if $j$ is in the neighborhood.
>
>
> $\mathbf{Q4}$: It seems that the discussion of the CLT in lines 172-174 is meaningful only if the degree is large.
>
> $\mathbf{A4}$: No. This is meaningful even if the degree is not large, as long as the graph structure is tree-like and the paramagnetic assumption is valid. The key point is that, what we model as Gaussian is the local field $h_w^a$ which, by definition (line 172), is a sum of a large number (of order $N$) of random variables with zero mean, as illustrated in Ansatz 1. This is developed by Abbara et al [7] to tackle the sparse coupling case.
>
> $\mathbf{Q5}$: In the context of Ansatz 1, does the expression (21) also describe the noise term for the estimator in the active set?
>
> $\mathbf{A5}$:  No. This is valid only for the inactive set.
>
> $\mathbf{Q6}$: With respect to the upper bound on lambda discussed in lines 235-237, would it be possible to consider a two-stage approach where one the final estimate is based on adding certain ``residuals'' to the estimates obtained from the ell-one regularized problem and then thresholding a second time.
>
> $\mathbf{A6}$: Yes, you are right. Debiasing by applying certain appropriate operations to the $\ell_1$ estimator is possible. But this is beyond the scope of this manuscript since we only focus on $\ell_1$ estimators without post-thresholding, in a similar way as that of $\ell_1$-LogR in Ravikumar et al.
>
> $\mathbf{Q7}$: How are the regular graphs sampled for the numerical experiments?
>
> $\mathbf{A7}$: Ideally, by a uniform sampling over all the regular graphs. However, it is non-trivial to  "exactly” realize the uniform sampling in practice. In the numerical experiments, we resorted to a widely used numerical package "NetworkX”.
>
>
> $\mathbf{Q8}$: What are the values of $\alpha$ used in Figure 1?
>
> $\mathbf{A8}$: $\alpha=M/N$ where $M$ is the number of samples while $N$ is the number of variables/spins. The x-axis of Figure 1 indicates the values of $\alpha$.

---

> > ### Author Response · Authors · 2021-09-02
> > **Feedback before discussion ends**
> >
> > Dear Reviewer gfUX,
> >
> > First of all, we would like to thank you again for your time and efforts in reviewing our manuscript.
> >
> > As the discussion period is expected to end soon, we look forward to your feedback on whether our rebuttals have addressed your previous concerns. We are happy to discuss if you still have any other concerns. Thank you very much.
> >
> > Best regards,
> >
> > Authors

---

> > > ### Comment · Reviewer_gfUX · 2021-09-12
> > > **feedback**
> > >
> > > Thank you for responding to my comments. I should be been more clear that I think the replica analysis is valuable and interesting. Overall, I think the result is nice. My score reflects that the presentation of the paper was very tough for me to follow. I was unable to determine which parts of the analysis were followed from the assumptions of the analysis and which parts could be verified rigorously. I understand that there is a challenge in communicating ideas that may be standard to some audiences (e.g. statistical physics) to the broader ML community. In this case, additional technical details for the arguments would have been helpful for me.

---

> > > > ### Author Response · Authors · 2021-09-15
> > > > **Thank you for your feedback.**
> > > >
> > > > Thank you very much for your feedback. We appreciate your clarification of the concern and positive comments on our results. We will do our best to improve the clarity in the revision to make it understandable to the broader community unfamiliar with the replica method as much as possible.

---

### Official Review · Reviewer_8TQ3 · 2021-07-20

**Rating:** 7
**Confidence:** 2

**Summary:**

The submission considers the problem of recovering the underlying graph structure of an Ising model defined on N nodes, given M i.i.d. samples.
More specifically, the authors focus on Ising models associated with typical random regular graphs in the paramagnetic phase.
Ideas from statistical physics are borrowed to study the performance of an estimator derived from an $\ell_1$-regularized linear regression problem.
It is claimed that using $M = \mathcal{O}(\log(N))$ samples is enough to obtain model selection consistency.
Non-asymptotic results are also given, and some numerical experiments are presented to support the findings.

**Ethical Concerns:**

NTR

**Limitations And Societal Impact:**

NTR

**Main Review:**

### Originality

#### Are the tasks or methods new?

Using an $\ell_1$-regularized linear regression estimator to recover the underlying graph structure of Ising models

#### Is the work a novel combination of well-known techniques? (This can be valuable!)

The authors borrow classical ideas from statistical physics (replica method, free energy) and optimization (leave one out) to study the performance of their estimator

#### Is it clear how this work differs from previous contributions?

The difference with previous work is explained in the introduction section.

#### Is related work adequately cited?

Related work is adequately cited, but the ideas and notions from statistical physics must be explained more carefully for the ML community to integrate it and build on it.
Please give references to the sampling algorithms used to generate the datasets $\mathcal{D}_M$

---

### Quality

#### Is the submission technically sound?

I didn't carefully check the mathematical details.

#### Are claims well supported (e.g., by theoretical analysis or experimental results)?

The supplementary material contains the proofs of the claims.

#### Are the methods used appropriate?

The methods used seem appropriate.

#### Is this a complete piece of work or work in progress?

The submission is work in progress regarding the scope of applicability of the results.

#### Are the authors careful and honest about evaluating both the strengths and weaknesses of their work?

It is not clear whether the assumptions made (random regular graphs, paramagnetic phase) are very restrictive and how these assumptions differ from other works like l1-LogR estimator for example.

---

### Clarity

#### Is the submission clearly written?

The paper is well written.

#### Does it adequately inform the reader? (Note that a superbly written paper provides enough information for an expert reader to reproduce its results.)

Please

* define what a random regular graph with coupling strength $K_0$ is
* explain the choice for the specific class of random regular graphs in the paramagnetic phase
* define the notions of paramagnetic phase and model consistency earlier
* detail the misspecification of the l1-LinR model
* define what a 'typical' random regular graph is. It seems a critical insight to call for the replica method (l148)
* explain why the overall analysis is more difficult with a larger class of graphs than 'typical' random regular graphs
* explain why Ansatz 2 can only be partly checked, and if it is a
* give more details on the free energy and explain why this is a key quantity
* clarify what l264 "spin snapshots are obtained with MC sampling". Are the samples exactly distributed according to the target Ising model or are they obtained using an MCMC techniques.
* clarify the mismatch between the claim l267 "to obtain standard error bars, we repeat the sequence of operations many times" and Figure 1 were the "error bars were obtained with only 5 runs".

* define the notations
  + $J_j$ l164
  + $\| \cdot \|_1$ equation (2)
  + $Extr$ l182
* clarify l193 "$I$ in (15) is equal to $[I]_O$"
* consider adopting the notation $m$ in place of $\mu$
* make the attributes of Figure 2 bigger

---

### Significance

#### Are the results important?

It is an interesting study of a natural estimator derived from an l1-regularized linear regression problem for the Ising model selection problem.

#### Are others (researchers or practitioners) likely to use the ideas or build on them?

The presentation of the ideas borrowed from statistical physics lacks of clarity for the ML community to build on it.
Besides, the emphasis l70 that this work "might be of particular interest for the statistical mechanics community" may seem surprising for a submission to a ML conference.

**Time Spent Reviewing:**

4

---

> ### Author Response · Authors · 2021-08-10
> **Thank you for your review. Response to Reviewer 8TQ3.**
>
> Thank you for your constructive comments and suggestions. Regarding your concerns, please find our responses below.
>
> $\mathbf{Q1}$: Ideas and notions from statistical physics must be explained more carefully for the ML community to integrate it and build on it.
> The presentation of the ideas borrowed from statistical physics lacks of clarity for the ML community to build on it. Besides, the emphasis l70 that this work  ``might be of particular interest for the statistical mechanics community" may seem surprising for submission to an ML conference.
>
> $\mathbf{A1}$: We will add sentences explaining the statistical physics ideas and notions more clearly in the revision. Regarding l70, we will modify accordingly and the point is that we provide some technical advances in the free energy computation using the replica method, which might be of interest to those who use statistical mechanics as a tool to analyze the related machine learning problems.
>
> We agree that ideas from statistical mechanics might seem unfamiliar to some part of ML community, but it is worth pointing out that statistical mechanics and machine learning are closely related to each other from the very beginning of the study of ML problems, e.g., perceptron, Boltzmann machine, neural networks, etc. Statistical mechanics has provided a powerful way to understand the basic principles of learning, even that of deep learning, please refer to [R1-R3] and [19]. Given the intrinsic connections, it is important to borrow and learn ideas from different fields to tackle challenging ML problems. In the ML community,  there have already been such active events with notable examples in [R4-R6]. Nevertheless, we will manage to revise the manuscript accordingly to make it as easy to understand as possible.
>
>
> $\mathbf{Q2}$: Add references of the sampling algorithms to generate the data. Clarify what l264  ``spin snapshots are obtained with MC sampling". Are the samples exactly distributed according to the target Ising model or are they obtained using MCMC techniques?
>
> $\mathbf{A2}$: We use the Metropolis-Hastings algorithm [R7-R9] to generate the datasets in a similar way as [7].  We will add the references and clarify how we generate the dataset in the revision.
>
>
> $\mathbf{Q3}$: It is not clear whether the assumptions made (random regular graphs, paramagnetic phase) are very restrictive and how these assumptions differ from other works like $\ell_1$-LogR estimator for example.
>
> $\mathbf{A3}$: We have briefly stated our limitations in section 5 (line 301-304) and we will make it clearer in the revision. We will also add discussions on differences in our assumptions from other works. It is worth noting that the restrictive paramagnetic assumption is not only limited to $\ell_1$-LinR, but also to other low-complexity estimators like $\ell_1$-LogR unless post-thresholding is used, as reported in Montanari et al [R10].
>
>
> $\mathbf{Q4}$: Explain why the overall analysis is more difficult with a larger class of graphs than 'typical' random regular graphs.
>
> $\mathbf{A4}$: The property essential for the analysis is that the graph structure is tree-like. If this is not correct, the ansatz 1 and the following assumption that the noise part of the local field is Gaussian are not justified. Some non-typical realizations from the random regular graph ensemble and some representative regular graphs such as grid graphs can have many short loops and thus our analysis is not exact for them, though it can be a good approximation. In addition to this essential problem, some technical problems arise in evaluating the eigenvalue distribution $\rho (\lambda)$ of the covariance matrix C and also in assessing the marginal distribution $P_{Ising}(s_0,s_{\Psi}|J^*)$, both of which are necessary quantities when solving the EOS.
>
>
> $\mathbf{Q5}$: Explain why Ansatz 2 can only be partly checked, and if it is a...
>
> $\mathbf{A5}$: It is partly checked since we only provide a numerical check of the consistency. We will change the expression from "partly checked” to  "numerically checked” in the revision.
>
> $\mathbf{Q6}$: Give more details on the free energy and explain why this is a key quantity.
>
> $\mathbf{A6}$: We will add more details and explanations of free energy in the revision. In short, the reason why the free energy is the key quantity is that it plays the role of the cumulant generating function. Namely, once the free energy is computed, one can evaluate averages of various quantities just by taking its derivatives with respect to external fields.
>
> $\mathbf{Q7}$: Define what a "typical" random regular graph is. It seems a critical insight to call for the replica method (l148).
>
> $\mathbf{A7}$: When seen from a randomly fixed node, a huge regular random graph looks like part of an infinite tree. This is the case for typical realizations from the uniform probability distribution on all the regular graphs. There are non-typical realizations that do not exhibit this "treeness’’ property, but the probability that such realizations appear tends to zero in the large $N$ limit. What we mean by "typical" is that we only consider such typical realizations having the "treeness" property. In the revision, we will state this point more clearly.
>
> $\mathbf{Q8}$: Remaining suggestions on the clarity.
>
> $\mathbf{A8}$: Thanks for the suggestions. We will happily amend these points as suggested in the revision.
>
>
>
> References:
>
> [R1] Engel, Andreas, and Christian Van den Broeck. Statistical mechanics of learning. Cambridge University Press, 2001.
>
> [R2] Zdeborová, Lenka. Understanding deep learning is also a job for physicists. Nature Physics 16.6 (2020): 602-604.
>
> [R3] Bahri, Y., Kadmon, J., Pennington, J., Schoenholz, S. S., Sohl-Dickstein, J., & Ganguli, S. (2020). Statistical mechanics of deep learning. Annual Review of Condensed Matter Physics, 11, 501-528.
>
> [R4] Theoretical physics for deep learning. Workshop at the 36th International Conference on Machine Learning https://go.nature.com/36gSRDb (ICML, 2019).
>
> [R5] Machine learning and the physical sciences. Workshop at the 33rd Conference on Neural Information Processing Systems https://go.nature.com/2Xd16w1 (NeurIPS, 2019).
>
> [R6] Statistical physics of machine learning. Workshop at the International Centre for Theoretical Sciences (ICTS), https://www.icts.res.in/discussion-meeting/SPMML2020, 2020.
>
> [R7] N. Metropolis, A. Rosenbluth, M. Rosenbluth, A. Teller, and E. Teller. Equations of state calculations by fast computing machines. J. of Chem. Phys., 21:1087–1092, 1953.
>
> [R8] W. K. Hastings, Monte Carlo sampling methods using Markov chains and their applications, Oxford University Press, 1970.
>
> [R9] S. Geman and D. Geman, “Stochastic relaxation, Gibbs distributions, and the bayesian restoration of images,” IEEE Transactions on pattern analysis and machine intelligence, no. 6, pp. 721–741, 1984.
>
> [R10] Montanari, A.; and Pereira, J. A. Which graphical models are difficult to learn? In Advances in Neural Information Processing Systems, 1303–1311, 2009.

---

> > ### Comment · Reviewer_8TQ3 · 2021-08-20
> > **Final rating**
> >
> > As a non-expert, I was convinced by the other reviewers and the authors' response to raise my rating from 5 to 7.
> > Note, in the experimental part, it is important to increase the number of runs to get more accurate results.

---

> > > ### Author Response · Authors · 2021-08-24
> > > **Thank you for your positive comments.**
> > >
> > > Thank you for your positive comments. In the revision, we will increase the number of runs in the experimental part as suggested.

---

### Official Review · Reviewer_Y3wG · 2021-08-02

**Rating:** 8
**Confidence:** 4

**Summary:**

The authors study the Ising model selection problem and compute analytically via the replica method the typical performances of the $\ell_1$-regularized linear regression. They confirm the theoretical result by running a good number of numerical simulations.
They demonstrate the sample complexity is such that O(log N) samples are enough to correctly identify the right neighborhood of a generic variable.
The theory presented is able to predict the precision and the recall rates also for finite values of the system size and a small number of samples. Moreover, the theory seems to work fine also for graphs with many loops.
The methodology of the present work can be extended also to other estimators.

**Limitations And Societal Impact:**

I strongly suggest the authors discuss more the limitations of their method.
In particular, what happens approaching the critical point.

I don't see any potential negative societal impact.

**Main Review:**

The present work looks scientifically valid and robust. The authors seem to master the replica computation. The results obtained are very original and are presented in a very clear way.
I consider these results very important as they show how the replica method can be used to compute the relevant estimators and the sample complexity. The numerical simulations run for not too large values of the system size and number of samples convincingly support the validity of the theory.

I have only one main concern. The quadratic loss function does not look like the appropriate function to minimize in order to correctly infer the interaction graph. Indeed on variables where the field is very large, even aligning the spin to the field, one gets a very large penalty.
Contrary to logistic regression and interaction screening, where the loss functions go to zero at large values of the argument.
The authors should explain how and why the quadratic loss function that penalizes a spin that aligns with a very strong field can provide the right answer to the Ising model selection problem.
Is it maybe because the authors are considering only the paramagnetic phase and so the fields are not very large? If it is so, it would be worth discussing it, also to clarify the limits of this approach.

In the numerical simulations, the authors study only the spin-glass model, that is couplings with a random sign.
Why not studying also the ferromagnetic model? In the latter case probably the coupling constant $K_0$ could be made larger and the theory tested also in the case where the model has a ferromagnetic phase.

The results for LinR and LogR are always so close maybe because the authors are studying the problem in a regime where the 2 losses are very similar. It would be very interesting to see what happens in the other regimes. How different would be the two estimators and which one is closer to the truth.

Minor point: in several places the authors write "intersection screening" instead of "interaction screening"

**Time Spent Reviewing:**

3

---

> ### Author Response · Authors · 2021-08-10
> **Thank you for your review. Response to Reviewer Y3wG.**
>
> Thank you for your positive comments and constructive suggestions. Regarding your concerns, please find our responses below.
>
> $\mathbf{Q1}$: I have only one main concern. The quadratic loss function does not look like the appropriate function to minimize in order to correctly infer the interaction graph. Indeed on variables where the field is very large, even aligning the spin to the field, one gets a very large penalty. Contrary to logistic regression and interaction screening, where the loss functions go to zero at large values of the argument. The authors should explain how and why the quadratic loss function that penalizes a spin that aligns with a very strong field can provide the right answer to the Ising model selection problem. Is it maybe because the authors are considering only the paramagnetic phase and so the fields are not very large? If it is so, it would be worth discussing it, also to clarify the limits of this approach.
>
> $\mathbf{A1}$: ​​First of all, we emphasize that the success of the quadratic loss is highly non-trivial even in the paramagnetic phase. However, in the paramagnetic phase of tree-like models, this can be understood intuitively as follows. On average, the condition for the $\ell_1$-LinR estimator is given as
>
> $<s_0s_k> - \sum_{j \ne 0} <s_j s_k> J_j  = \lambda \partial |J_k|$, ($k = 1,\ldots, N$)
>
> where $<\cdots>$ and $\partial |J_k|$ represent average with respect to the Boltzmann distribution (7) and the sub-gradient of $|J_k|$, respectively. In the paramagnetic phase, $<s_is_j>$ decays exponentially with respect to the distance $d_{ij}$ between sites $i$ and $j$ as $\tanh^{d_{ij}}(K_0)$ in its magnitude. This guarantees that once the connections $J_k$ of sites in the first nearest neighbor set $\Psi$ are given so that
>
> $<s_0s_k> = \sum_{j \in \Psi} <s_j s_k> J_j + \lambda \rm{sign} (J_k)  (\forall{k} \in \Psi)$  (*)
>
> holds, the other conditions are automatically satisfied by setting all the other connections that are not from $\Psi$ to zero.  For appropriate choices of $\lambda$, ( * ) has solutions of  $\rm{sign}(J_k^*)J_k>0, \forall{k} \in \Psi$. Namely $\forall{k} \in \Psi$, the estimate $(J_k)$ has the same sign as the true value $J_k^*$. These conclude that on average the $\ell_1$-LinR estimator successfully recovers the network structure up to the connection signs if $\lambda$ is chosen appropriately. Although this argument is based on the properties of tree-like models, the results of experiments on 2D square lattices shown in Fig. 1 indicate that it also holds, at least in a good approximation level, even if graphs include cycles.
>
> The key of the above argument is that $<s_is_j>$ decays exponentially with respect to the distance of two sites, which does not hold after the phase transition. Actually, we have a preliminary result that $\ell_1$-LinR starts to fail in the network recovery just at the phase transition point. This is not limited to $\ell_1$-LinR; the matched estimator $\ell_1$-LogR also exhibits similar behavior unless post-thresholding is used, as reported in Montanari et al [R1]. However, we would like to leave its details for another occasion as the analysis is rather complicated and is difficult to include in the current manuscript. We will add explanations and clarify the limitations of our method in the revision.
>
> [R1] Montanari, A.; and Pereira, J. A. Which graphical models are difficult to learn? In Advances in Neural Information Processing Systems, 1303–1311, 2009.
>
>
> $\mathbf{Q2}$: In the numerical simulations, the authors study only the spin-glass model, that is couplings with a random sign. Why not studying also the ferromagnetic model? In the latter case probably the coupling constant could be made larger and the theory tested also in the case where the model has a ferromagnetic phase.
>
> $\mathbf{A2}$: There is no special reason to use the spin-glass model, since both the spin-glass and ferromagnetic models essentially show the same behavior in the paramagnetic phase on tree-like graphs. Note that in simulations for 2D grid, the ferromagnetic mode is studied. Enhancing the coupling constant will be interesting especially in the context related to phase transitions, whose detailed study is ongoing.
>
>
> $\mathbf{Q3}$: The results for LinR and LogR are always so close maybe because the authors are studying the problem in a regime where the 2 losses are very similar. It would be very interesting to see what happens in the other regimes. How different would be the two estimators and which one is closer to the truth.
>
> $\mathbf{A3}$: The two estimators actually behave differently. Please see, for example, Figure 6 in the supplementary material. The RSS is much smaller for LogR in this plot, which is natural since the estimated values on the active set of LogR is closer to the true ones compared to those of LinR: the difference between the matched and mismatched cases clearly appears here. Note that the LinR’s estimate values never accord with the true one even when the dataset is infinitely large. We will clarify this in the revision.
>
> $\mathbf{Q4}$: I strongly suggest the authors discuss more the limitations of their method. In particular, what happens approaching the critical point.
>
> $\mathbf{A4}$: Thanks for the suggestion. We will add more discussions in the revision and clarify the limitations of our method clearly in the revision.
>
>
> $\mathbf{Q5}$: Minor point: in several places the authors write  ``intersection screening" instead of "interaction screening"
>
> $\mathbf{A5}$: We will amend it in the revision.

---

> > ### Comment · Reviewer_Y3wG · 2021-08-20
> > **Useful reply**
> >
> > I find the reply by the authors very useful, especially A1 which explains why the mismatched LinR can recover correctly the network. I believe that adding this information to the manuscript will be very valuable for the readers.

---

> > > ### Author Response · Authors · 2021-08-24
> > > **Thank you for your positive comments.**
> > >
> > > Thank you for your positive comments and we will happily add this information in the revision.

---

### Decision · Program_Chairs · 2021-09-27

**Decision:**

Accept (Poster)

**Comment:**

The authors in this paper study the learning of a Boltzmann machine, aka the Inverse Ising model, a classical problem in graphical models and Markov Random Fields. They propose an analysis of the performance of the l1-regularised linear regression estimator in finding the underlying non-zero coefficients. The analysis is performed using a non-rigourous but powerful heuristic approach from statistical mechanics, the replica method, that has been used in many other machine learning problems, in idealistic situations. The author confirmed the theoretical result by running a large number of numerical simulations. They show the sample complexity is such that O(log N) samples are enough to correctly identify the right neighbourhood of a generic variable. The theory presented seems to be powerful enough to predict both  the precision and the recall rates for finite dimension, and seems to give fairly good prediction for graphs with many loops. Interestingly, the methodology of the present work can be apparently generalised and extended also to other estimators.

The review process was intense, with six reviewer and a large number of forum replies. All reviewers testified of the quality of presented results. From a technical point of view, computation contained in this manuscript was found to  involved and complex, yet it was found provides a non-trivial analytical result. The comparison with the numerics however strongly supports the claim that the replica computation is giving the correct answer, even though it is not rigorous. This paper was thus judged to represent a non-trivial contribution from statistical physics to machine learning. However, the use of the non-rigorous replica method and its acrobatic non-rigorous mathematics was judged to be sometimes dazzling.

The rebuttal saw a number of discussions, and most reviewers agreed that the paper was impressive, and score were increased during the process. A criticism that remained, though, from a minority of reviewers, was that the derivation is likely inaccessible for non-experts. While this is true, it can be said of many theoretical at Neurips, including rigorous ones, and there is a long standing tradition of welcoming such non-rigorous paper at Neurips.

Given the agreement on the quality of the results, the very good ranking & grading of the paper, the extensive numerical simulations that confirm the validity of theory, we believe the paper to be largely worthy of a publication at a venue like Neurips, and recommend acceptance.